# Multiple associative structures created by reinforcement and incidental statistical learning mechanisms

Miriam C. Klein-Flügge [1,2]*, Marco K. Wittmann [1,2], Anna Shpektor[2], Daria E.A. Jensen [1,2,3] & Matthew F.S. Rushworth[1,2]

Learning the structure of the world can be driven by reinforcement but also occurs incidentally through experience. Reinforcement learning theory has provided insight into how prediction errors drive updates in beliefs but less attention has been paid to the knowledge resulting from such learning. Here we contrast associative structures formed through reinforcement and experience of task statistics. BOLD neuroimaging in human volunteers demonstrates rigid representations of rewarded sequences in temporal pole and posterior orbito-frontal cortex, which are constructed backwards from reward. By contrast, medial prefrontal cortex and a hippocampal-amygdala border region carry reward-related knowledge but also flexible statistical knowledge of the currently relevant task model. Intriguingly, ventral striatum encodes prediction error responses but not the full RL- or statistically derived task knowledge. In summary, representations of task knowledge are derived via multiple learning processes operating at different time scales that are associated with partially overlapping and partially specialized anatomical regions.

---

[1] Department of Experimental Psychology, University of Oxford, Tinsley Building, Mansfield Road, Oxford OX1 3TA, UK. [2] Wellcome Centre for Integrative Neuroimaging (WIN), University of Oxford, Nuffield Department of Clinical Neurosciences, Level 6, West Wing, John Radcliffe Hospital, Oxford OX3 9DU, UK. [3] Department of Psychiatry, University of Oxford, Warneford Hospital, Oxford OX3 7JX, UK. *email: miriam.klein-flugge@psy.ox.ac.uk

The ability to learn about the consequences of our actions is critical for survival. Reinforcement learning (RL) provides a framework for formally studying learning and has inspired a major reconceptualization of its underlying neural mechanisms.

In most studies of reward-based learning, one or several stimuli are associated with varying amounts of reward and their values have to be inferred and updated over time. Consequently, two neural representations are of interest: the encoding of the expected reward at the time of the stimulus, and the deviation from the expected outcome, or prediction error (PE), at the time of the outcome. The majority of research to date has focused on the PE at the time of outcome, present in dopamine neurons in the ventral tegmental area (VTA) and its projection area ventral striatum[1–4], and shown they satisfy PE axioms[5,6], reflect task context and task demands[7–11], signal value and identity deviations[12,13] as well as social information[14,15].

There has also been particular interest in two classes of RL often referred to as model-free and model-based[16] but their underlying neural mechanisms are less well understood. In model-free RL actions or stimuli are directly reinforced by subsequent delivery of reward while in model-based learning actions and stimuli are reinforced in a manner that reflects knowledge of task structure. While model-free and model-based learning can be distinguished at a behavioral level, it has been difficult to associate them with different neural structures or processes. Suggestions that model-free and model-based learning might be linked to striatum and prefrontal cortex respectively have not been supported empirically[16,17]; model-based and model-free PEs were co-localized in ventral striatum.

Here we took an alternative approach to investigate the neural mechanisms associated with model-free and model-based learning. Rather than focusing on the PEs that occur during learning, we focused on the knowledge or associative structures that are formed by the learning process. In order to do this, participants learned not just associations between a single stimulus and reward but between chains of stimuli leading to reward[2,18,19] as well as the statistical relationships between stimuli regardless of reward. This allowed us to test whether the associative structures derived from different learning processes might prove more distinguishable than their PEs. We therefore undertook two main series of analyses of behavior and neural activity that contrasted knowledge learned from RL versus statistical relationships. We also tested whether RL-acquired knowledge may be static and inflexible[20] compared to the cognitive maps formed flexibly through statistical learning[21].

Another computation that arises when multiple actions or states occur before encountering reward is to establish which stimuli or actions are eligible to be assigned reward. Recent work suggests that the amygdala might mediate this process through a mechanism that spreads reward to stimuli in close temporal proximity[22–24]. Here we asked whether this process might also be present in space, i.e. whether eligibility might initially be assigned to stimuli in spatial proximity to reward.

Participants attended two sessions prior to scanning to learn associations between stimulus sequences and reward. During the fMRI experiment, we then probed the neural correlates of the associative structures that they had constructed in these behavioral sessions. We provide evidence for dissociable brain networks that represent associations derived from RL versus statistical learning and which operate at different time scales.

## Results

On separate days prior to scanning, participants ($n = 26$) performed a behavioral task where they learnt to associate a four-step sequence of stimuli (ABCD) with reward (rewarded sequence: RewSeq). Unlike in previous work assessing sequence learning e.g.[25–27], however, the sequence was not learnt from its starting element. Instead, participants pressed buttons to move around a $3 \times 4$ grid towards stimuli that were highlighted by the computer one after the other in a continuous stream (Fig. 1a, Supplementary Fig. 1a–c). Importantly, the start of the rewarded sequence was not signaled but occurred after an unpredictable number of other stimuli. When D was reached, a reward appeared, and participants had to infer the rewarded sequence and determine its length through careful and repeated observation. Unusually, this allowed us to test one of the key predictions of RL – that associations between stimuli and reward are constructed by propagating reinforcement from the stimuli immediately prior to reward to earlier stimuli[28]. As a consequence, participants should know the end of the sequence before its beginning, and associations learnt via RL should comprise stimulus–stimulus (e.g., A to B) as well as stimulus–reward relationships (e.g., D to Reward). Unbeknownst to participants, they transitioned through another four-step sequence (A′B′C′D′) equally frequently but this sequence was not rewarded (control sequence: ConSeq). This allowed us to assess (i) knowledge of the stimulus route leading to reward (the RL-acquired knowledge) while controlling for statistical learning, (ii) the existence of incidentally learned statistical knowledge of stimulus transitions which should be present for both RewSeq and ConSeq, and (iii) the possibility of spread of reward effects across space.

**Participants anticipate sequence transitions and reward**. To establish that participants had acquired knowledge of the rewarded sequence in the pre-scan learning task, we first probed, using reaction times (RTs), whether participants were able to predict the occurrence of reward. Participants were instructed to press a button as soon as either reward (a picture of a treasure box) or a similarly frequent but unpredictable control stimulus (a picture of a bridge) appeared on the screen. An $8 \times 2$ repeated-measures ANOVA with factors block and stimulus type (reward/bridge) showed main effects of stimulus type and block and an interaction between stimulus type and block (all $F > 15$, $p < 0.001$, $\eta^2 = 0.95, 0.56, 0.39$, Fig. 1b). RTs were faster for reward, became faster over blocks, and speeding was more prominent across blocks for reward (Bayesian repeated-measures ANOVA: full model P(M|data) = 1, BFm = 6.85e9). This demonstrates that participants were able to link reward occurrence to the preceding stimulus sequence.

Secondly, we probed knowledge of the stimulus–stimulus associations between sequence elements by analyzing RTs to the stimuli preceding the reward. More precisely, we examined RTs to the first movement that initiates transitioning between two stimuli of RewSeq (or ConSeq), e.g. the button press that initiates the path from A to B or A′ to B′ (and similarly B to C and C to D or B′ to C′ and C′ to D′). We reasoned that participants who anticipate A's successor will initiate the first movement towards B faster. Indeed, participants showed overall faster RTs to RewSeq transitions compared to equally frequent ConSeq transitions ($2 \times 3$ repeated-measures ANOVA with factor sequence (RewSeq/ConSeq) and transition (AB,BC,CD); effect of sequence type $F(1,25) = 96.36$, $p < 0.001$, $\eta^2 = 0.79$). RTs also became faster as participants progressed through the sequence (effect of transition: $F(2,50) = 73.68$, $p < 0.001$, $\eta^2 = 0.75$) and this speeding was more pronounced for RewSeq compared to ConSeq, and thus, when approaching reward (interaction transition × sequence type: $F(2,50) = 42.20$, $p < 0.001$, $\eta^2 = 0.63$; Bayesian repeated-measures ANOVA: P(Full Model|Data) = 1, BF = 3.3e33; Fig. 1c). Thus, participants were able to anticipate correct sequence successor

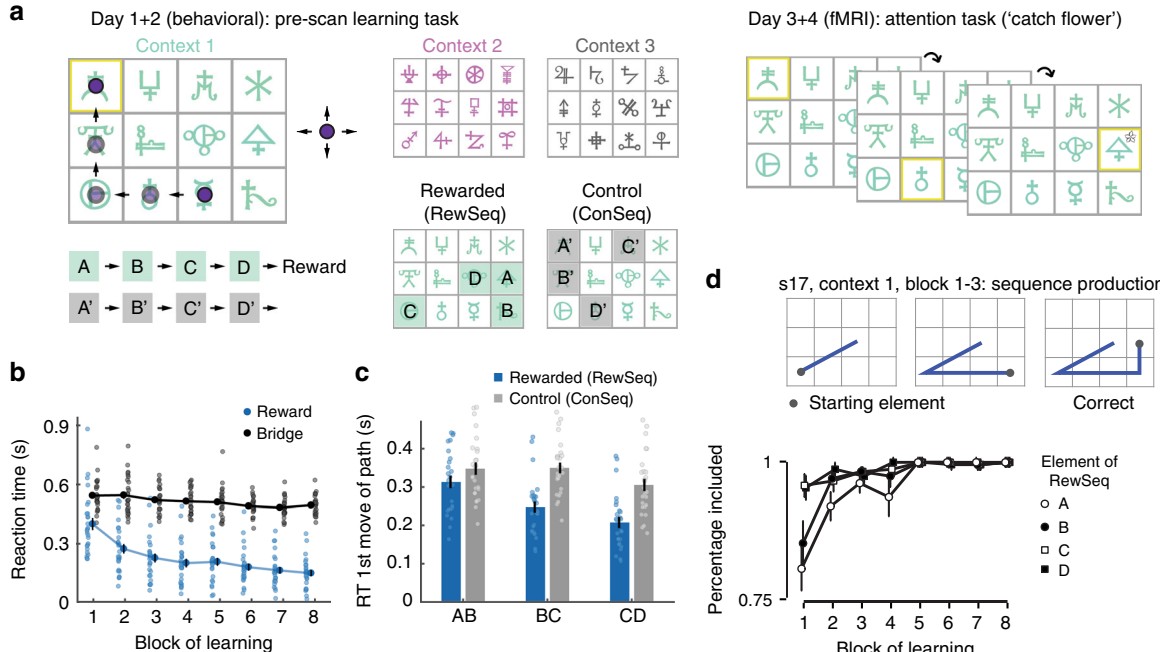

**Fig. 1** Learning of reinforced sequences occurs backwards from reward. **a** Pre-scan learning task (left): on day 1 + 2 of testing, participants moved an agent (purple circle) to highlighted stimuli (yellow) using up/down/left/right buttons. For each of three fixed 3 × 4 layouts (contexts: green, pink, gray), they had to learn a hidden sequence (ABCD) that led to reward. This reinforced sequence (RewSeq) was embedded in a continuous stream of highlighted stimuli and thus learnt by trial and error, and its beginning and length inferred over time. A control sequence (ConSeq) composed of other stimuli (A'B'C'D') was encountered equally frequently but not followed by reward. Scan task (right, day 3 + 4): one stimulus was highlighted after another, but now participants had to press a button when they detected a flower on the highlighted square. By carefully controlling the order of stimuli, we could probe representations of RL-driven sequence knowledge and statistical knowledge. **b** Participants anticipated reward at the end of the sequence: RTs to reward were faster than RTs to a bridge shown at unpredictable times, and reward-RTs became faster over training (error bars = SE; dots = individual participants). **c** Participants RTs for initiating the first movement towards the next stimulus were faster as they progressed through the rewarded sequence (from A to B, B to C, and C to D), and speeding was more pronounced for RewSeq compared to ConSeq. Thus, participants anticipated the correct sequence successor stimulus. Looking at only ConSeq transitions (gray bars), there was a significant effect of transition, suggesting statistical learning in the absence of reward. **d** Sequence knowledge was constructed backwards from reward. (Top) Sequences produced by one participant after the first, second, and third block of the pre-scan learning task. Initially, the sequence was thought to contain two stimuli (CD), but additional stimuli were appended block by block, with full sequence knowledge reached after block 3. (Bottom) Percentage of times the stimuli A/B/C/D were included in the sequence produced after each learning block. C and D were included first, B on average after block 2, and A was included last. Error bars denote SE; see Supplementary Fig. 1

elements. Additional analyses confirmed participants' preference for RewSeq over ConSeq stimuli during choice (Supplementary Methods, Supplementary Note 1 and Supplementary Fig. 1d).

**Sequence knowledge is acquired backwards from reward.** To establish how knowledge of the rewarded sequence was initially acquired, we examined participants' ability to produce the correct sequence. After each pre-scan learning block participants were asked to enter the correct order of stimuli that they thought led to reward. The length of sequences entered by participants ranged between 1 and 7 stimuli after one block of learning but between 4 and 5 stimuli after four blocks. Some participants initially entered the correct last few elements preceding reward (BCD or CD) and then gradually extended the sequence to A. Others over-specified the sequence and appended additional elements to its beginning which they subsequently gradually pruned (e.g. xxABCD). In all cases, the sequence was learnt backwards (for an example participant, see Fig. 1d). To quantitatively assess this, we calculated the percentage of times each of the four elements A, B, C, and D was included in the sequences participants produced during the 8 learning blocks. A 4 × 8 repeated-measures ANOVA revealed effects of element ($p = 0.002$, $F(3,75) = 5.46$, $\eta^2 = 0.18$), block ($p < 0.001$, $F(7,175) = 9.54$, $\eta^2 = 0.27$) and element × block ($p < 0.001$, $F(21,525) = 4.33$, $\eta^2 = 0.15$), with elements C and D included first, element B included on average from block 2, and

the starting element A included last (Bayesian repeated-measures ANOVA: full model $P(M|data) = 0.993$, BFm = 600.70; Fig. 1d, Supplementary Note 2 and Supplementary Fig. 1e, g).

**Behavioral evidence for statistical learning.** Participants also showed evidence for reward-unrelated statistical learning: initiating button presses from A' to B', followed by B' to C' and C' to D' showed progressive RT speeding[29–32] because transitions, despite never being associated with reward, became increasingly predictable (Supplementary Note 3; gray bars in Fig. 1c).

**Neural representations of sequence knowledge.** On separate days within the same week, we invited participants to two MRI sessions to investigate the neural representations that had formed during the initial learning sessions. During scanning, we highlighted one stimulus at a time in pseudo-random order on the same 3 × 4 grids presented during the pre-scan learning task. The participants' task was now simply to press a button when they detected a flower on the highlighted stimulus (10% of trials; catch trials). This focused their attention on the currently relevant stimulus (Fig. 1a). Knowledge of the rewarded sequence was no longer relevant for performing the task.

We first examined differences in BOLD activity between rewarded and control sequence elements, thus probing changes in BOLD that were driven by the presence of reinforcement. We

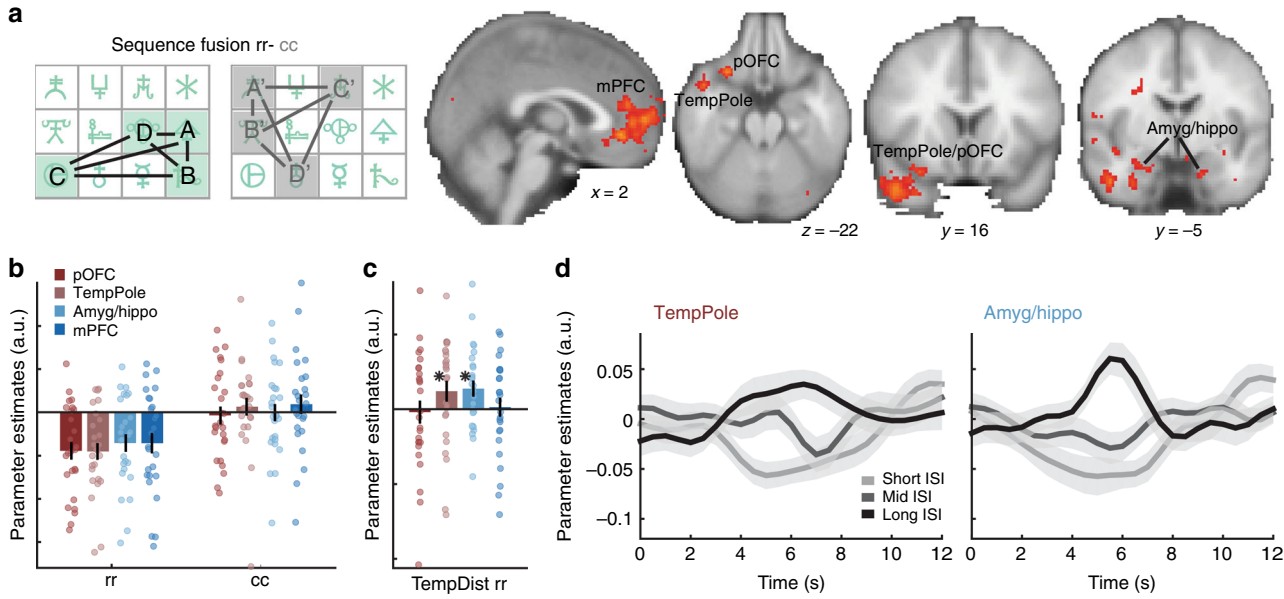

**Fig. 2** Reinforced sequence knowledge in pOFC, temporal pole, amyg/hippo and mPFC. **a** Regions where elements of the rewarded sequence have grown together more in their neural representation than elements of the unrewarded but equally frequent control sequence. Repetition suppression predicts less activation for a RewSeq element that follows another element of RewSeq (i.e., after a pair including two elements of A, B, C, or D: rr), compared to a ConSeq element that follows another element of ConSeq (pairs made up of A′, B′, C′, or D′: cc), if their representations overlap more strongly (i.e., recruit the same neural populations). In this contrast, we identified medial prefrontal cortex (mPFC), temporal pole (tempPole) and (within the same cluster) posterior orbitofrontal cortex (pOFC; all cluster-corrected), and a region at the border of amygdala and hippocampus (amyg/hippo; TFCE-corrected within bilateral amygdala mask; shown at $z > 2.3$ uncorrected). **b** For illustration, parameter estimates are shown in spheres centered on the peak activation in these four regions. **c** Repetition suppression effects in temporal pole and amyg/hippo were modulated by the temporal distance between successive RewSeq elements, with stronger suppression for shorter inter-stimulus intervals (ISI) as predicted (short < 2 s; mid = [2,3.5] s, long > 3.5 s; *$p < 0.05$ in one-sample $t$-test). **d** The raw BOLD time-series in temporal pole (left) and amyg/hippo (right) in response to the second of two successive RewSeq elements after a short, mid or long ISI illustrate the BOLD modulation shown in panel **c**. Error bars denote SE; see also Supplementary Fig. 2

contrasted reward-sequence elements that were preceded by another reward-sequence element (rr pairs) with control-sequence elements that were preceded by another control-sequence element (cc pairs). We hypothesized that rewarded sequence elements may have become more similar in their neural representation than the corresponding elements of the control sequence (sequence fusion; Fig. 2a). In other words, we expected cross-stimulus suppression, and thus a smaller BOLD signal, for repeated occurrence of rewarded sequence elements (rr), compared to control sequence elements (cc). This contrast was therefore intentionally non-directional and the included transitions did not have to obey the correct sequence order (e.g. CB or BD transitions, see Methods and Supplementary Table 2).

Several regions showed differences in the neural representation of rewarded compared to control sequence elements (rr-cc): (a) medial prefrontal cortex (mPFC), spanning parts of areas 32 and 14 m, (b) temporal pole (tempPole), and within the same extended cluster (c) posterior orbitofrontal cortex (pOFC), comprising posterior area 47/12 and aspects of anterior insula (all cluster-level corrected; peak MNI coordinates mPFC: 0, 40, −6, $z = 4.0$; tempPole: 48, 10, −28, $z = 4.19$; pOFC: 30, 18, −22, $z = 3.73$; Fig. 2a, b; for a complete table of activations see Supplementary Table 1). Because of a priori hypotheses about the role of the amygdala, we also examined the same contrast within the amygdala and found a significant activation in right amygdala (peak coordinate: 28,−6,−22; $z = 2.97$; $p < 0.05$ using threshold-free cluster enhancement (TFCE[33]) in a small volume containing left and right amygdala; Fig. 2a). The peak of this activation was located at the border between amygdala and hippocampus and is therefore referred to as amygdala-hippocampus border (amyg/hippo). All four regions showed less activation for the rr compared to the cc condition (for illustration, see Fig. 2b),

suggesting representations of rewarded sequence elements had become more similar to one another than control sequence elements. Importantly, because we had matched, between rewarded and control sequences, how often participants experienced the sequence elements and their transitions, this effect could not be explained by frequency or experience. Thus, in these regions, reward had strengthened associations between sequence elements.

The (rr-cc) contrast probed knowledge of associations between rewarded sequence elements via cross-stimulus suppression but may also have be driven by a main difference in BOLD activity between rewarded and control sequence elements. To confirm our interpretation as cross-stimulus suppression effects, we performed several control analyses. First, we examined the impact of the temporal delay between successive sequence elements. Neural adaptation effects should scale with the temporal delay between stimuli and this was indeed the case in two of our four ROIs, temporal pole and amyg/hippo (Fig. 2c, d, Supplementary Fig. 2a and Supplementary Note 6). Secondly, we confirmed that none of our ROIs were showing a main effect difference between rewarded and control elements (contrast xr-xc: Supplementary Fig. 2b, Supplementary Note 6). Finally, we confirmed that effects of cross-stimulus suppression indeed broadly applied to transitions across the rewarded sequence and were not driven solely by a subset of sequence pairs (Supplementary Note 7 and Supplementary Fig. 2e).

**Neural representations of RL-driven associative structures.** Nevertheless, this type of response pattern – indicating a shared representation of the elements that belong to the rewarded sequence regardless of precise order – would not be sufficient for

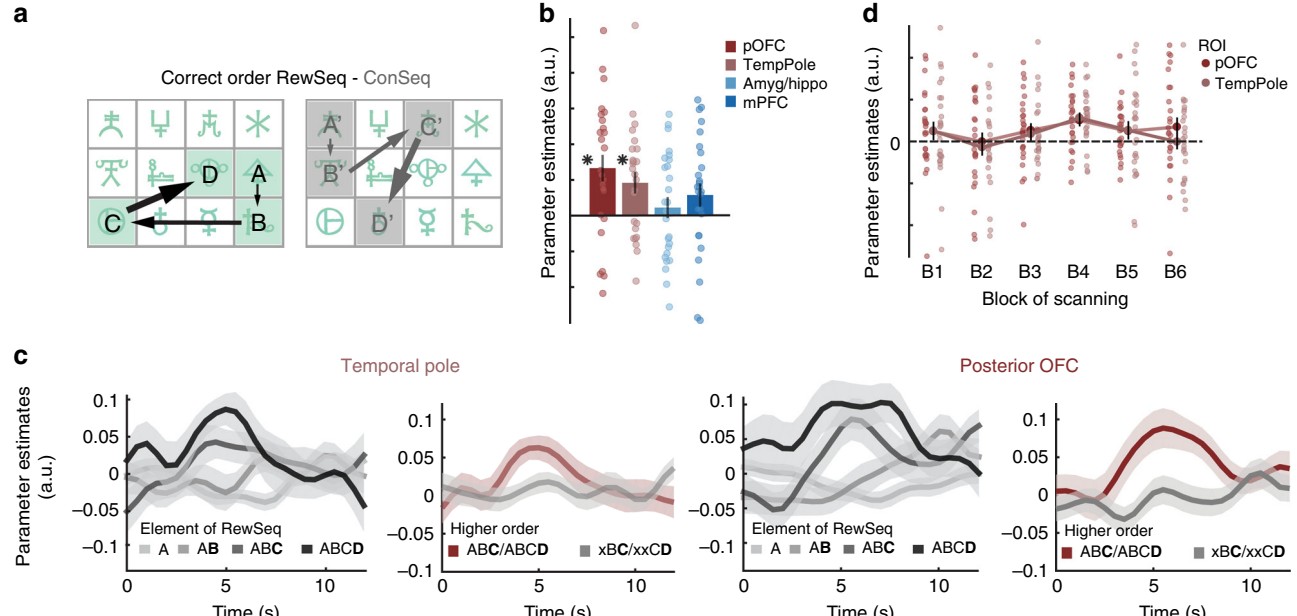

**Fig. 3** Correct sequence order (path to reward) in temporal pole and pOFC. **a** The encoding of correctly ordered sequence associations were examined using occurrences of B after A, C after AB, and D after ABC, and by contrasting RewSeq and ConSeq. In other words, this contrast probed RL-acquired directed higher-order stimulus–stimulus associations. **b** pOFC and tempPole, but not mPFC and amyg/hippo, showed stronger BOLD responses for correct sequence order for RewSeq vs ConSeq and thus tracked the path to reward learnt pre-scanning; *$p < 0.05$ in one-sample $t$-test. **c** Illustration of the effects: (Left) The average time-series to A, (A)B, (AB)C, and (ABC)D illustrate an increase in BOLD activation as participants progress through the correctly ordered associations. (Right) Higher-order associations were necessary for driving the BOLD signal: BOLD average for C and D elements when higher-order conditioned chains were respected (red; average of ABC or ABCD, bold highlights indicate time-locking) compared to when only the pair-structure was fulfilled (average of xBC, xxCD; gray). **d** The BOLD representation of correct sequence order was robust and did not change over scan blocks despite being irrelevant for the task performed during scanning. Error bars denote SE; see also Supplementary Fig. 3

anticipating sequence elements in the correct order, or to anticipate reward. Thus, it does not capture the full extent of RL knowledge acquired prior to scanning. We therefore reasoned that a BOLD signature of the correctly ordered rewarded sequence should also be present. A brain region in which reward is back-propagated to previous stimuli, for example, via TD learning, should demonstrate a particular sensitivity for the correctly ordered stimulus sequence that has led up to reward in the past. Importantly, we expected such a signature to get stronger as reward is approached, thus going in the opposite direction to the suppression effects reported for undirected pairs. Within the same GLM as used above (GLM1, Supplementary Fig. 2d), we tested whether any of our four ROIs reflected knowledge of the correct sequence order (Fig. 3a). We compared activity between RewSeq and ConSeq when the correct stimulus transitions were experienced (i.e. B after A, C after AB, and D after ABC). Two regions, the temporal pole and adjacent pOFC, exhibited stronger BOLD for correctly ordered transitions of RewSeq compared to ConSeq ($p$(pOFC) = 0.0014, $t(25)$ = 3.59; $p$(tempPole) = 0.0051, $t(25)$ = 3.07; one-sample $t$-tests; Fig. 3b; Supplementary Table 2). Examination at the whole-brain revealed bilateral effects in both regions (Supplementary Fig. 3). Further inspection of pOFC/tempPole signals showed a BOLD build-up as participants advanced through the rewarded sequence (GLM main effect of progress: $p$ = 0.012, $t(25)$ = 2.70, ROI and ROI × progress: $p$ > 0.1; one-sample $t$-test; illustrated in Fig. 3c), and it showed that the entire third- and fourth-order structure (e.g., ABC not just BC) had to be fulfilled for this build-up to be present ($2 \times 2$ ANOVA ROI × HigherOrder: effect of HigherOrder $F(1,25)$ = 5.712; $p$ = 0.025; Supplementary Note 8 and Fig. 3c). Such a signal could reflect temporally discounted reward expectancy or reward proximity. However, since participants' behavior showed that the reinforced associative structure is formed backwards, this BOLD

signature likely formed via backpropagation of reinforcement (Fig. 1d, Supplementary Fig. 1e).

Notably, unlike in the pre-scan learning task, knowledge of the rewarded sequence was irrelevant for participants during the scan task. Performance in the scan task simply depended on responding to catch trials. We therefore tested whether the correct sequence order effects reported above decreased as the scanning session progressed. Contrast estimates for correct sequence order (RewSeq vs ConSeq) were extracted from pOFC and temporal pole, separately for each block. Because frequentist statistics cannot provide evidence in favor of the null hypothesis, we conducted a $2 \times 6$ Bayesian repeated-measures ANOVA with factors ROI (pOFC and tempPole) and block. This revealed that the Null Model was the best model and almost six times more likely than any other model (BFm = 5.81, $P$(M|data) = 0.59), including a model with an effect of Block (BFm = 1.43, $P$(M|data) = 0.26). Thus, representations of the correctly ordered reinforced sequence in temporal pole and pOFC were remarkably inflexible throughout the experiment despite changing task demands. They appear almost to have been frozen in a stable state, possibly because no new reward-learning interfered with the originally formed representations.

**BOLD representations of statistical task relationships.** Knowledge of the correctly ordered sequence is important for guiding an agent towards reward and represents the RL-acquired associative knowledge in this task. Nevertheless, it is a sparse and possibly quite inflexible code. For example, if correct order representations in tempPole and pOFC are learned via TD, then they rely upon fixed stimulus-to-stimulus mappings, and hence, swapping the order of two elements in the sequence would mean that an entirely new code has to be formed and the previous one over-

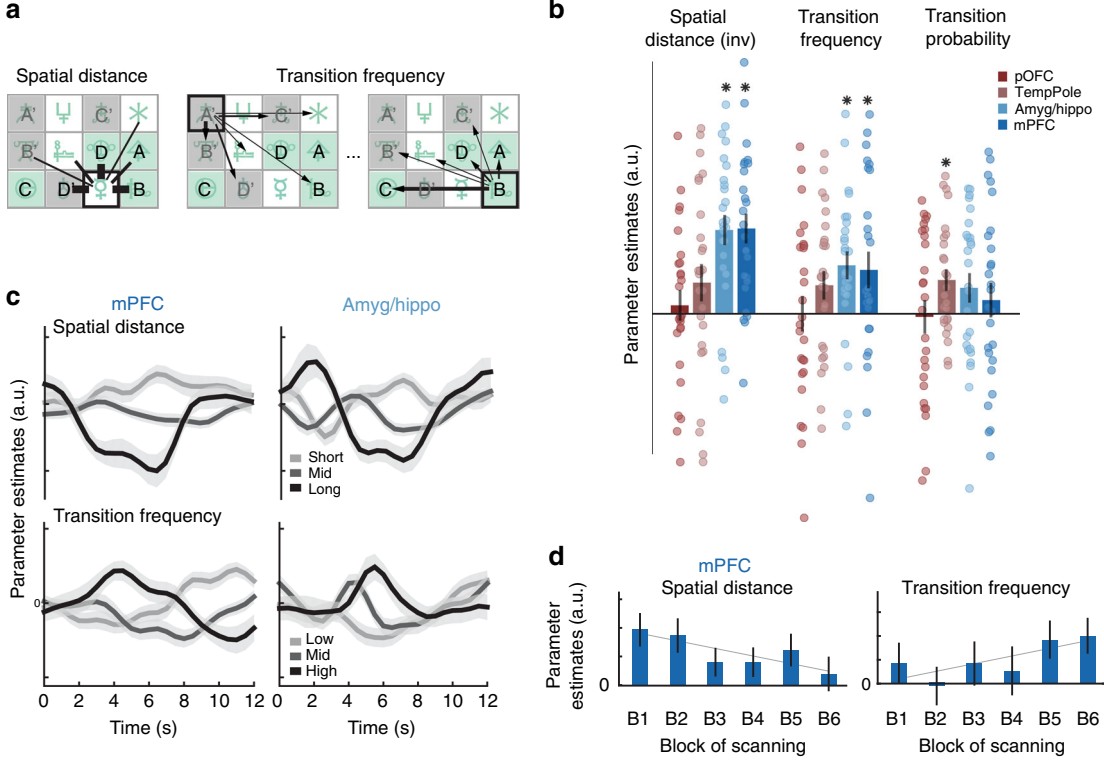

**Fig. 4** Statistical knowledge of the task space in mPFC and amyg/hippo. **a** Two measures of statistical (incidental) knowledge were examined. In other words, stimulus–stimulus relationships that are true across all stimuli and independent of reward: spatial link distance, i.e., the number of steps from one element to another; and transition frequency, i.e., how often a transition from one element to another was experienced. Transition frequency was updated on a trial-by-trial basis (for model details, see Methods). **b** Both spatial link distance (sign-flipped for illustration) and transition frequency were reflected in BOLD activity in mPFC and amyg/hippo but not temporal pole or pOFC. Intriguingly, transition frequency (absolute measure of link between stimulus i and j) was a better predictor of activity in mPFC and amyg/hippo than transition probability (right: normalized measure of likelihood of going from i to j, given i), suggesting an absolute coding of the global task space rather than conditional knowledge of transitions normalized by the frequency of experiencing the initial states; *$p < 0.05$ in one-sample $t$-test. **c** Both effects are illustrated in both regions by separating trials into those with short/mid/long spatial distances (top), or low/mid/high transition frequency (bottom) between successive elements. **d** Spatial distance modulations decreased, and transition frequency modulations increased over time in mPFC. Error bars denote SE; see also Supplementary Fig. 4

written. By contrast, a neural code with knowledge of spatial or statistical contingencies (e.g., transitions that are generally more or less likely to occur in the task) would allow more flexibility in learning new relationships[21]. We tested this idea by examining neural representations related to the spatial proximity of subsequent states (a simple prior on likely state transitions) and to the statistical frequency of these transitions themselves. Importantly, such representations would rely on incidental learning mechanisms and relate to all stimuli, whether reinforced or not. They cannot be driven by reward associations. We expected suppression of BOLD, and thus weaker signals for nearby or likely transitions. However, note that the effects described below went in the opposite direction (see also Supplementary Table 2).

We first asked whether information about spatial distances between all elements, measured in terms of steps that make up a path on the grid, was present in our previously defined ROIs (GLM2, Supplementary Fig. 2d; Fig. 4a). This spatial distance contrast was not correlated with the ROI-defining contrast rr-cc (Pearson's $r = -0.02$; see also Supplementary Fig. 2f). BOLD signals in both mPFC and amyg/hippo but not temporal pole or pOFC scaled with distance, with stronger (rather than weaker) responses for stimuli that were closer in space to the previous stimulus ($p$(amyg/hippo) = 7.91e-6, $t(25) = -5.6$; $p$(mPFC) = 7.54e-6, $t(25) = -5.62$; $p$(pOFC) and $p$(tempPole) > 0.1; all one-sample $t$-tests; mPFC global peak: 0, 50, −14, $z = 5.0$; peak within mPFC spherical ROI: 0, 36, −6, $z = 4.01$; amyg/hippo global peak: 30, −4, −24, $z = 5.26$; Fig. 4b, c). At the whole brain, spatial

relationships were reflected in the activity of an extended network of brain regions which included mPFC and the amygdala-hippocampus border (all cluster-corrected; Supplementary Fig. 4a). Note that during the pre-scan learning task, larger spatial distances were associated with longer travel times (correlation trajectory time and spatial distance: Pearson's $r = 0.5$ and $r = 0.66$ on days 1 and 2). Thus, the experience of both time and distance could have contributed to the formation of this neural representation.

In addition to spatial proximity, a more important piece of information for anticipating the next element during the scan task, is the likelihood of transitioning between any two stimuli on the grid (Fig. 4a, Supplementary Fig. 4c). We tested, using a model with trial-by-trial updates (see Methods), whether BOLD responses scaled with the frequency of an experienced transition from the previous to the current stimulus in any of our ROIs. Again, there was no correlation between this contrast and the ROI-defining contrast ($r = -0.01$; Supplementary Figs. 2f and 4c). Indeed, mPFC and the amygdala-hippocampus border, but not temporal pole or pOFC, had knowledge of these higher-order statistical relationships ($p$(amyg/hippo) = 0.002, $t(25) = 3.46$; $p$(mPFC) = 0.024, $t(25) = 2.39$; $p$(tempPole), and $p$(pOFC) > 0.05; one-sample $t$-tests; Fig. 4b, c; Supplementary Fig. 4a); they responded more strongly to transitions with a high frequency of occurrence.

We confirmed in a control analysis that spatial distance and transition frequency responses were not driven by the subset of

stimuli of the rewarded sequence. There was no difference between rewarded and control stimuli in either mPFC or amyg/hippo in terms of BOLD modulations by spatial distance and transition frequency (all $p > 0.15$, paired $t$-tests; Supplementary Note 9 and Supplementary Fig. 4d).

To further understand the nature of the statistical associations (task model), we probed whether transition frequency was represented in terms of conditional transition probabilities between stimuli given the initial stimulus or as absolute state-transition frequencies. The latter is arguably a more global, flexible, and abstract representation of the task space and less dependent on the precise nature of the experiences during the time when it was acquired. BOLD activity in both mPFC and amyg/hippo was explained by a representation of state-transition frequencies but not by the conditional probability of a transition (Fig. 4b, Supplementary Note 10).

To examine the flexibility of statistical knowledge coding over time, we speculated spatial distance might have been a prior imposed during training, when participants were constrained by 1-step movements to walk around the $3 \times 4$ grid. If so, spatial distance coding should become less pronounced as participants experience that the scan task does not have spatial constraints. By contrast, transition frequency coding might increase the longer participants perform the scan task because it is a better predictor of the next element than spatial distance (21% versus 9.5% accuracy in predicting successor; chance is 8%). We inspected parameter estimates for spatial distance and transition frequency in mPFC across blocks of scanning and this was precisely what we found (Fig. 4d). A linear regression revealed a significant interaction of time and contrast (one-sample $t$-test: $p = 0.0247$, $t(25) = -2.34$) with stronger transition frequency coding in later blocks and spatial distance coding gradually diminishing over time. Such a pattern was not observed in any of the other ROIs (one-sample $t$-tests: all $p > 0.3$), not observed when replacing transition frequency by transition probability (one-sample $t$-tests: all $p > 0.05$), but it was present for transition frequency in mPFC whether or not transition probability was included in the model (one-sample $t$-test: $p = 0.039$, $t(25) = -2.19$ for a model also including conditional probability).

To complete the picture, we tested whether statistically learnt knowledge in mPFC and amyg/hippo was, over blocks of scanning, replacing the undirected knowledge of the rewarded sequence, as probed by our initial ROI-defining contrast (sequence fusion: rr-cc). There was no evidence for a decline in the sequence fusion contrast over blocks of scanning in either region (one-sample $t$-tests: mPFC: $p = 0.56$; amyg/hippo: $p = 0.38$). Thus, even though mPFC and amyg/hippo hold flexible statistical knowledge, the reward-driven associations they also hold are persistent and inflexible.

Taken together, knowledge from two separate learning mechanisms converged in amyg/hippo and mPFC. Both regions carried undirected knowledge of the rewarded sequence, as probed by the ROI-defining contrast, but they also represented reward-independent associations which were true across all stimuli. They responded more strongly to likely transitions both in space (closer) as well as statistical contingencies (more frequent) suggesting full knowledge of task relationships and an encoding of associations acquired through incidental learning mechanisms. In contrast to the representations found in temporal pole and pOFC, statistical representations are not linked to the directed reward associations learned prior to scanning. Moreover, over time, mPFC transitioned from a spatial towards a frequency coding scheme.

**Dissociating tempPole-pOFC versus hippo/amyg-mPFC networks.** Above we showed that BOLD responses in temporal pole and pOFC reflected the correctly ordered rewarded sequence, while mPFC and amyg/hippo carried knowledge of statistical relationships between all stimuli. Formal statistical testing confirmed a dissociation between the patterns of BOLD activation held by these two different networks (Supplementary Note 11), consistent with work showing major anatomical differences between pOFC and tempPole on the one hand and hippo/amyg and mPFC on the other hand. For example, strong monosynaptic connections within but not across these two networks[34–36], and stronger resting-state coupling within compared to between networks (Supplementary Fig. 5).

**Establishing eligibility through spread of reward in space.** Tracking state transitions as well as representing learned sequence-reward associations is essential for forming an appropriate representation of the external world. However, sometimes rewards can influence the response to stimuli that are not actually predictive of reward but which simply occur close in time to the reward and this temporal spreading of reward has been linked to activity in the amygdala[24,37]. Here we tested for a spatial spreading of reward which was possible because stimuli had spatial relationships on the $3 \times 4$ layout. Specifically we asked whether the amygdala showed changes in activity to stimuli that were close in space to the location where reward had occurred during the pre-scan learning task. We defined any element that could be reached by one step (either horizontally or vertically) as neighbors and compared activity for neighboring elements of D with the neighboring elements of D′ (the final elements of the rewarded and control sequences; Fig. 5a, b). This contrast was examined in an amygdala ROI previously associated with temporal spreading of reward (see Methods, ref. [38]). Consistent with our hypothesis, we found evidence for a stronger response to neighbors of D compared to neighbors of D′ in the amygdala (one-sample $t$-test: $p = 0.03$, $t(25) = 2.30$; peak in left amygdala: $-18, -8, -16, z = 3.0$; note that the amygdala was the strongest peak at the whole-brain). In participants behavior, on the contrary, we did not observe an effect of spatial spreading of reward (Supplementary Note 12). Note that the number of neighbors was exactly matched for the reinforced and non-reinforced sequences so that difference between rewarded and non-rewarded sequences were not confounded by differences in spatial uncertainty.

An important next question was whether the spread of reward effect in the amygdala might relate to the strength of the contingent BOLD representation of the rewarded sequence. Formation of sequence knowledge could compete with reward-spreading, such that people who strongly respond to relationships between rewarded sequence elements (more suppression for rr versus cc) have a weaker spatial spread effect (positive correlation because of sign of cross-stimulus suppression). Alternatively, if these mechanisms work hand in hand, the two signatures might correlate negatively. Indeed, we found such a negative correlation within the amygdala region defined above and this effect was only significant in the first block of scanning, i.e. when learning effects were the most pronounced ($p$ values corrected for multiple comparisons: Pearson's correlation: $r = -0.626$, $p = 0.0036$ for block 1, all other blocks: $p > 0.15$; Fig. 5c). In summary, a spread of reward effect apparent in the amygdala is related to the representation of RL knowledge in the same structure and may exist to guide the construction of a more refined representation of sequence associations by motivating behavior in the vicinity of a potential reward.

**PE-related activity in VS reflects RL knowledge.** The ventral striatum (VS) plays an important role in learning from reinforcement, in particular through its encoding of PEs at the time of outcome. We therefore tested whether VS was involved in representing associative knowledge in our task. First, we asked

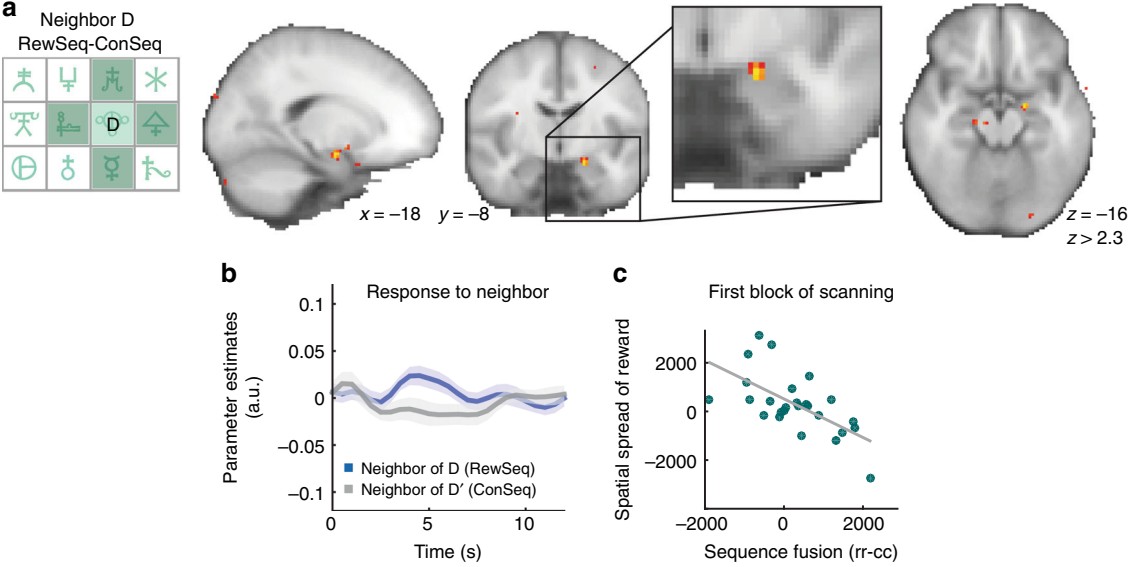

**Fig. 5** Spatial spread of reward effects in amygdala. **a** Probing differences in the BOLD signal for neighbors of the rewarded goal element D and neighbors of the control sequence element D′, i.e. the final elements of the equally frequent RewSeq and ConSeq, showed increased activation for the neighbors of D in centromedial amygdala (3-mm spherical ROI centered on previous coordinate from Jocham et al.[38]; amygdala is the largest peak at the whole-brain). This is consistent with a mechanism that spreads reward to nearby locations (spatial spread), and thus similar to a previously reported amygdala signature related to a temporal spread of reward. **b** Average time-series to neighbors of D and D′ illustrate this effect. **c** People with more spreading of reward in space (effect in **a** and **b**) show a more pronounced representation of sequence knowledge in central amygdala (more negative means more pronounced cross-stimulus suppression; see Fig. 2) but this relationship is only present in the first block of scanning, i.e., when the information from the learning task is likely to be represented most strongly and before transitions and reward locations are scrambled in the scan task. Error bars denote SE

whether it exhibited activity reflecting knowledge constructed via reinforcement (RewSeq versus ConSeq; correct sequence order), knowledge of the task state-transition structure (spatial distance; transition frequency), or information about reward proximity (spatial spread). In an anatomically defined ROI (see Methods), none of these effects were present (one-sample $t$-tests: all $p > 0.4$; Fig. 6a). This absence of associative knowledge cannot be explained by a lower signal in VS: the mean temporal signal-to-noise ratio (tSNR) in VS was better than in amyg/hippo and pOFC ROIs, and no different from vmPFC or temporal pole ROIs, nor from the mean tSNR across the brain.

However, even if there is no evidence that VS activity reflects stored associative knowledge, it is possible that its activity reflects the PEs that might be experienced given that such knowledge is present elsewhere in the brain. Notably, RL theory predicts that PEs move, during learning, to the earliest reward-predicting stimulus, here the starting element A of the rewarded sequence. We therefore tested, whether VS would code A more strongly for the rewarded compared to the control sequence. Indeed, we found responses to A over and above A′ in lateral VS/putamen[39] (cluster-corrected $p < 0.05$ in the right hemisphere: peak $-24$, $14$, $-8$, $z = 3.30$). But even in an ROI centered on this coordinate, none of the other tests described above were significant (one-sample $t$-tests: all $p > 0.1$; Fig. 6b–d). Finally, consistent with RL predictions, VS coded for the picture of the flower (the reward during scanning; one-sample $t$-test: $p(\text{anatVS}) = 4.47\text{e-}09$, $t(25) = 8.75$; $p(\text{funcVS}) = 1.32\text{e-}09$, $t(25) = 12.98$) and the picture of the treasure (the reward during learning; one-sample $t$-test: $p(\text{funcVS}) = 0.009$, $t(25) = 2.82$ only in the functionally defined ROI). These appeared at somewhat (treasure) or entirely (flower) unpredictable times during scanning (Fig. 6a, b).

## Discussion

We investigated the associative structures formed through reinforcement and incidental learning mechanisms and describe two

neural circuits with distinct coding schemes (Fig. 7). Temporal pole and posterior OFC represented elements of a reinforced sequence and increased with progress through correctly ordered associations, thus encoding RL-driven associative structures. By contrast, activity in mPFC and a region at the intersection between amygdala and hippocampus reflected map-like knowledge of the spatial and statistical associations between all stimuli, suggesting knowledge of associative structures acquired via statistical learning. Strikingly, the ventral striatum, the key region showing PEs at outcome time, did not represent complex associative structures formed via RL or statistical learning, but consistent with TD, it coded the reward PE associated with the first stimulus that was predictive of reward, i.e., the start of the rewarded sequence.

Our first key finding demonstrated BOLD representations of the reinforced sequence in temporal pole and pOFC, with an increase in activity as participants progressed through the correctly ordered sequence (Fig. 3 and Supplementary Fig. 3; Fig. 7). While such a BOLD increase would be consistent with temporally discounted reward expectancy or reward proximity, participants' behavior demonstrates that the reinforced sequence was acquired backwards. Thus, while we did not investigate BOLD activity during learning, this suggests that these neural signals might have formed through backwards propagation of reward as posited in RL models (Fig. 1 and Supplementary Fig. 1). Participants were able to remember the sequence at the end of testing (100% accuracy) despite scrambled transitions being shown during the scan sessions. In line with this, BOLD signatures of the correctly ordered sequence in temporal pole and pOFC did not change over time, suggesting this code was inflexible and robust to interference. However, neither temporal pole nor pOFC responded to the picture of the treasure, i.e. the reward itself, suggesting a coding of knowledge formed as a result of RL rather than a role in forming this knowledge.

A number of distinct cytoarchitectonic areas are situated in pOFC where it transitions into anterior insula both in humans and

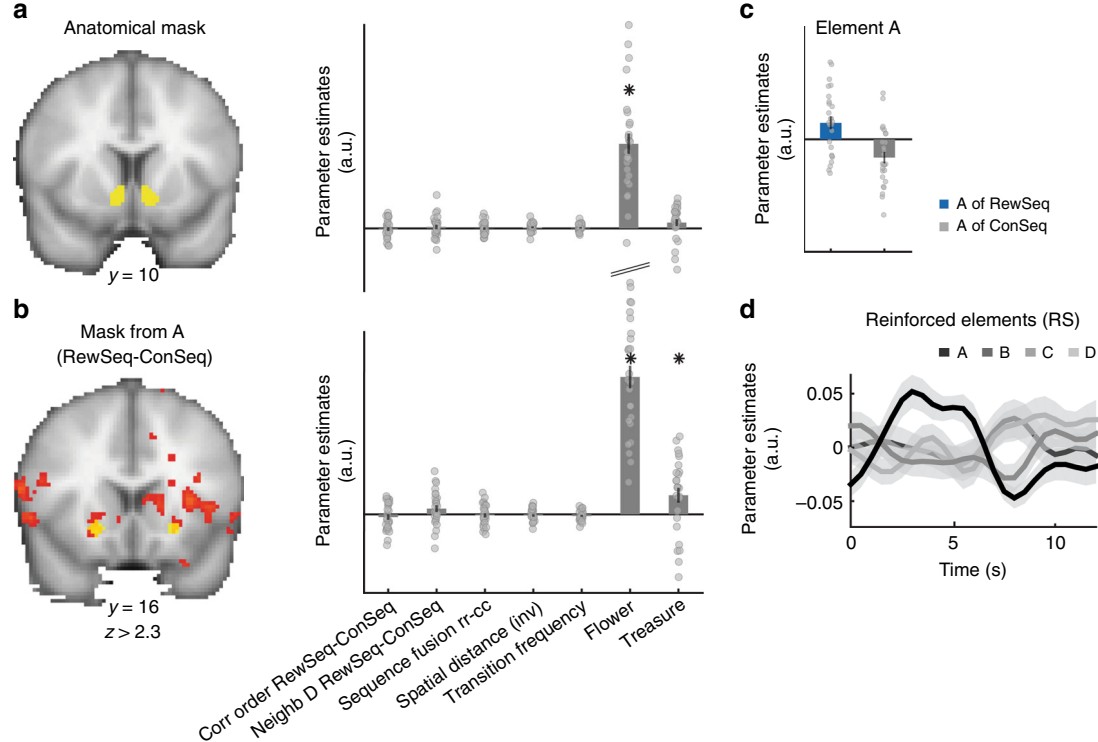

**Fig. 6** Ventral striatum does not code RL- or map-like knowledge. **a** Anatomical mask of ventral striatum (VS) and extracted parameter estimates for all effects of interest in this study: only the BOLD response to a stimulus containing a flower is significant in this region. Responding fast enough (<800 ms) to a flower leads to monetary reward and thus the flower is linked to unexpected reward during scanning. **b** Mask of VS extracted from a contrast comparing the first sequence element A between RewSeq and ConSeq. This VS region responds to both the flower and treasure and thus to unexpected occurrences of currently and previously reinforced stimuli. **c, d** Illustration of the BOLD response to element A of the sequence in the mask shown in **b**; error bars denote SE; * in **a**, **b** indicates $p < 0.05$ in one-sample $t$-test

macaques[40,41] and they likely have distinct functions. Aspects of the association between a visual stimulus and outcome are coded in an adjacent more medial region of pOFC ([42]; MNI coordinates: −18, 5, −23 in the previous study versus 30, 18, −22 here). The medial pOFC may be particularly concerned with expectations about particular outcome types[42] while the more lateral pOFC found here may be concerned with the associative links between multiple predictive stimuli. Other more lateral and anterior OFC regions have been linked with credit assignment during stimulus-outcome learning, particularly when multiple cues are potential eligible causes of outcomes[22,23,42–45].

Prior work has highlighted ventral anterior temporal cortex in encoding visual stimulus–stimulus associations[46–48], and in associating visual cues with reward[49]. While the focus has been on the perirhinal cortex less is known about the more polar and lateral regions of temporal cortex that were activated in our study although the interactions between the perirhinal cortex, temporal pole, and lateral inferotemporal cortex have been emphasized[50,51]. The coding of sequence knowledge in temporal pole and interconnected pOFC may rely on multiple stimulus–stimulus associations which would be consistent with the visual inputs of these regions[52–54]. Notably, sequences that rely on associations between multiple actions rather than stimuli (typically taught in a quite distinct manner), depend on different circuits, e.g.,[25–27]. Similarly, sequences for which progress is internally tracked rather than externally cued rely on structures such as anterior cingulate cortex[49,55], an area more suited to forming and updating progress in an internal task model[56,57].

Our second main finding related to BOLD signatures of statistical knowledge in mPFC and an area at the interface of amygdala and hippocampus (Fig. 4 and Supplementary Fig. 4;

Fig. 7; activity patterns from an adjacent hippocampal region studied by[58] are reported in Supplementary Fig. 6; BOLD activity in hippocampus reflected statistical knowledge but did not distinguish between rewarded and control sequences). Statistical knowledge provides increased flexibility to adapt to changing task goals. For instance, it can expedite the learning of new or altered sequences without requiring unlearning of rigid stimulus–stimulus associations such as those seen in pOFC and temporal pole.

We identified two interesting signals in mPFC and amyg/hippo. The first related to spatial relationships, with stronger BOLD responses for transitions to spatially closer stimuli. The spatial layout of stimuli was a major determinant of the task space in pre-scan sessions when participants only moved between adjacent elements. However, other transitions were experienced during the scanning period. Notably the mPFC signal tracking spatial relationships diminished over time in the scan task. The second signal reflected higher-order associations between map elements, namely the frequency of transitioning between any two stimuli. The more frequent a transition, the more strongly mPFC and amyg/hippo responded to the successor. Intriguingly, this code was not conditional on the previous state but instead reflected global transition frequencies, i.e., the likelihood of an experienced transition given all experienced transitions on the map. This signal got stronger as scanning progressed.

The task state-transition knowledge identified in mPFC and amyg/hippo can be considered a form of cognitive map[21]. Relatedly[59], Constantinescu, O'Reilly and colleagues recently showed that conceptual knowledge can be encoded in a map-like manner using a grid code in the same mPFC region and an adjacent mPFC area has been linked to the representation of the current state in a map of task space[60].

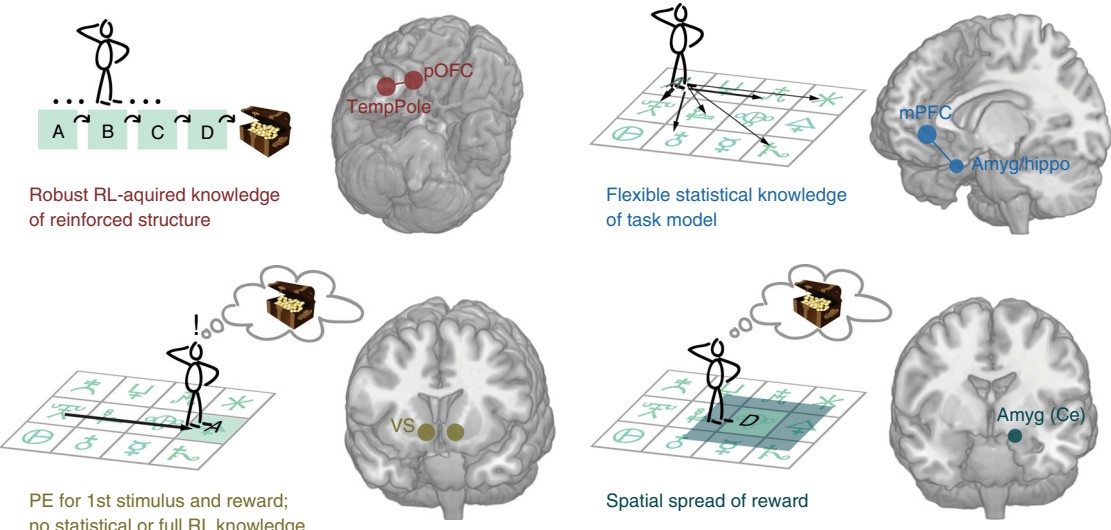

**Fig. 7** Schematic summary. Schematic overview of the key findings of this study. Note that this schematic focusses on the differences in the neural representations we identified. All four regions spanning the medial and lateral networks shown in the top row also showed BOLD signatures that suggested knowledge of undirected relationships within the rewarded compared to the control sequence (sequence fusion, Fig. 2a)

It is important to note that reward-learning and processes akin to those that we refer to as statistical learning can proceed simultaneously[21] and that the associative structures that form as a result of RL and incidental learning of task statistics do not simply map onto stimulus–reward and stimulus–stimulus associations respectively. RL can mediate relationships between reward-predictive stimuli, and thus stimulus–stimulus associations (A–B, B–C), as well as stimulus–reward associations (A-reward, B-reward); indeed stimulus–stimulus associations are a key focus of classic RL treatments[28] of animal studies using higher order conditioning, and they are particularly important when the order of transitioning through the stimuli matters for reaching reward. Hence, stimulus–stimulus associations are not purely derived by incidental learning. Indeed we show that, stimulus–stimulus associations are learned faster than one would expect from purely statistical learning; propagating the reward backwards across multiple stimuli speeds up learning.

Intriguingly, activity in mPFC and amyg/hippo simultaneously reflected several dissociable aspects of the task. On the one hand its activity reflected statistical learning – knowledge of transition frequencies between task states regardless of reinforcement. On the other hand, its activity reflected associations between rewarded sequence elements over and above associations between control sequence elements (regardless of order); this latter representation depended on reinforcement. While statistical knowledge was flexibly updated and constantly changing in mPFC, reinforcement-based task representations did not change over time, even though this knowledge was irrelevant to the task performed during scanning. In summary, RL and statistical learning mechanisms converge in mPFC but have different temporal properties. It has previously been pointed out that learning related representations acquired over different time scales exist simultaneously in different brain regions[61] and even simultaneously within the same brain region[62–64]. In the current study, however, the representations are not simply derived from the same learning process operating at different timescales but as a result of different learning processes operating at different timescales.

Our third key result demonstrated a spatial spread of reward signal in centromedial amygdala (Fig. 5 and Supplementary

Fig. 5; Fig. 7), a region with strong dopaminergic innervation[65] and direct connections to autonomic brainstem structures[66,67]. This signature is interesting in light of other work that has suggested amygdala activity reflects a temporal spread of reward effect[22,23]. The amygdala was shown to leak reward signals into subsequent trials even if this reduced optimal choice behavior[22] and amygdala signals became particularly pronounced when spreading reward was the only viable learning mechanism[23]. In both studies, suppression of non-contingent signals in the amygdala facilitated contingent learning mediated by OFC. This could explain work showing that OFC-lesion induced impairments in value-guided choices are reversed by subsequent amygdala lesions[68], and that neurotoxic amygdala lesions improve reversal learning[69]. Here we show that the amygdala's role in learning about states close in time to reward extends to states close in space to reward.

A temporal spreading of reward may be useful when few choices are available, and the same choice is repeated multiple times, or when the actions during which reward is acquired are in themselves extended in time. Correspondingly, spatial spreading of reward may provide a behavioral advantage when similar outcomes cluster around similar locations, which is true in many ecological scenarios. It is also possible that spreading reward to nearby stimuli can help determine which stimuli are eligible to be assigned reward in initial stages of learning (eligibility trace[28]). We examined if the representation of the reinforced sequence (the contingent knowledge in our task) related to the spread of reward signal in the amygdala. We found that more pronounced spread signals related to more pronounced sequence representations. This suggests that contingent and non-contingent learning mechanisms may have similar sources or that one drives the other.

Our final striking observation was that the ventral striatum BOLD response did not distinguish between rewarded and control sequence elements, relate to correctly ordered reinforced knowledge, task statistics, or a spread-of-reward signal (Fig. 6 and Supplementary Fig. 6; Fig. 7). Instead, ventral striatum showed PE-like responses at the occurrence of the first element of the reinforced sequence and thus precisely the signal that would be predicted by TD theory.

## Methods

**Participants.** Twenty-seven participants (11 female, age 19–35, mean 26 ± 0.77) took part in this study. Screening criteria included no history of neurological or psychiatric disorder, normal or corrected-to-normal vision and suitability for undergoing MRI scanning. One participant who failed to complete the experiment because he fell asleep repeatedly in the scanner was excluded from all analyses. The final sample thus included 26 participants. The study was approved by the University of Oxford Central Research Ethics Committee (MSD-IDREC-C1-2013-095) and all participants gave written informed consent.

**Experimental procedure.** Participants agreed to take part in four sessions on separate days within 1 week (Supplementary Fig. 1a). The first two sessions were behavioral and lasted 1–1.5 h and the third and fourth sessions included behavioral and MRI testing and lasted around 2.5 h (of which 1.5 h was in the MRI scanner). Participants were reimbursed £60 for their time (£10 for behavioral and £20 for scan sessions) and could earn up to a maximum of £40 (£10 per session) for their performance on the task. The average total payment was £96.7 ± 0.37.

**Task design and procedure of pre-scan learning sessions.** An array of 3 × 4 abstract stimuli were presented on the screen in a fixed configuration (Fig. 1a, left; Supplementary Fig. 1b, c). Unbeknownst to the participant, four of these twelve stimuli made up a sequence (referred to as ABCD) which led to reward (rewarded sequence RewSeq). The aim of the behavioral training was to learn what caused the reward, i.e. to learn the association between the sequence of four stimuli and reward. More precisely, participants were instructed to find out how to unlock the treasure and while we did not explicitly use the word sequence at any point, we gave some examples in the instructions that would likely have made them realize that learning was about more than one stimulus. The elements of RewSeq did not have to be adjacent in space but they had to be experienced in the correct order for reward to occur. Importantly, in contrast to prior work related to the acquisition of sequence knowledge (mostly in the motor domain, e.g., refs. [25–27]), participants had to infer the length and start of the sequence neither of which we instructed.

The order of events was as follows (Fig. 1a, Supplementary Fig. 1c). The computer highlighted one stimulus at a time with a yellow border and participants were asked to move an agent (purple dot) to the instructed location using right/left/up/down arrow keys (right hand index/middle/ring fingers). As soon as they reached the highlighted stimulus, passing by several sequence irrelevant stimuli in the process, the next stimulus was highlighted, and so forth. The highlighting of stimuli happened in a continuous stream and in pseudo-random order. The 3 × 4 layout of stimuli only disappeared in two situations: (a) when the reward (picture of treasure) appeared (six times per block), which was as soon as the participant reached stimulus D after A, B, and C had been highlighted; (b) when a distractor stimulus (picture of a bridge) appeared on the screen at unpredictable times (six times per block). Participants were instructed to press the space bar as fast as possible upon presentation of the reward or bridge (left hand index finger). The frequency-matched bridge stimulus served as a reaction time (RT) control for the reward, thus enabling us to measure reward anticipation. Participants were told that only the highlighted stimuli mattered for learning but not the route they walked from one highlighted stimulus to the next. Task timings were determined by the participant's walking speed and chosen paths and by their RTs to bridge and reward.

Learning was repeated with different sequences in three different contexts which were signaled with layouts of differently colored and shaped stimuli (green, purple, and gray; Fig. 1a; Supplementary Fig. 1b). Thus, a total of three rewarded sequences (RewSeq) made up of different stimuli in different locations had to be learnt. In addition to the RewSeq, the remaining eight stimuli in each context made up a control sequence ConSeq (comprising elements A'B'C'D') which was highlighted in the correct order as frequently as RewSeq but which did not lead to reward and was thus not subject to reinforcement; and a second control sequence ConSeq2 (A″,B″,C″,D″) which was less frequent and also unrewarded. Originally, we had planned to compare ConSeq2 to ConSeq to probe statistical learning, or in other words ConSeq2 served as a frequency control for ConSeq. Because of the rare occurrences of ConSeq2, however, this analysis was lacking in power and model-based analyses that included all stimuli seemed more appropriate to investigate statistical learning (see below).

Participants were randomly assigned to one of two groups at the start of the experiment. The stimuli assigned to the RewSeq and ConSeq sequences were counterbalanced between the groups to rule out the possibility that any differences in learning depended on specific features of the stimuli or details of the sequence layout. In other words, what was RewSeq in a given context in group 1 became ConSeq in group 2 and vice versa. In both training sessions, participants completed four blocks (96 stimuli, plus 6 rewards) in each context, and thus a total of twelve training blocks on day 1 and twelve training blocks on day 2. Blocks were sorted by context on day 1 and completed in random order on day 2 but with context still clearly signaled using the differently colored stimuli.

Immediately following each learning block participants completed a block of binary choices (n = 54) between combinations of two stimuli highlighted using green and blue squares in the usual configuration (i.e., as part of the 3 × 4 map layout) to probe any preference for the rewarded stimuli (choice task). Subjects were instructed to indicate which of the highlighted stimuli they associated with

reward (Supplementary Fig. 1c). After this they had three opportunities to enter the correct sequence by walking around in the 3 × 4 maze and pressing space bar on the stimuli they wanted to be included, following the correct order (sequence completion task; Supplementary Fig. 1c). Their performance on these two tasks determined their additional payment from the training sessions (1 point (~0.61 pence) for any choice of an RewSeq stimulus i.e. A, B, C, or D; 12 points (~7.5 pence) for entering a correct sequence; % of points out of maximum possible points were translated into % of £10 received (max £7.3 for choice, max £2.7 for sequence completion): mean £9.07 ± 0.19 on day 1; £9.90 ± 0.02 on day 2). Importantly, feedback on performance was not given during the task but only at the end of each day's session.

**Task design and procedure of scanning sessions.** During scanning, the screen displayed one of the familiar 3 × 4 layouts of abstract stimuli in each block. Again, stimuli were highlighted one at a time and in pseudo-random order (ISIs drawn from truncated Poisson-distribution: mean 2.8 s, range 1.5–10 s). However, there was no agent and participants did not have to press several buttons to walk around. Instead, they performed an incidental task to ensure their attention was focused on the highlighted stimulus. Participants were asked to search for a small flower on the currently highlighted stimulus (Fig. 1a, right; Supplementary Fig. 1c). If they detected a flower and pressed a single button (right index finger on a button box) within 800 ms of the stimulus being highlighted, they earned a bonus reward (total of 42 flowers per block corresponding to £10; mean £8.50 ± 0.28 on scan day 1, on average caught 35.79 ± 1.18 of 42 flowers; mean £9.20 ± 0.14 on scan day 2; on average caught 38.10 ± 0.81 of 42 flowers). Occasionally, the picture of treasure was shown in-between the highlighted stimuli on the grid (n = 24 per block out of a total of 420 stimuli), just like during the pre-scan learning task. It was shown in six possible places: after AB, xxCD, ABCD, A'B', xxC'D', A'B'C'D'. The 3 × 4 map display disappeared for the duration of the picture of treasure (shown centrally) and participants had to search for a flower on the treasure stimulus just as they did for any other stimulus. Hence, what used to be the behaviorally relevant rewarding stimulus had no particular meaning during the scanning task and no response was required unless it contained a flower.

Before entering the scanner, participants performed one block of the pre-scan learning task per context, identical to those performed during the preceding behavioral sessions to refresh their memory of the learnt sequences (Supplementary Fig. 1a). In the scanner, participants performed three blocks of ~20 min of the scanning task described above, one in each context and with contexts drawn in random order. Following scanning, they completed the choice and sequence completion tasks from the training once per context to probe their sequence knowledge. Finally, at the very end of the second day of scanning, participants performed a short memory task (Supplementary Fig. 1c, Supplementary Note 5). Each stimulus (n = 3 × 12 = 36) was shown on its own in the 3 × 4 layout with the other eleven stimuli grayed out: once in its correct position and place, once in a different color (a color from one of the other two contexts) and once in a different position but its original color. For each of these 3 × 36 = 108 stimuli, participants were asked to judge whether the stimulus was correct (as experienced during training) or wrong (right index and middle finger, respectively).

**Stimulus generation.** The transition frequencies experienced during training and scanning are shown in Supplementary Fig. 4c. All stimuli and all transitions between RewSeq and ConSeq elements appeared equally often during the pre-scan learning task. During scanning, the order of stimuli did not obey the transitions experienced during training because the protocol was optimized for several different analyses. For instance, we wanted to be able to compare the BOLD response to D stimuli in trials where it was preceded by A, B, C with trials in which this was not the case. This also meant that during scanning, stimulus frequencies were not entirely balanced but the relevant transitions between RewSeq and ConSeq sequences that entered our contrasts were always balanced. Custom-written MATLAB (The MathWorks, Inc., Natick, Massachusetts, US) code, with different constraints for the pre-scan learning task and the fMRI scanning task, was thus used to generate the respective pseudo-random sequences of highlighted stimuli. All stimulus presentation was programmed in MATLAB and performed using the Psychophysics Toolbox[70].

**Statistical analysis of behavior from pre-scan learning task.** During the pre-scan learning task, RTs to the reward and bridge provided measures of the presence and absence of reward anticipation, respectively. This was because the bridge was shown at unpredictable times whereas reward only occurred when the correct sequence ABCD had been experienced. RTs to the six occurrences of reward and bridge per block were averaged for each block and across the three contexts, yielding eight RT measures per person (four blocks per context on day 1 and day 2). An 8 × 2 repeated-measures ANOVA with factors block and stimulus type (reward/bridge) thus measured effects of reward anticipation (Fig. 1b). All ANOVAs were performed in JASP (JASP Team (2018), https://jasp-stats.org) and were JZS Bayes factor ANOVAs[71,72] with default prior scales, which allowed both frequentist and Bayesian statistics and thus enabled measuring the likelihood of the null hypothesis.

RT changes to the first movement of a trajectory from one highlighted stimulus to the next (i.e., when one highlighted stimulus had just been reached and the next stimulus was highlighted by the computer) were used to probe the anticipation of stimulus transitions. If the currently highlighted stimulus can serve to predict the next one, participants should be able to move the agent in the direction of the newly highlighted stimulus faster. We measured the RT to the first agent movement separately for transitions AB, BC, and CD, for both RewSeq and ConSeq, averaged across blocks and contexts. Hence, we performed a 2 × 3 repeated-measures ANOVA with factors sequence (RewSeq/ConSeq) and transition (AB, BC, CD; Fig. 1c). Note that all the transitions included in this ANOVA were experienced equally frequently and were thus equally predictable from a probabilistic point of view. To examine consolidation of transition knowledge across days, this ANOVA was also run as a 2 × 2 × 3 ANOVA with an additional factor Day (Supplementary Note 4; Supplementary Fig. 1f).

To assess any progression in participants' ability to complete the entire sequence (a way of probing explicit sequence knowledge), we calculated the percentage of times each of the four elements A, B, C and D was included in the sequences participants produced during the sequence completion task at the end of each learning block. We averaged across the three repetitions and across contexts, which resulted in a 4 × 8 repeated-measures ANOVA with factors Element (A/B/C/D) and Block (Fig. 1d).

**Statistical analysis of behavior during fMRI scanning task.** RTs produced to flowers shown during the fMRI scanning task were fitted using a linear regression with the following explanatory variables: (1) Distance in links to previous highlight; (2) transition frequency (for more details see trial-by-trial model below); (3) any member or RewSeq; (4) any member of ConSeq; (5) constant term. RTs during scanning were faster for shorter distances and for elements of the RewSeq ($p$ (LinkDistance) = 4.67e-12, $t(25)$ = 19.73; $p$(memberRewSeq) = 0.021, $t(25)$ = −6.75; for all other $p$ apart from constant term: $p > 0.1$).

We summarized the performance on the memory test performed after the two scan sessions (i.e. as the last test during data acquisition). We examined the memory of the RewSeq and ConSeq sequence elements as the percentage of correct responses in the memory task on stimuli shown in the correct place and position (recognition) and those shown in either the wrong color or position (error detection). We performed a 2 × 2 repeated-measures ANOVA with factors sequence (R/C) × type of recall (recognition/error detection). The overall percentage of correctly remembered stimuli was taken as a measure for overall task space knowledge.

**Trial-by-trial model of statistical relationships.** We designed a simple model to capture the underlying statistical relationships present in the scan task. The aim was to obtain a trial-by-trial measure of the frequency (or probability) with which participants expected each of the possible transitions from one element to another within the task space. For the first scan session, we set the prior to the frequencies experienced during the pre-scan learning task. Frequency here refers to the actual number of times a transitions was experienced during training, in integer numbers. These were saved in a 13 × 13 asymmetrical matrix TF (12 stimuli + reward), storing, in each cell TF $(i, j)$, the value of the frequency of going from stimulus $i$ to stimulus $j$ (as shown in Supplementary Fig. 4c). Note that the sum along the rows and columns indicated how often participants experienced stimulus $i$ and $j$ overall, respectively. During the pre-scan learning task, each stimulus was shown equally often, so normalizing TF by rows or columns, thus transforming frequencies into probabilities, did not have any effect apart from a scaling of the matrix as a whole. For each trial in the scan task (starting from trial 2), the prior was then updated for the experienced transition from $i$ to $j$ by adding 1, i.e.

$$\text{TF}(i, j) = \text{TF}(i, j) + 1;$$

This meant that every experienced transition carried equal weight during updating. The matrix TF was then demeaned and used as a prior for the next trial. We decided to demean to ensure that the overall scale of the numbers remained unchanged while at the same time avoiding a monotonically increasing parametric regressor. We checked carefully whether this decision mattered: for example, the correlation between the demeaned regressor and an alternative that did not include this demeaning step but detrended the resulting regressor at the end, was $r$ = 0.9936.

For computing transition probability TP$(j|i)$, the demeaning term from the TF computation was remembered, added back onto the updated TF matrix, and each row of TF was then normalized by the sum of this row (reflecting the frequency with which the initial stimulus $i$ occurred) to obtain TP, and thus a measure of how likely it was to go from stimulus $i$ to $j$, given $i$. A parametric regressor was constructed based on the expected transition frequency of going from the previous $(i)$ to the current $(j)$ stimulus (i.e. based on the TF cell before the update for the current trial was performed). This regressor was time-locked to the presentation of the current stimulus $j$. For the second scan session, the same model was used with the only difference being that the prior was set to the transition frequencies from the first day of scanning.

In modified versions of GLM2 (see below), conditional transition probability either replaced the parametric regressor of transition frequency, or both were included simultaneously. This was possible because during scanning, not all stimuli

appeared equally often, so transition frequency and transition probability were distinguishable. We simultaneously kept a record of stimulus frequency per se by adding +1 to a 1 × 12 vector tracking how often a stimulus was experienced.

To establish whether tracking transitions and thus statistical task regularities provided a superior prediction of upcoming elements than a simpler prior of spatial distance, we calculated the mean probability of all experienced transitions using (a) knowledge of transition frequencies, (b) knowledge of spatial distances, or (c) a flat prior, i.e. equal likelihood of any transition. For spatial distance coding, we assumed that the probability of a transition linearly decreases with increasing distance but conclusions remained unchanged using exponential or logarithmic decays. Because the resulting mean probability is a function of the stimulus order which was identical across participants (despite randomization of blocks), mean values are reported without statistical tests.

**FMRI Data acquisition.** Images measuring the blood oxygenation-level-dependent (BOLD) signal were acquired on a Siemens Prisma 3T MRI scanner using a multiband T2*-weighted echo planar imaging (EPI) sequence with acceleration factor of four and using a 32-channel head-coil. Slices were acquired in interleaved order and at an oblique angle of −30° to the AC-PC line to reduce signal dropout in orbitofrontal cortex. The voxel size was 2 × 2 × 2 mm with a 1-mm gap; TE = 32.4 ms; repetition time = 1354 ms; flip angle = 74°; number of slices: 72. One run (~20 min) contained approximately 850 volumes. A fieldmap (2 × 2 × 2 mm) was obtained for each subject to allow for corrections in geometric distortions. In one of the two scanning sessions of each participant, a structural MPRAGE scan was acquired with 192 slices; slice thickness = 1 mm; TR = 1900 ms; TE = 3.94 ms; voxel size = 1 × 1 × 1 mm. In addition, physiological recordings were taken during the functional MRI blocks to measure the participant's pulse and breathing.

**FMRI data preprocessing.** Image preprocessing was implemented in FMRIB Software Library (FSL)[73] and consisted of bias correction using the bias field obtained from segmentation[74], motion correction[75], distortion correction using fieldmaps, brain extraction, high-pass filtering, and spatial smoothing with a 5-mm FWHM kernel. A hard regression was performed to regress out noise explained by 24 motion regressors (the original six produced by FLIRT, their derivatives, and the resulting 12 regressors squared; see e.g.[76]) and by 33 physiological noise regressors created using the PNM toolbox (because of the short TR, these were not voxel-wise regressors: oc = 4; or = 4; multc = 2; multr = 2; rvt[77]). In addition, conservative independent component analysis was used to identify and remove obvious artifacts (using MELODIC in Fmrib's Software Library; http://fsl.fmrib.ox.ac.uk/) and ICA noise components were regressed out of the data using a soft regression. Images were registered to the high-resolution structural image (BBR) and then the standard MNI152 template using nonlinear registration (12 degrees of freedom)[78].

**FMRI Data analysis using general linear models.** We constructed two general linear models (GLMs). The first (**GLM1**) was designed to probe differences in BOLD between the reinforced sequence RewSeq and the equally frequent sequence ConSeq. It contained 18 regressors of interest:

(1) rr – the occurrence of an RewSeq element (A, B, C, or D) after another RewSeq element
(2) xr – the occurrence of an RewSeq element after an element not part of RewSeq
(3) corrOrderRewSeq – occurrences of RewSeq that follow the correct order, i.e. B after A (A<u>B</u>), C after AB (AB<u>C</u>), or D after ABC (ABC<u>D</u>)
(4) Spatial neighbors of D (any element that can be reached within one step from D)
(5) Goal element D when preceded by a spatial neighbor
(6) Goal element D preceded by any other element
(7) Starting element A
(8–14) the equivalent regressors for the control sequence ConSeq
(15–16) regressors (1–2) but for the less frequent control sequence ConSeq2
(17) Picture of treasure
(18) Picture of flower.

The contrasts of interest in GLM1 were rr-cc (Fig. 2a), which probes any differences in BOLD to the second of two successive stimuli from RewSeq compared to ConSeq (for more details, see Results); corrOrderRewSeq-corrOrderConSeq which probes differences between RewSeq sand ConSeq when transitioning through the correctly ordered associations (Fig. 3b and Supplementary Fig. 3). Both of these measured knowledge that had formed as a result of reinforcement learning. One other contrast of interest in GLM1 related to the spreading of reward in space: we compared spatial neighbors of D with spatial neighbors of D′ (Fig. 5). And a final contrast compared BOLD responses to the starting elements A and A′ (Fig. 6).

The second GLM (**GLM2**) was designed to probe statistical learning and therefore did not split the onsets of RewSeq and ConSeq elements. It contained 8 regressors of interest:

(1) The onset of any stimulus (RewSeq, ConSeq, or ConSeq2 i.e. all 12 on the 3 × 4 map)
(2) Parametric effect of spatial link distance (i.e., the number of steps it would take to walk from the previously highlighted stimulus to the current one)
(3) Parametric effect of transition frequency extracted from the trial-by-trial model (i.e., estimated frequency of going through the transition that led to the currently highlighted stimulus)
(4) Parametric effect of any reward expectation (A, AB, ABC, and ABCD)
(5) Parametric effect of any statistical sequence violation (i.e. $\overline{A}x$, $A'x$, $AB\overline{x}$, $A'B'$ $\underline{x}$, $ABC\overline{x}$, and $A'B'C'\overline{x}$, where x is not the stimulus that would follow the correct order of the sequence)
(6) Parametric effect of expected stimulus frequency (updated on a trial-by-trial basis as explained above)
(7) Picture of treasure
(8) Picture of flower.

The relevant contrasts in GLM2 were the parametric effect of spatial link distance and transition frequency (Fig. 4) which were taken as measures for statistical learning. The responses to treasure and flowers were virtually identical in both GLM1 and GLM2 and are shown for the ventral striatum based on GLM2 (Fig. 6).

All regressors were convolved with the double-gamma HRF in FEAT and GLMs were estimated for each subject and session. The shared variance between any two regressors (Pearson's $r^2$) in these convolved GLMs did not exceed 0.25 for either GLM1 or GLM2 (Supplementary Fig. 2d). The analysis involved multiple further levels: at the second level, the sessions from the same day in each subject were combined using fixed-effects (second stage); at the third level, the sessions from the two days of each subject were combined using FLAME1 (mixed-effects[79]). Finally, the fourth level corresponded to the group analysis and used FLAME1 to combine across subjects. We originally performed analyses with a cluster-forming threshold of $p < 0.1$ ($z > 2.3$) but repeated them with a more conservative cluster-forming threshold of $p < 0.001$ ($z > 3.1$), following recent recommendations[80]. This did not change the key results: (1) Sequence fusion RewSeq-ConSeq (Fig. 2a) still shows significant cluster-corrected results in temporal pole and mPFC, merely the extension to pOFC does not survive correction at the stricter threshold; (2) spatial distance, transition and stimulus frequency coding (Supplementary Fig. 4a, b) do not change apart from few voxels around the edge of clusters; (3) correct sequence order (Supplementary Fig. 3) becomes weaker but remains cluster-level significant in pOFC; (4) only lateral ventral striatum (putamen) is no longer significant for the contrast A in RewSeq-ConSeq (Fig. 6b).

Because of a priori hypotheses, significance in the amygdala was established using a mask. For the initial ROI-defining contrast rr-cc we had no strong hypotheses about the location within the amygdala, so we applied non-parametric threshold-free cluster enhancement (TFCE[33]) to establish significance at $p < 0.05$ within an anatomical mask of bilateral amygdala (Harvard Subcortical Atlas). For establishing effects of spatial spread, we used the average amygdala coordinate from[38] ($x = −21$, $y = −6$, $z = −19$) and tested for significance in a 3-mm sphere around this coordinate. For full transparency, we also report the cluster extent at uncorrected levels: for the initial contrast (rr-cc) the peak in the amygdala was $z = 2.8$ within the amygdala anatomical mask and $z = 2.97$ in the same cluster but just adjacent reaching into the hippocampus. The cluster extent at $p < 0.01$, $z > 2.3$ was 27 voxels within the amygdala mask in the right hemisphere, and 46 in the right and 11 in the left hemisphere when considering voxels of the same cluster within either amygdala or adjacent hippocampus, i.e. including the border. For the spatial spread contrast, the strongest activation was located within the anatomical amygdala mask at $z = 3.0$. The cluster extent at $p < 0.01$, $z > 2.3$ is 19 voxels.

Finally, because of a large body of literature relating the hippocampus to statistical learning, we also constructed a 3-mm spherical ROI in bilateral hippocampus based on the peak coordinate in[58] (Left: −18, −19, −22; Right: 24, −25, −22).

To further examine some of the effects revealed in GLM1 and GLM2, two slightly modified variants of the original models were tested. First, to probe whether effects obtained for the contrast rr-cc in GLM1 (Fig. 2a) were modulated by the interval between subsequent RewSeq or ConSeq stimuli, four additional parametric regressors were added to GLM1 and modeled the temporal distances between the preceding and current stimulus for the four columns: rr, cc, xr, and xc. Second, one regressor was replaced in, or added to, GLM2 so that instead of, or in addition to, transition frequency, transition probability was modeled. These variables were quite correlated ($r^2 = 0.72$) and were compared in their performance both separately and when competing for variance to establish if one explained the ROI's BOLD signals better than the other. The two extended GLMs were convolved using MATLAB's conv.m with spm_hrf.m and then fit directly to the ROI time-courses.

**ROI selection, parameter estimate, and time-course extraction**. Spheres of 5 mm were constructed around the peaks from the above described contrast rr-cc in mPFC (peak MNI coordinate 0, 40,−6, $z = 4.0$), temporal pole (peak MNI coordinate 48, 10, −28, $z = 4.2$) and pOFC (peak MNI coordinate right: 30, 18, −22, $z = 3.73$ and left: −38, 16, −20, $z = 3.32$), and a sphere of 3 mm was constructed around the amygdala because of the small size of this region (peak MNI coordinate right: 28, −6, −22, $z = 2.9$; left 26, −8, −26, $z = 2.8$; see Supplementary Fig. 2b).

An anatomical ROI for ventral striatum (nucleus accumbens) was constructed from the Harvard Subcortical Atlas (probability threshold 0.25).

Parameter estimates extracted from these regions are in some cases shown for illustration only (Fig. 2b). Additional statistical tests on parameter estimates in the form of simple $t$-tests were only performed when these were orthogonal to the contrast based on which the spheres were defined (e.g. Fig. 2c for temporal delay effects or Figs. 3d and 4d for progression of effects over the duration of scanning). Changes in parameter estimates over the six blocks (e.g. Figs. 3d and 4d) were established by fitting a linear regression model with an effect of time (linear trend), and if appropriate any main effects (e.g., contrast) and interactions, to the parameter estimates extracted from each block based on GLM1 or GLM2. All $t$-tests were two-sided except in one case: when testing for a modulation of the rr effect by the time between successive stimuli, we expected repetition suppression to be stronger for shorter time intervals (Fig. 2c and Supplementary Fig. 2a). Importantly, while parameter estimates extracted from the ROI sphere in temporal pole were only just significant in this test, there was a large cluster of voxels significant in this contrast which was located immediately adjacent to our ROI in temporal pole.

Time-series were extracted from the same spheres described above. Again, no statistical tests were performed on these data and they serve illustrative purposes only. Time-series were up-sampled slightly for illustration (original TR = 1.354 sec, up-sampled TR = 0.5 sec). For illustrating the effect of inter-stimulus interval on the suppression effects reported in Fig. 2, the short ISI condition contains any trials with ISI < 2 s, the mid ISI condition contains ISIs between 2–3.5 s and the long ISI condition ISIs > 3.5 s. This division ensured all bins contained roughly equal numbers of trials (Fig. 2d). Similarly, for the effects of spatial link distance and transition frequency, trials were split into three equally large bins (Fig. 4c).

**Reporting summary**. Further information on research design is available in the Nature Research Reporting Summary linked to this article.

## Data availability
The data underlying all figures are provided as part of the Open Science Framework (OSF; https://osf.io/y325a; Digital Object Identifier (DOI): https://doi.org/10.17605/OSF.IO/Y325A). The OSF project contains a zip folder with all source Data files (.mat), a spreadsheet detailing the names of the source data files and all scripts relevant for producing the figures (make_figX.m). Unthresholded fMRI maps of all contrasts are available on Neurovault (https://identifiers.org/neurovault.collection:5765).

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

## Acknowledgements

MCKF was funded by a Sir Henry Wellcome Fellowship (103184/Z/13/Z), AS by a Wellcome Trust PhD studentship (203836/Z/16/Z), and MFSR, DEAJ, and MKW by an MRC grant (MR/P024955/1) and a Wellcome Senior Investigator Award (WT100973AIA). We would like to thank all members of the Behrens and Rushworth labs for great discussions on this project.

## Author contributions

M.C.K.F. and M.F.S.R. designed the experiment, M.C.K.F. and A.S. conducted the experiment, M.C.K.F., M.K.W. and M.F.S.R. conceived analyses, M.C.K.F., A.S. and D.E. A.J. conducted analyses, and all authors wrote the manuscript.

## Competing interests

The authors have no competing interests to declare.
