## [Peer Review File · Nature Communications]

Reviewers' Comments:

Reviewer #1:

Remarks to the Author:

Klein-Flugge and colleagues teach subjects about rewarded and unrewarded sequences on a grid of symbols, then study the neural correlates of various distinct types of associations implied by this training. BOLD activity reflects signatures of several different sorts of associations, which seem to differentiate two sets of areas as having information about the statistical regularities in the grid, and/or their predictiveness of reward. My overall sense is there are interesting and suggestive observations here, but it's a tough read and a bit hard to pin down exactly what the conclusions are. I think this is in part because there does not seem to be a common, formal sort of statistical or computational definition of the various statistical structures learned by the subjects, which would motivate the different analyses and help to interpret the results. For instance, the transition learning model is not clearly related to the initial CSS analysis, and makes many different assumptions than that one. Instead there is a somewhat more scattershot series of contrasts that are more informally motivated and interpreted, and it's hard to decide how they all fit together and if they are mutually consistent. I think more followup or refinement would be helpful to really nail down the results, and I offer some suggestions in this regard below.

Major comments:

1. A major line of argument in the paper is that areas of interest are distinguished (roughly into two groups) on the basis of whether or not particular contrasts come up significant there. Thus the patterns of effects and noneffects in Figures 2c, 3b, and 4b are used to suggest two distinct networks (see Figure 7a,b), and both of these are distinguished from striatum, where none of the effects are significant. But if I am not mistaken, it is not formally tested or demonstrated that the areas' responses actually significantly differ from one another in this respect. As the slogan goes, a difference in significance is not necessarily itself significant; see Nieuwenhaus et al. "Erroneous analysis of interactions."

2. As I understand it, the analyses are meant to distinguish BOLD activity sensitive to two broad sorts of relationships in the task: those concerning the relationship of stimuli to reward, and those concerning the relationship of stimuli to one another. But the set of analyses presented are overlapping and mutually inconsistent in a way that makes it hard to really nail down what is ultimately going on.

For instance, most of the analyses are based on peaks of an initial "rr vs cc" contrast that seeks areas that show cross-stimulus-suppression, moreso for stimuli involved in the rewarded than the unrewarded sequence but not otherwise distinguishing among different stimulus relationships. Followup tests then detect other effects in these areas that complicate this picture.

For instance, the tests in Figure 4 seek sensitivity to spatial distance and transition frequency, both defined regardless of reward. But this is different from the initial ROI-defining contrast in several respects. It is reward-independent. It is directed (i.e. it distinguishes A->B transitions from B->A). And it seems to be characterized by greater activation to stimuli when they are more expected, the opposite of what is implied by cross stimulus suppression. Which is right?

That the descriptions of activity implied by these various contrasts are mutually inconsistent would seem to imply that the ultimate activity may be something else than straightforwardly implied by either regressor. For instance, it might be possible that the significant cross stimulus suppression effect in the first instance is actually driven by a subset of *directed* transitions, such that rr > cc on

average but there is only suppression for pairs in which the first r comes before the second r in the sequence. (This might happen if directed predictive activity at each step drives cross-stimulus suppression, rather than representational change making all stimuli in a group more similar to one another.)

Conversely, why does it make sense to look for reward-agnostic distance or conditional frequency effects in areas already screened for reward sensitivity? And if you find it, is it really reward-agnostic? It seems at least possible that in this case the distance and conditional frequency sensitivity are actually driven by such effects only within the rewarded sequence? Finally, what are we to make of the superposition of frequency related facilitation and cross-stimulus suppression. For an A->B transition within the rewarded sequence, there is less suppression than a B->A one? Or what?

Figure 3 raises analogous questions of how to reconcile the CSS findings with the various other ones. Perhaps it would be helpful to pursue an increasingly refined CSS view (i.e. is there more or less cross stimulus suppression for stimuli in reward sequences, spatially distinct, etc), breaking down the rr vs cc effects, more in the spirit of 2c,d.

3. The foregoing points out the importance of clearly distinguishing activity related directed vs undirected stimulus-stimulus relationships. Another issue which could be better reconciled is the relationship between spatial adjacency and transition sequences. As I understand it, during training there is an instructed stimulus-stimulus sequence that is spatially disjoint, but subjects actually experience these as mediated by spatial paths over a series of intermediate stimuli, which are not experimenter controlled and apparently never directly used in analysis. Then in the scanning phase, a jumping cursor now presents stimulus-stimulus sequences skipping the spatially intervening stimuli. Given the results in Figure 4d, it seems quite possible that subjects initially learn associations (eg conditional transition frequencies) not over the instructed stimulus pairs directly, but over the spatially adjacent transitions actually experienced in training. This raises a series of questions in that virtually any of the results in the paper (from rr vs cc on down) could at least be potentially better explained in these terms. For instance, does the rr vs cc effect also hold for stimuli which are not in the official sequences but were included in the paths? It also provides the opportunity for a unified, formal view of the result from Figure 4d, in which a single learning model trained on experience which is initially temporally smooth (the actual paths taken by the subjects) and later temporally jumpy (the cursor in-scanner) interpolates from one regressor to the other. Similarly does the neighborhood effect in Figure 5 actually just follow adjacent squares, or is it sensitive to the series of spatially adjacent transitions into or out of the reward square actually experienced in training? That is, the notion of "spatial spread" here is confounded by temporal spread, given the experienced path statistics.

Minor points

- How were subjects instructed? They were explicitly told they were trying to discover the rewarded sequence?

- CS is unfortunate terminology for control sequence, given that it often means conditioned stimulus in this literature.

- Figure 2c is missing axis labels

- I'm willing to concede to convention, authority, or the authors' poetic license here, but I'm personally wary about the verb "encoded" used extensively in this article to mean "is correlated with," as in "BOLD activity in area x is correlated with regressor y." I realize this is shorthand but to my ear this particular word smacks a bit of assigning a purpose to the modulation when of course, absent any

causal manipulation (and given the possibility that significant correlations might be driven by other correlated regressors) there is no reason to assume the activity is actually doing anything or "encoding" this particular signal. I find this most potentially confusing in the section on ventral striatum which concludes the area "plays no role" and has "no involvement" in this encoding, which combines this oddly active phrasing with null hypothesis affirmation.

- I don't see why expectation building up as the reward sequence progresses (figure 3c) has anything at all to do with the idea that the structure is learned by backpropagation (line 223). This will be true of any notion of temporally discounted reward expectancy or reward proximity, regardless how the associations underlying it were acquired or how it is computed.

- Is the A-B A-B-C A-B-C-D activity increase really dependent on the third and fourth order structure? Would you see the same thing if you looked for X-B-C and X-X-C-D?

- The transition learning model is confusing. What does it mean to demean it (by rows? columns??)? And given that it's expressed in transition counts, how strong is the prior? Is it initialized with elements between 0 and 1 according to the trained probabilities, but then counts up by adding 1? In this case the "prior" would wash out very quickly and play almost no role. Why not initialize with the number of transitions actually experienced during training?

- Why are the two types of test questions in supp figure 4 described as explicit vs implicit recall? I think a memory person would describe this as a recognition (not a recall) test, and I don't see why endorsing lures should be implicit but correct recognition should be explicit?

- The discussion of Figure 1e suggests the CS is learned in a forward manner, in contrast to RS. I don't see any evidence anywhere in the results that the CS is learned at all?

Reviewer #2:

Remarks to the Author:

The paper „Multiple associative structures created by reinforcement and incidental statistical learning mechanisms" asks how the brain encodes the knowledge resulting from reinforcement learning. It contrasts associative structures that arose from reinforced stimulus sequences against structures that arose from non-reinforced stimulus sequences. The findings include reduced activity to reinforced sequence elements (compared to non-reinforced sequence elements) in medial prefrontal cortex, temporal pole, posterior orbitofrontal cortex (pOFC) and amygdala/hippocampus. Among these regions, correct reinforced sequences (compared to correct non-reinforced sequences) activated temporal pole and pOFC. Both spatially close and frequently experienced transitions were associated with medial prefrontal cortex and amygdala/hippocampus activation, regardless of sequence type. Spatial proximity to final elements from reinforced stimulus sequences activated amygdala/hippocampus more than final elements from non-reinforced stimulus sequences. Finally, the ventral striatum responded to the first element of the reinforced stimulus sequence over and above the first element of the non-reinforced sequence.

There are several aspects of the study to like, particularly the detailed analysis of how reinforcement affects the encoding of time and space within sequences. However, some aspects temper enthusiasm, particularly how the authors aim to claim novelty, the lack of explanation for the unexpected finding that typical reward regions are less active for reinforced than non-reinforced sequence elements and how the ventral striatum results are framed.

I noticed the following issues that should be addressed:

1. The framing of the paper aims to set up a contrast between process (conditioning) and product (referred to as "learned knowledge") of learning. If I understand correctly, learning about reinforced sequences essentially corresponds to a situation of higher-order conditioning. This kind of learning has been investigated before (e.g., Seymour et al., 2004, *Nature*; Gilboa et al., 2014, *Current Biology*; Pauli et al., 2015, *Journal of Neuroscience*) which the authors may want to spell out. More importantly, neuroimaging took place in extinction, which should have led to continued learning (namely extinction learning), a point that the authors gloss over and that makes the presumed contrast between process and product much less clear-cut. By extension, the authors may have implicitly dealt with neural regions that show extinction learning at different time scales (for related work see e.g., Glascher & Büchel, 2005, *Neuron*). The fact that the identified regions show relations to reinforcement throughout the imaging phase could suggest that these regions learn slowly. This point seems to obliterate the present framing of the paper and in the minimum needs to be mentioned.

2. It seems somewhat counterintuitive that a direct comparison of elements from the reinforced sequences should activate typical reinforcement regions such as OFC and medial prefrontal cortex *less* than elements from the control sequences. This result is counterintuitive and critical because the regions for subsequent ROI analysis are identified by this analysis.

3. If I understand correctly, ventral striatum activity follows exactly what one would expect from a region that implements temporal difference (TD) learning (perhaps with the exception that extinction appears to take relatively long. However this may partly be explained by the long acquisition period and by specifics of the instructions for the scanner session, as also suggested by the other findings of persistent reinforcement-related activity). In particular, according to the TD model the earliest element (A) of the reinforced sequence should be encoded (uniquely in case of perfect learning) as predictive of reinforcement and that is exactly what the data show. However, the authors do not seem to interpret (and value) this finding as indicative of learned knowledge. Instead, they focus on negative findings in the ventral striatum (even though negative findings are hard to interpret and non-ventral striatal regions have been involved in both model-based and model-free learning previously (see e.g. the summary figure of the comment by Chen et al., 2015, *Journal of Neurophysiology*)). Moreover, it is conceivable that spatial proximity to initial elements A is coded more strongly in the striatum than to final elements. In any case, a more balanced interpretation of the ventral striatum findings seems warranted.

4. The authors make a clear-cut distinction between time and space. However, in their task, the two appear to be correlated to some degree by design (it takes longer to move to spatially more distant elements). If the authors cannot exclude this issue they should discuss it.

5. The summary appears to lack a concluding interpretative sentence

6. Did the authors also test participants' ability to produce the correct control sequence after each pre-scan block?

7. Conceptually, it does not seem to make much sense to combine cluster-level correction with small volume correction/ROI analyses. Voxel-level correction should be used instead

8. Was the number of neighbors of the final element exactly matched for reinforced and non-reinforced sequences? At least the example in figure 1 suggests otherwise. A potential confound with an incomplete match would be a difference in (spatially defined) uncertainty

9. In the last results paragraph, the sentence "The encoding of the first predictive element A was not

specific to ventral striatum" seems to be a bit lost.

Reviewer #3:

Remarks to the Author:

In this manuscript by Miriam Klein-Flugge et al., the authors explore how cognitive maps are formed through reinforcement and updated through experience in humans. Specifically, the authors state three goals: to explore 1) the neural signature of knowledge learned through reinforcement, 2) the neural mechanisms that track statistical relationships over time, and 3) whether the eligibility of reward might be assigned to stimuli in close spatial proximity to reward (a similar process has been shown for stimuli temporally close to reward).

Through a carefully designed sequence learning task where the start of the sequence was not signaled to the participant, the authors confirmed a central prediction of a popular computational model of reinforcement learning named TD RL: sequence knowledge is constructed backwards from reward. Using an impressively thought-out and carefully designed task during MRI scanning, the authors were able to test the neural signature of sequence knowledge learned during the previous task independent of task demands. The authors identified four brain regions that coded for reinforced sequence knowledge: medial prefrontal cortex (mPFC), posterior orbitofrontal cortex (pOFC), temporal pole, and a region at the border of amygdala and hippocampus. More detailed and careful analyses revealed that the path to reward (elements in the correct sequence) was coded by temporal pole and pOFC as an inflexible map, whereas knowledge about the statistics of transitions between elements of the task space (i.e., a more flexible map of task space) were coded by mPFC and amygdala/hippocampus. A more central amygdala region responded to elements that were spatially adjacent to the element previously associated with reward during training, complementing previous research that uncovered a role for the amygdala in spreading of reward to elements that were spatially proximal to reward. None of these regions coded reward information directly, suggesting that the ventral striatum may be involved in forming the knowledge about rewarded sequences, but did not carry a map of that knowledge in the absence of reward.

I find the paper timely, important, and of interest to a wide readership. There has been a lot of interest in better understanding how cognitive maps are formed during learning and used during decision making. This study is meticulous, the authors do a good job at presenting complex analyses clearly, and the results confirm some clear predictions and present some new intriguing effects (e.g., involvement of temporal pole).

I recommend publication. I have only minor points to bring up:

1) I was surprised that the authors stated that they had no a priori reason to look at the hippocampus. The authors mention cognitive maps as described by Tolman. A rich literature on cognitive maps implicates the hippocampus, hence my surprise. If the authors would elaborate on why they had no a priori reason to look at the hippocampus, that would be useful to readers like me.

2) The authors make a strong argument for the inflexibility of correct sequence mapping in pOFC and temporal pole. The parameter estimates plotted in Figure 3d seem variable across blocks, with perhaps an effect of day (block 1-3 vs block 4-6). I appreciate the authors' attempts to support the null using Bayes factors, but the methods include few details of what priors were used in this test. Inclusion of these details is appreciated.

3) The authors mention a second control sequence in the methods, but include no analyses of this

CS2. What was the rationale behind this second control sequence and can the authors offer justification for not including any analyses of this CS2, even in the supplement.

4) Do the authors have any behavioral data to test the spatial spreading of reward? For example, did the participants choose between a D-adjacent element and a non-rewarded element during the choice task?

5) Finally, I would recommend that the authors make their statistical fMRI maps publicly available, for example through neurovault.org. This will allow the reader to have a full picture of the exploratory analyses and can facilitate future meta-analyses.

Reviewers' comments:

Reviewer #1 (Remarks to the Author):

Klein-Flugge and colleagues teach subjects about rewarded and unrewarded sequences on a grid of symbols, then study the neural correlates of various distinct types of associations implied by this training. BOLD activity reflects signatures of several different sorts of associations, which seem to differentiate two sets of areas as having information about the statistical regularities in the grid, and/or their predictiveness of reward. My overall sense is there are interesting and suggestive observations here, but it's a tough read and a bit hard to pin down exactly what the conclusions are. I think this is in part because there does not seem to be a common, formal sort of statistical or computational definition of the various statistical structures learned by the subjects, which would motivate the different analyses and help to interpret the results. For instance, the transition learning model is not clearly related to the initial CSS analysis, and makes many different assumptions than that one. Instead there is a somewhat more scattershot series of contrasts that are more informally motivated and interpreted, and it's hard to decide how they all fit together and if they are mutually consistent. I think more followup or refinement would be helpful to really nail down the results, and I offer some suggestions in this regard below.

Thank you very much for your feedback and thoughtful comments below. We appreciate that the paper summarizes results that emerge from a number of different analyses and that it can seem as if they do not hang together easily. When originally writing the paper, we thought that it was a strength to show that task-related behaviour and brain activity cannot be accounted for by a unitary mechanism but only by quite distinct mechanisms mapping onto distinct neural structures. While this might not be the way that a computer scientist would design a mechanism for acquiring knowledge of task structure and reward contingencies, we think that it is possible, perhaps even likely, that the brain would learn task structures via multiple mechanisms with distinct capabilities that have evolved partly separately. But we can also see in many of the suggestions below that there are ways to test these ideas more rigorously and to be clearer in communicating them.

Part of the reason for why there isn't a common formal definition or model of the different structures/types of associations we examine, is that (a) they are different in nature and that (b) unlike in other RL tasks, the reward-learning is not occurring 'online' during scanning. In general, RL models do not capture statistical relationships that are reward-unrelated whereas models that capture statistical relationships tend to be built upon incidental observations, which are usually assumed to be independent of reward (e.g. in Tolman's maze, the rats that learn in the absence of reward still learn the maze structure). That's why we constructed two different GLMs and have two strands of analyses for examining these two types of questions. Broadly speaking, one GLM, which was constructed to look at the RL learning side of the task, models stimuli that were in the rewarded sequences separately from those that were not. The second GLM, which was designed to look at statistical learning, treats every stimulus the same and looks at relationships between them that are true across all 12 stimuli and modulated parametrically. One could imagine a hybrid model which learns both reward-relationships as well as statistical associations simultaneously. However, when we designed the task, to us the novel contribution seemed to be to investigate behavioral and neural correlates of 'ordered' and more

complex (i.e. multi-step) reward-structures. Statistical learning has been associated with mPFC-hippocampal circuits before (which we acknowledge). We therefore took the decision to study the associative structures of such multi-step sequential learning which is why the learning aspect of the task, which a computational model would capture nicely, is not actually examined in the scanner. So while we do not offer trial-by-trial computational models to predict neural responses as is commonly done for RL learning when learning happens during scanning, we agree with many of the points raised below and have tried to flesh out commonalities and differences in our various contrasts and findings in order to make the manuscript easier to follow and convincing in its conclusions. We hope that you will find that this has significantly improved the manuscript. We start by making our approach as explicit as possible in the Introduction, for example as follows (pp2-3):

Our aim in the current study was to take an alternative approach to investigate the neural mechanisms associated with model-free and model-based learning. Rather than focusing on the PEs that occur during learning, instead we focus here on the knowledge or ‘associative structures’ that are formed by, and subsequently guide, the learning process. **In order to do this we examined a situation in which participants did not just learn associations between a single predictive stimulus and subsequent reward but instead had the opportunity to learn both predictive links between chains of stimuli leading up to reward^{2,20,21} and statistical relationships between stimuli regardless of whether they were predictive of reward.** This allowed us to test whether the associative structures derived from different learning processes might prove more distinguishable than their PEs. **We therefore undertook two main series of analyses of behavior and of neural activity that contrasted the consequences of these two types of learning in our experiment.**

Major comments:

1. A major line of argument in the paper is that areas of interest are distinguished (roughly into two groups) on the basis of whether or not particular contrasts come up significant there. Thus the patterns of effects and noneffects in Figures 2c, 3b, and 4b are used to suggest two distinct networks (see Figure 7a,b), and both of these are distinguished from striatum, where none of the effects are significant. But if I am not mistaken, it is not formally tested or demonstrated that the areas’ responses actually significantly differ from one another in this respect. As the slogan goes, a difference in significance is not necessarily itself significant; see Nieuwenhaus et al. “Erroneous analysis of interactions.”

We entirely agree and apologize for the oversight. The main results figures that we base the argument for the existence of two potential networks on, namely mPFC with amyg/hippo versus tempPole with pOFC, are Figures 3b (correct order encoding, which are the stimulus-stimulus associations leading to

reward) versus 4b (both spatial distance and transition frequency which are statistically learned stimulus-stimulus associations learned independently of reward).

We will answer this point in two ways:

- (a) By reporting the appropriate statistical interaction test requested by the reviewer.
- (b) By laying out more clearly, in terms of brain functional organization why we had a priori reasons to expect that pOFC and temporal pole versus amyg/hippo and mPFC might form functional units, respectively.

Part (a)

We now ran ANOVAs on the parameter estimates extracted from the contrasts associated with these two claims:

- (1) A 2x2 ANOVA with factor Network (tempPole & pOFC versus amyg/hippo and mPFC) and Node (anterior/posterior, where pOFC and mPFC are the anterior nodes of the two networks both found within frontal cortex and tempPole and amyg/hippo are posterior nodes in the temporal lobe, respectively) based on the parameter estimates in Fig 3b for correct sequence order
- (2) A 2x2x2 ANOVA with factors Network, Node, and Contrast on the parameter estimates in Fig 4b involving spatial distance and transition frequencies. Note that in both cases parameter estimates were extracted from an ROI constructed on an orthogonal contrast (see comment on question 2 below).

In other words, we ran an ANOVA on the four bars on the left hand-side (2 x 2) and one on the middle/right set of bars (2 x 2 x 2) of the below plot.

The rationale for running two rather than one combined ANOVA is that the parameter estimates come from two different GLMs, so the arbitrary unit of 'parameter estimate' may not be comparable across contrasts.

The 2 x 2 Anova for ‘correct sequence order’ shows a significant effect of Network $F(1,25)=9.893$, $p=0.004$ but no effect of Node or Network x Node (both $p>0.2$). Consistently, the winning model in a Bayesian repeated-measures ANOVA entails only the factor Network: $P(M|data)=0.647$, $BF_M=7.343$.

Within Subjects Effects

	Sum of Squares	df	Mean Square	F	p
Network	545137.411	1	545137.411	9.893	0.004
Residual	1.378e +6	25	55102.115		
Node	882.827	1	882.827	0.023	0.881
Residual	957830.570	25	38313.223		
Network * Node	150873.215	1	150873.215	1.318	0.262
Residual	2.862e +6	25	114461.560		

Note. Type III Sum of Squares

Bayesian Repeated Measures ANOVA ▾

Model Comparison

Models	P(M)	P(M data)	BF_M	BF_{10}	error %
Network	0.200	0.647	7.343	1.000	
Network + Node	0.200	0.134	0.618	0.207	1.616
Null model (incl. subject)	0.200	0.106	0.472	0.163	0.772
Network + Node + Network * Node	0.200	0.091	0.402	0.141	1.879
Node	0.200	0.022	0.090	0.034	1.763

Note. All models include subject.

The 2 x 2 x 2 ANOVA for both the effects of spatial distance and transition frequency shows a significant effect of Network ($F(1,25)=30.82$, $p<0.001$) and no other significant main effects or interactions, only a trend for an effect of Contrast x Network $p=0.074$. In the Bayesian analysis, while there isn't one clear winning model and three models have quite similar performance, importantly all of these equally good models include the factor network. More specifically, the three similarly good models are a model with only a factor Network $P(M|data)=0.195$, $BF_M=4.37$, one with both effects of Contrast + Network $P(M|data)=0.256$, $BF_M=6.19$ and a final model with effects of Contrast + Network + Network x Contrast ($P(M|data)=0.251$, $BF_M=6.02$). Thus, in all cases, there is evidence for a difference between the pOFC-tempPole and the mPFC-amyg/hippo networks using frequentist and Bayesian statistics.

Repeated Measures ANOVA ▾

Within Subjects Effects

	Sum of Squares	df	Mean Square	F	p
contrast	202797.399	1	202797.399	1.550	0.225
Residual	3.272e +6	25	130876.390		
Network	982445.109	1	982445.109	30.820	< .001
Residual	796924.552	25	31876.982		
Node	60536.509	1	60536.509	2.837	0.105
Residual	533418.592	25	21336.744		
contrast * Network	113434.820	1	113434.820	3.472	0.074
Residual	816880.675	25	32675.227		
contrast * Node	1.395	1	1.395	6.954e -5	0.993
Residual	501394.815	25	20055.793		
Network * Node	77479.478	1	77479.478	2.020	0.168
Residual	958815.397	25	38352.616		
contrast * Network * Node	3750.339	1	3750.339	0.110	0.743
Residual	854045.285	25	34161.811		

Note. Type III Sum of Squares

Bayesian Repeated Measures ANOVA ▾

Model Comparison ▾

Models	P(M)	P(M data)	BF _M	BF ₁₀	error %
Contrast + Network	0.053	0.256	6.188	1.000	
Contrast + Network + Contrast * Network	0.053	0.251	6.017	0.979	32.098
Network	0.053	0.195	4.374	0.764	2.337
Contrast + Network + node	0.053	0.074	1.437	0.289	3.860
Network + node	0.053	0.055	1.054	0.216	4.343
Contrast + Network + Contrast * Network + node	0.053	0.047	0.896	0.185	4.001
Contrast + Network + node + Network * node	0.053	0.035	0.660	0.138	3.893
Network + node + Network * node	0.053	0.025	0.455	0.096	2.899
Contrast + Network + Contrast * Network + node + Network * node	0.053	0.022	0.409	0.087	4.359
Contrast + Network + node + Contrast * node	0.053	0.016	0.300	0.064	4.860
Contrast + Network + Contrast * Network + node + Contrast * node	0.053	0.010	0.183	0.039	5.389
Contrast + Network + node + Contrast * node + Network * node	0.053	0.006	0.117	0.025	3.442
Contrast + Network + Contrast * Network + node + Contrast * node + Network * node	0.053	0.005	0.088	0.019	5.620
Contrast + Network + Contrast * Network + node + Contrast * node + Network * node + Contrast * Network * node	0.053	0.001	0.024	0.005	6.737
Contrast	0.053	6.607e -5	0.001	2.583e -4	2.309
Null model (incl. subject)	0.053	6.257e -5	0.001	2.446e -4	1.911
Contrast + node	0.053	1.728e -5	3.111e -4	6.756e -5	2.385
node	0.053	1.614e -5	2.906e -4	6.311e -5	2.030
Contrast + node + Contrast * node	0.053	3.667e -6	6.601e -5	1.434e -5	6.868

Note. All models include subject.

We also ran additional ANOVAs without any definitions of a priori 'Networks':

- (3) A 4x1 Anova with factor ROI based on the parameter estimates in Fig 3b for correct sequence order
- (4) A 4x2 Anova with factors ROI x contrast on the parameter estimates in Fig 4b involving spatial distance and transition frequencies.

The 4 x 1 Anova for 'correct sequence order' shows a significant effect of ROI $F(3,75)=3.352$, $p=0.023$ (winning model in Bayesian repeated-measures ANOVA: factor ROI, $P(M|data)=0.67$; $BF_m=2.07$).

Within Subjects Effects ▼

	Sum of Squares	df	Mean Square	F	p
roi_inCorrOrd	696893.452	3	232297.817	3.352	0.023
Residual	5.197e +6	75	69292.299		

Note. Type III Sum of Squares

Model Comparison ▼

Models	P(M)	P(M data)	BF _M	BF ₁₀	error %
roi	0.500	0.674	2.068	1.000	
Null model (incl. subject)	0.500	0.326	0.484	0.484	0.465

Note. All models include subject.

The 4 x 2 Anova for both spatial distance and transition frequency shows a significant effect of ROI (but not contrast or ROI x contrast): $F(3,75)=12.237$, $p<0.001$. In the equivalent Bayesian repeated-measures ANOVA, the winning model had factors Contrast and ROI effects: $P(M|data)=0.519$; $BF_M=4.318$.

Within Subjects Effects ▼

	Sum of Squares	df	Mean Square	F	p
Con	202797.399	1	202797.399	1.550	0.225
Residual	3.272e +6	25	130876.390		
roi	1.120e +6	3	373487.032	12.237	< .001
Residual	2.289e +6	75	30522.114		
Con * roi	117186.553	3	39062.184	1.349	0.265
Residual	2.172e +6	75	28964.277		

Note. Type III Sum of Squares

Model Comparison

Models	P(M)	P(M data)	BF _M	BF ₁₀	error %
Con + roi	0.200	0.519	4.318	1.000	
roi	0.200	0.406	2.736	0.783	2.239
Con + roi + Con * roi	0.200	0.074	0.318	0.142	2.357
Con	0.200	5.829e -4	0.002	0.001	2.205
Null model (incl. subject)	0.200	5.551e -4	0.002	0.001	1.389

Note. All models include subject.

To test whether the ROI effects in both cases were indeed driven by a difference between the two proposed networks, we compared the mean of pOFC/tempPole with the mean of mPFC and amyg/hippo (i.e. the average of the red and blue bars in the above figure, respectively).

All t-tests came out significant but pointed in opposite directions as expected for CorrOrder versus SpatDist/TransFreq:

- CorrOrder: $t(25)=3.15$, $p=0.0041$
- SpatDist: $t(25)=-5.5588$, $p=8.85e-06$
- TransFreq: $t(25)=-2.44$, $p=0.022$

This supports the claim that there are two networks with distinct contributions.

The reviewer also mentions Figure 2c. We have now tried to clarify in the text that this was a test we thought should be significant in all regions if we are tapping into repetition suppression effects in the initial contrast used for defining our ROIs (rr-cc). We knew it was a tough test because we are asking for a modulation of repetition suppression effects by temporal distance between successive stimuli (in fact, we are not aware this has been successfully reported within the fMRI community). But it was an effect that would be expected based on animal work where suppression after repeated exposure scales with the distance between the two repetitions. This result, unlike the results that we have focused on in the reply up to this point, however, is not part of our argument for distinct networks. We think all four regions encode some knowledge about elements belonging to RS vs CS.

We have now tried to clarify these various points in the text as follows:

Pages 7-8:

While the (rr-cc) contrast should reflect participants' knowledge of associations between rewarded sequence elements **consistent with cross-stimulus suppression effects**, it may also be driven by a main difference in BOLD activity between **rewarded and control sequence** elements. To confirm our interpretation **as cross-stimulus suppression effects**, we examined the impact of the temporal delay between successive sequence elements. Neural adaptation effects are expected to scale with the temporal delay between stimuli, with stronger suppression for stimuli presented closer in time. We therefore modelled the temporal delay between occurrences of **rewarded sequence** elements and the temporal delay between occurrences of **control sequence** elements parametrically. We extracted parameter estimates from regions of interest (ROIs) defined as spheres around peak coordinates in the above contrast (see Methods and **Suppl Fig 2c**). **Even though this is a very demanding test that is rarely performed in repetition suppression experiments we found that BOLD activity for the second of two successive elements of RewSeq (rr) but not ConSeq (cc) indeed scaled with the temporal distance between the stimuli in two of our four ROIs: temporal pole and amyg/hippo (temporal distance rr: p(mPFC)=0.83; p(pOFC)=0.79; p(tempPole)=0.049, t(25)=1.72; p(amyg/hippo)=0.008, t(25)=2.54; temporal distance cc: all p>0.1; Figs 2c,d and Suppl Fig 2a)**. This bolstered our interpretation of a shared neural representation of rewarded sequence elements in these regions.

Pages 13-14:

Dissociating the two functional networks: tempPole-pOFC versus hippo/amyg-mPFC

Above we showed that BOLD responses in temporal pole and pOFC reflected the correctly ordered rewarded sequence, while mPFC and amyg/hippo carried knowledge of statistical relationships between all stimuli. To formally assess whether these two networks indeed carry different information, we ran additional analyses relating to the contrasts that differentiated between these areas: correct order and spatial/statistical transition. The first 2 x 2 ANOVA focused on the parameter estimates extracted from the correct order contrast in all four ROIs (Figure 3b). It included the factors Network (tempPole & pOFC

versus amyg/hippo and mPFC) and Node (anterior/posterior, where pOFC and mPFC are the anterior nodes of the two networks both found within frontal cortex and tempPole and amyg/hippo are posterior nodes in the temporal lobe, respectively). There was a significant effect of Network ($F(1,25)=9.893$, $p=0.004$) but no effect of Node or Network x Node (both $p>0.2$). Moreover, consistent with this result, a Bayesian repeated-measures ANOVA showed that the winning model indicated that only the factor Network significantly explained variation in activity between the regions ($P(M|data)=0.65$, $BFm=7.34$).

The second 2 x 2 x 2 ANOVA comprised the same conditions and the additional factor Contrast (spatial distance or transition frequency). This ANOVA focused on BOLD responses related to statistical knowledge (Figure 4b). Again, we found a significant effect of Network ($F(1,25)=30.82$, $p<0.001$) and no other significant main effects or interactions, only a trend for a Contrast x Network interaction ($F(1,25)=3.472$, $p=0.074$). The conclusions drawn from this analysis were bolstered by a Bayesian repeated-measures ANOVA that revealed three similarly good models of the data, all of which included a factor of Network (Model 1 had a main effect of Network only: $P(M|data)=0.195$, $BFm=4.37$; Model 2 included two main effects of Contrast and Network: $P(M|data)=0.256$, $BFm=6.19$; Model 3 contained three effects Contrast + Network + Network x Contrast: $P(M|data)=0.251$, $BFm=6.02$). In summary, in all cases, there was evidence for a difference between the activity patterns in the pOFC-tempPole and mPFC-amyg/hippo networks.

Part (b)

There were several reasons, given our prior knowledge of the brain's organization, for thinking that the two functional networks composed of pOFC and tempPole on the one hand and hippo/amyg and mPFC on the other hand, are anatomically plausible networks:

(i) There are strong monosynaptic connections within but not across these two networks (Kondo/Price, 2005; Carmichael Price 1995; Saleem/Price 2008)

(ii) specifically, temporal pole and pOFC clusters, despite being in different lobes, are spatially very close in terms of a major connection pathway in the brain – the uncinate fascicle (Schmamann and Pandya, 2006; Croxson et al., 2005) roughly at the position of the green crosshair in the below figure) and which is very close to the location of both temporal pole and pOFC activations (here shown for the initial ROI-defining contrast 'rr-cc' at a lenient threshold of uncorrected $z>2.3$ to illustrate the clusters' close spatial proximity). Similarly, mPFC and hippocampus are connected via another major white matter bundle – the fornix (Croxson et al., 2005; Aggleton et al., 2015).

(iii) Resting state data based on the average of 200 people from the Human Connectome Project (HCP) highlights coupling within each network, but not across networks at rest. For example, when we look at resting state connectivity of **mPFC** (little white dot in the left panel below) – we see that activity is strongly coupled with that in amyg/hippo, PCC and a few other regions, but no strong coupling to temporal pole or pOFC. Hotter colors indicate increased resting-state coupling.

When we look at resting state connectivity of **pOFC**, there is still some mPFC and amyg/hippo connectivity but particularly strong resting-state connectivity with temporal pole.

Thus, we can show that differences in the BOLD signatures observed in temporal pole-pOFC versus mPFC-amyg/hippo are clear in both network-agnostic ANOVAs and ANOVAs that take these anatomical priors into account. We hope this has convinced the reviewer that these effects are robust.

We have included several new sections and a new figure (Suppl Fig 5) in the manuscript describing these set of results (see above and below). Thank you again for spotting our oversight.

Pages 13-14:

The identification of different response patterns in the two networks is consistent with a large body of work suggesting that there are major anatomical differences between pOFC and tempPole on the one hand and hippo/amyg and mPFC on the other hand. There are strong monosynaptic connections within but not across these two networks³⁵⁻³⁷. For example, temporal pole and pOFC clusters, despite being in different lobes, are connected via the uncinate fascicle^{38,39} while the hippocampus and mPFC are interconnected via the fornix^{38,40}. These network connections are not just clear in tracer data but can also be appreciated using human resting-state data; there is strong within-network activity coupling (between pOFC and tempPole and between hippo/amyg and mPFC) but weaker across-network coupling (**Suppl Fig 5**; source: Human Connectome Project (HCP) Data). Altogether, this provides robust evidence that the patterns of BOLD activation in the pOFC-temporal pole and amyg/hippo-mPFC were dissociable, with pOFC and temporal pole reflecting knowledge of the correctly ordered rewarded sequence and amyg/hippo and mPFC showing signatures of statistical learning. As previously noted, both types of learning, statistical and reward-learning, mediated aspects of stimulus-stimulus learning in our task. The

dissociations between regions are less to do with the type of association but the mechanism of learning by which the association was derived.

2. As I understand it, the analyses are meant to distinguish BOLD activity sensitive to two broad sorts of relationships in the task: those concerning the relationship of stimuli to reward, and those concerning the relationship of stimuli to one another. But the set of analyses presented are overlapping and mutually inconsistent in a way that makes it hard to really nail down what is ultimately going on.

For instance, most of the analyses are based on peaks of an initial “rr vs cc” contrast that seeks areas that show cross-stimulus-suppression, moreso for stimuli involved in the rewarded than the unrewarded sequence but not otherwise distinguishing among different stimulus relationships. Followup tests then detect other effects in these areas that complicate this picture.

For instance, the tests in Figure 4 seek sensitivity to spatial distance and transition frequency, both defined regardless of reward. But this is different from the initial ROI-defining contrast in several respects. It is reward-independent. It is directed (i.e. it distinguishes A->B transitions from B->A). And it seems to be characterized by greater activation to stimuli when they are more expected, the opposite of what is implied by cross stimulus suppression. Which is right?

That the descriptions of activity implied by these various contrasts are mutually inconsistent would seem to imply that the ultimate activity may be something else than straightforwardly implied by either regressor. For instance, it might be possible that the significant cross stimulus suppression effect in the first instance is actually driven by a subset of *directed* transitions, such that rr > cc on average but there is only suppression for pairs in which the first r comes before the second r in the sequence. (This might happen if directed predictive activity at each step drives cross-stimulus suppression, rather than representational change making all stimuli in a group more similar to one another.)

Conversely, why does it make sense to look for reward-agnostic distance or conditional frequency effects in areas already screened for reward sensitivity? And if you find it, is it really reward-agnostic? It seems at least possible that in this case the distance and conditional frequency sensitivity are actually driven by such effects only within the rewarded sequence?

Finally, what are we to make of the superposition of frequency related facilitation and cross-stimulus suppression. For an A->B transition within the rewarded sequence, there is less suppression than a B->A one? Or what?

Figure 3 raises analogous questions of how to reconcile the CSS findings with the various other ones. Perhaps it would be helpful to pursue an increasingly refined CSS view (i.e. is there more or less cross stimulus suppression for stimuli in reward sequences, spatially distinct, etc), breaking down the rr vs cc effects, more in the spirit of 2c,d.

Thanks – we apologize for the confusing presentation of the different associations. Because there are several points in this comment, we will reply in multiple parts:

(1) First, to answer the reviewer's general point

(a) We will argue that stim-stim and stim-reward learning form interleaved computations because RL learning is only initially about the formation of stim-reward associations after which reward propagates back across stimuli (stim-stim); also, incidental learning and reward learning occur simultaneously (see Tolman).

(b) We provide additional support to show that participants' behavior in this task demonstrates both types of learning (by reward and incidental/statistical observations) because RT speeding effects are present even in the control sequence (new analysis), while they are stronger in the rewarded sequence (Fig 1c).

(c) We explain why as a consequence, we focus subsequent MRI analyses on regions that have non-directed knowledge of the rewarded sequence. We do this by first defining a broad contrast (rr-cc) that identifies areas with privileged knowledge of the previously rewarded sequence. Among those diverse regions, we then go on to examine which part of the network retains precise knowledge of the temporal sequence of rewarded elements (pOFC, tempPole) and which part computes newly observed transitions between stimuli and is hence able to integrate incidental learning with reward memory (mPFC, amyg/hippo)

The reviewer raises the grave concern that the two analyses approaches we present and use to identify the two brain networks are overlapping and mutually inconsistent and that there might hence be an additional process underlying both sets of results. In subsequent points **(2) and (3)** below we demonstrate in detail that these contrasts are not mutually exclusive. This is a very important point, because it means that they provide independent pieces of evidence for distinct computations performed in mPFC/amyhippo and pOFC/tempPole.

(1a) Conceptually, we agree that many of our analyses can be linked to stim-reward and stim-stim associations. Yet these two types of knowledge do not map onto the outputs of RL and statistical learning in a simple way.

In a majority of reinforcement learning tasks, RL learning is not about stim-stim associations, so it seems tempting to attribute stim-stim learning to statistical learning. However, consider a situation in which a sequence of elements predicts reward, a common phenomenon in real-life. Knowledge of the relationships between the predictive stimuli, and thus stim-stim associations, is something that can be derived by RL; indeed it is a key focus of many classic RL treatments (e.g. Sutton and Barto; 1998) just as it is in many animal behaviour studies of reinforcement learning that consider higher order conditioning chains for many decades. In other words, rather than learning only stim-reward associations such as A-rew, B-rew, C-rew, D-rew, RL can mediate learning of A-B, B-C, and C-D which are stim-stim associations and these are particularly important when the order of transitioning through the stimuli matters to reach reward. So in our task stim-stim associations are not purely incidental knowledge, they are part of the reward learning. We show that, if they lead to reward, such stim-stim associations are learned faster than one would expect from purely statistical learning. Or, in other words, propagating the reward backwards across those stim-stim relationships speeds up learning.

Finally, note that Tolman already showed in 1937 in his classic maze experiments, that reward-learning and processes akin to those that we refer to as statistical learning, occur simultaneously. Statistical learning cannot be 'switched' off, we always learn implicitly at all times from the associations (or

transitions) we observe in the world, whether or not they lead to reward. We have tried to clarify this in our revised manuscript as follows:

Page 3: In temporal difference (TD) RL, over the run of many trials, rewards are associated with previous states and previous actions. This enables the back-propagation of value for example during classic conditioning¹ but also during spatial navigation^{22,23}. **It implies that RL-acquired knowledge can comprise both stimulus-reward and stimulus-stimulus associations.**

Page 4: On separate days prior to scanning, participants (n=26) performed a behavioral task where they learnt to associate a four-step sequence of stimuli (ABCD) with reward (rewarded sequence: **RewSeq**). Unlike in previous work assessing sequence learning^{e.g., 29–31}, however, the sequence was not learnt from its starting element. Instead, participants pressed buttons to move around a 3x4 grid towards stimuli that were highlighted by the computer one after the other in a continuous stream (**Fig 1a, Suppl Fig 1a-c**). Importantly, the start of the rewarded sequence was not signaled but occurred after an unpredictable number of other stimuli. When D was reached, a reward appeared, and participants had to infer the rewarded sequence and determine its length through careful and repeated observation. Unusually, this allowed us to test one of the key predictions of RL – that associative structures between representations of stimuli and reward and between different stimuli are constructed by propagating reinforcement from the stimuli that occurred immediately prior to reward to earlier stimuli²³. **As a consequence, participants should know the end of the sequence before its beginning, and associations learnt via RL comprise both stimulus-stimulus (e.g., A to B) as well as stimulus-reward associations (e.g., D to Reward).** Unbeknownst to participants, they also transitioned through another four-step sequence (A'B'C'D') equally frequently but this sequence was not rewarded (control sequence: **ConSeq**). This procedure allowed us to assess (i) knowledge of the stimulus route leading to reward (i.e., the RL-acquired knowledge) while controlling for stimulus-stimulus learning that is merely driven by statistical learning mechanisms, (ii) the existence of incidentally learned statistical knowledge of stimulus transitions which should be present for both **RewSeq** and **ConSeq**, and (iii) the possibility of spread of reward effects across space. Note that both types of learning, statistical and reward-learning, could influence formation of stimulus-stimulus associations in our task and that the dissociations between neural structures reported here are less to do with the type of association but the mechanism of learning by which such associations were derived.

Page 19: *Contrasting types of knowledge representation*

It is important to note that reward-learning and processes akin to those that we refer to as statistical learning can proceed simultaneously²⁵ and that the associative structures that form as a result of RL and

incidental learning of task statistics do not simply map onto stimulus-reward association and stimulus-stimulus association learning respectively. Knowledge of the relationships between reward-predictive stimuli, and thus stimulus-stimulus associations, is something that can be derived by RL; indeed it is a key focus of many classic RL treatments²³ just as it is in many animal behavior studies of reinforcement learning that have considered higher order conditioning chains for many decades. In other words, rather than learning only stimulus-reward associations such as A-reward, B-reward, C-reward, D-reward, RL can mediate learning of A-B, B-C, and C-D which are stimulus-stimulus associations and these are particularly important when the order of transitioning through the stimuli matters in order to reach reward. In summary knowledge of stimulus-stimulus associations is not purely derived by incidental learning but occurs as a result of reward learning. Indeed we show that, when they lead to reward, such stimulus-stimulus associations are learned faster than one would expect from purely statistical learning; propagating the reward backwards across those stimulus-stimulus relationships speeds up learning.

(1b) Evidence for simultaneous learning from statistical and reinforcement learning is present in our behavioural data. In Fig 1c also part of the original manuscript and shown again below, we plot reaction times (RTs) for the initiation of a trajectory. This analysis thus probes purely statistical learning in the

case of the control sequence, but it probes the summation of both learning processes (reward learning and statistical learning) for the rewarded sequence. But in both cases the knowledge we are testing is stim-stim knowledge or in other words whether participants anticipate transitions from one stimulus to the next. Not unexpectedly, there is a speeding from AB to BC to CD for both the control sequence (grey) and the rewarded sequence (blue). We originally only reported this main effect of 'transition' for both sequences together (in addition to the interaction of sequence x transition). However, we did not report the critical analysis that suggests that stim-

stim learning occurred in the control sequence (i.e. in the absence of reward and thus purely driven by statistical learning). The key analysis is an ANOVA solely run on the control sequence transitions for Fig 1c (i.e. the grey bars). It shows a significant effect of transition: $F(2,50)=28.553$, $p<0.001$; model with factor transition: $BFm=1.522e+6$, $P(M|data)=1$. We would expect the slope of the rewarded sequence to be as steep as the one for the control sequence if learning was purely driven by statistical observations. The additional speeding up in the rewarded sequence must be due to reward boosting learning.

Within Subjects Effects ▼					
	Sum of Squares	df	Mean Square	F	p
Transition	0.033	2	0.016	28.553	< .001
Residual	0.029	50	5.711e -4		

Note. Type III Sum of Squares

Model Comparison					
Models	P(M)	P(M data)	BF _M	BF ₁₀	error %
Transition	0.500	1.000	1.522e +6	1.000	
Null model (incl. subject)	0.500	6.571e -7	6.571e -7	6.571e -7	1.026

Note. All models include subject.

In the revised manuscript we have clarified this result as follows (page 6):

Behavioral evidence for statistical learning

We next probed whether participants showed evidence for incidental learning which should be unrelated to reward and driven by repeated observation of statistical relationships, such as the transitions between control sequence stimuli. We returned to the RTs for the first movement that initiates transitioning between two stimuli of the control sequence during the pre-scan learning task (grey bars in **Fig 1c**, also part of the larger ANOVA above). We asked whether initiating the button press from A' to B', followed by B' to C' and C' to D' would show RT speeding because transitions, despite never being associated with reward, became increasingly predictable. Note that these transitions were experienced as frequently as those in the rewarded sequence. A 1x3 repeated-measures ANOVA with factor Transition (A'B'/B'C'/C'D') run only on the control sequence transitions showed a significant effect of transition ($F(2,50)=28.553, p<0.001$; Bayesian repeated-measures ANOVA shows evidence for a model with factor Transition: $BF_M=1.522e+6, P(M|data)=1$). Thus RTs became faster as participants progressed through the control sequence, but as shown above, this speeding was less pronounced than in the rewarded sequence.

(1c) Of course a valid question, conceptually and with respect to the neural signals, is whether it is reasonable to identify our regions of interest using this initial rr-cc contrast and whether mPFC and hippo/amyg should encode something about the rewarded structure as probed by this contrast. Our ROI definition based on the rr-cc contrast does indeed focus our paper solely on regions that have *some*

reward-related knowledge. This was done on purpose. We tried to design a broad/inclusive contrast that is, as stated by this reviewer, non-directional and merely probes knowledge of the existence of relationships between RS elements compared to CS elements.

We have now realized that in our manuscript we focused mostly on mPFC and amyg/hippo's encoding of transition frequency and spatial distance, not pointing out in the discussion that, overall, given the initial rr-cc contrast, what this means is that these two regions code a *combination* of knowledge acquired using RL and statistical learning. In the revised manuscript we clarify that this means that pure statistical learning is enhanced by reward and that knowledge from two separate learning mechanisms converges in mPFC and amyg/hippo.

Given the reviewer's interest in this point we also now in the revised manuscript discuss an additional analysis to rule out an alternative possibility that mPFC and amyg/hippo flexibly code the task-relevant knowledge, and that the rr-cc contrast probes a representation formed during the pre-scan learning task, but that this representation diminishes over the duration of scanning when distinguishing between rewarded and control sequences is no longer relevant for behaviour. To follow this up, we tested whether rr-cc changed over the six scan blocks in mPFC or amyg/hippo and this was not the case ($p=0.56$ for mPFC; $p=0.38$ for amyg/hippo; see figure below). This confirms that even though these structures hold flexible statistical knowledge, the reward-driven associations they also hold are persistent and inflexible even in these regions:

Taken together, in light of the reviewer's comments, we have now tried to clarify, in several places in the manuscript, that both types of learning, statistical and reward-learning, can mediate aspects of stimulus learning and that the dissociations we are reporting in terms of the neural structures are less to do with the type of association but with the mechanism of learning by which the association was derived (reward-driven versus driven by observation of statistical contingencies). We hope this will help to clarify the general point made by the reviewer above and we apologize again that this was confusing in the original manuscript. Relevant sections of the revised manuscript include the following:

Pages 6-7

To test for differences in BOLD activity between rewarded and control sequence elements, we contrasted **reward-sequence** elements that were preceded by another **reward-sequence** element (i.e., 'rr' pairs) with **control-sequence** elements that were preceded by another **control-sequence** element (i.e., 'cc' pairs). This contrast captures effects of cross-stimulus adaptation to elements from the same sequence **and is intentionally broad and non-directional in the sense that it merely probes knowledge of**

the existence of undirected relationships between rewarded sequence elements compared to control sequence elements. In other words, sequence elements included in this contrast were not always shown in the correct order but sometimes backwards or with jumps in-between sequence elements (e.g. CB or BD, see Methods). The identified brain regions therefore encode associations between rewarded sequence elements such that elements part of the rewarded sequence have become more similar in their neural representation than the corresponding elements of the control sequence ('sequence fusion'; Fig 2a).

Pages 7-8

While the (rr-cc) contrast should reflect participants' knowledge of associations between rewarded sequence elements consistent with cross-stimulus suppression effects, it may also be driven by a main difference in BOLD activity between rewarded and control sequence elements. To confirm our interpretation as cross-stimulus suppression effects, we examined the impact of the temporal delay between successive sequence elements. Neural adaptation effects are expected to scale with the temporal delay between stimuli, with stronger suppression for stimuli presented closer in time. We therefore modelled the temporal delay between occurrences of rewarded sequence elements and the temporal delay between occurrences of control sequence elements parametrically. We extracted parameter estimates from regions of interest (ROIs) defined as spheres around peak coordinates in the above contrast (see Methods and Suppl Fig 2c). Even though this is a very demanding test that is rarely performed in repetition suppression experiments we found that BOLD activity for the second of two successive elements of RewSeq (rr) but not ConSeq (cc) indeed scaled with the temporal distance between the stimuli in two of our four ROIs: temporal pole and amyg/hippo (temporal distance rr: $p(\text{mPFC})=0.83$; $p(\text{pOFC})=0.79$; $p(\text{tempPole})=0.049$, $t(25)=1.72$; $p(\text{amyg/hippo})=0.008$, $t(25)=2.54$; temporal distance cc: all $p>0.1$; Figs 2c,d and Suppl Fig 2a). This bolstered our interpretation of a shared neural representation of rewarded sequence elements in these regions. In a second test, we confirmed that none of our ROIs were simply showing a main effect difference between rewarded and control elements. We examined rewarded and control elements when they were preceded by an element not part of their own sequence (xr or xc). None of our ROIs showed a significant difference for the contrast xr-xc (all $p>0.1$; Suppl Fig 2b). Thus, our ROI-defining contrast captured relationships between pairs of stimuli as probed by cross-stimulus suppression, rather than differences in the overall BOLD main activation to rewarded and control stimuli.

Pages 12-13

To complete the picture, we tested whether statistically learnt knowledge in mPFC and amyg/hippo was, over blocks of scanning, replacing the undirected knowledge of the rewarded sequence, as probed by our initial ROI-defining contrast (sequence fusion: 'rr-cc'). If so, we would expect a decline in the sequence fusion contrast over blocks of scanning. We did not observe such a change in either region (mPFC: $p=0.56$; amyg/hippo: $p=0.38$; nor in tempPole: $p=0.6$ or pOFC: $p=0.41$), suggesting that even though mPFC and amyg/hippo hold flexible statistical knowledge, the reward-driven associations they also hold were persistent and inflexible. We note that in our hands, BOLD modulations related to rewarded knowledge (rr-cc) were consistent with cross-stimulus suppression effects and were thus smaller for stronger associations. By contrast, BOLD modulations related to representation of statistical knowledge were stronger for more expected transitions. This suggests that these relationships are not stored via shared neural representations because this would have led to a suppressed BOLD signal to the second element.

Taken together, knowledge from two separate learning mechanisms converged in amyg/hippo and mPFC. Both regions carried undirected knowledge of the rewarded sequence, as probed by the initial ROI-defining contrast, but they also represented reward-independent associations which were true across all stimuli: They responded more strongly to likely transitions both in space (closer) as well as statistical contingencies (more frequent) in a way that suggests full knowledge of task relationships, and thus an encoding of associations acquired through incidental learning mechanisms. In contrast to the representations found in temporal pole and pOFC, neither of these two *statistical* representations (*spatial distance or transition frequency*) is linked to the *directed* reward associations learned in the pre-scan task. Moreover, over time, the type of coding for consecutive grid elements in mPFC transitioned from a spatial towards a frequency coding scheme.

Pages 19-20

Contrasting types of knowledge representation

It is important to note that reward-learning and processes akin to those that we refer to as statistical learning can proceed simultaneously (Tolman, 1937) and that the associative structures that form as a result of RL and incidental learning of task statistics do not simply map onto stimulus-reward association and stimulus-stimulus association learning respectively. Knowledge of the relationships between reward-predictive stimuli, and thus stimulus-stimulus associations, is something that can be derived by

RL; indeed it is a key focus of many classic RL treatments (e.g. Sutton and Barto; 1998) just as it is in many animal behavior studies of reinforcement learning that have considered higher order conditioning chains for many decades. In other words, rather than learning only stimulus-reward associations such as A-reward, B-reward, C-reward, D-reward, RL can mediate learning of A-B, B-C, and C-D which are stimulus-stimulus associations and these are particularly important when the order of transitioning through the stimuli matters in order to reach reward. In summary knowledge stimulus-stimulus associations are not purely derived by incidental learning but as a result of reward learning. Indeed we show that, when they lead to reward, such stimulus-stimulus associations are learned faster than one would expect from purely statistical learning; propagating the reward backwards across those stim-stim relationships speeds up learning.

Intriguingly, activity in mPFC and amy/hippo reflected several statistically dissociable aspects of the task at the same time. On the one hand its activity reflected statistical learning – knowledge of transition frequencies between task states regardless of reinforcement. On the other hand, its activity reflected associations between rewarded sequence elements, regardless of whether elements occurred in the correct or incorrect order, more than it did control sequence elements; in contrast to the first type of representation this second type does depend on reinforcement. While the knowledge acquired from statistical learning was flexibly updated and constantly changing in mPFC, as in other areas, reinforcement-based task representations in both amyg/hippo and mPFC were robust and inflexible and did not change over time, even though this knowledge was irrelevant to the task performed during scanning. In summary, RL and statistical learning mechanisms converge in mPFC but have different temporal properties.

(2) In the next part of our answer, we want to clarify in more **detail the relationships between the initial contrast and subsequent contrasts** shown in Fig 4 (transition frequency and spatial distance).

(2a) Most importantly, we can show statistically and explain in a hopefully intuitive way why they are **independent effects** but not mutually contradictory effects as we will show below.

For visualization, in the full transition space (shown in Fig S4 for the true underlying pre-scan and scan transitions between all 12 stimuli and reward) the rr-cc contrast probes whether the dark brown transitions within the rewarded sequence are stronger (more suppression) than the lighter brown transitions within the control sequence. We have left the diagonal empty as repetitions of the same element don't occur.

By contrast, in the transition frequency contrast in Figure 4 that the reviewer is referring to, we are asking for a modulation according to the stimulus-stimulus transitions that are encountered during the course of the fMRI sessions and the representations of which are updated on a trial-by-trial basis. Visually, this regressor basically moves slowly from the pre-scan transitions (shown on the left below) to the transitions experienced by the end of scanning (shown on the right below; also shown in Figure S4).

Testing for possible correlations between the 'patterns' gives a first hint that they are not related. The correlations between the above rr-cc contrast pattern and the following two transition frequency patterns are $r=0.0068$ and $r=-0.0178$ and thus nearly zero. The two effects were originally extracted from different GLMs (for which the correlations were shown in the original FigS2 pasted here for the reviewer's reference).

To test the correlations more precisely, we combined the two original GLMs which shows the following relationships between *all* regressors from both GLMs combined (note that correlations are shown for illustration here; this model was not estimated in this form, e.g. flower and treasure regressors were identical in both GLMs and are thus 100% correlated):

This shows that rr or cc (cc is labelled rr_CS here) regressors do not correlate with transFreq. To illustrate this more simply, below we extracted the correlations between just the relevant regressors: the rr-cc contrast and the transition frequency as well as spatial distance regressors. Both regressors were orthogonal to the rr-cc contrast ($\rho = -0.01$ for transFreq and rr-cc and $\rho = -0.02$ for SpatDist and rr-cc) which suggests that these signals can simultaneously be present within the same region. In light of

the reviewer's comments we have now included plots like the above for clarification and visualization of the various contrasts tested in the manuscript and report the correlations between them.

Page 10:

We first asked whether information about spatial distances between elements (regardless of whether they comprised **rewarded or control sequence elements**), measured in terms of steps that make up a path on the grid, **was present** in our previously defined ROIs (GLM2, **Suppl Fig 2d; Fig 4a**). This 'spatial distance' contrast was not correlated with the original ROI-defining contrast rr-cc (Pearson's $r=-0.02$; see also **Suppl Fig 2f**).

Page 11:

We tested, using a model with trial-by-trial updates (see Methods), whether **BOLD responses scaled with** the frequency of an experienced transition from the previous to the current stimulus in any of the previously defined ROIs. **Again, there was no correlation between this transition frequency contrast and our original ROI-defining contrast ($r=-0.01$; Suppl Figs 2f and 4c).**

(2b) As a further demonstration of the independence of reward-agnostic effects (i.e. the spatial distance and the transition frequency effects) from the reward-related effects we conducted the following control analysis. We extended the original GLM and modelled onsets of RS and CS elements separately, each with their own distance and frequency parametric regressors. In other words, we duplicated the main onset and parametric regressors (regressors 1-6 in the original GLM2 pasted below), specifying them separately for stimuli from the rewarded and control sequence.

The below summarizes, for both mPFC and amygd/hippo, the effects of spatial distance and transition frequency (originally reported in Fig 4) separately for the rewarded and control sequence (RS=blue vs CS=grey). There was no difference between either parametric regressor fitted to RS and CS trials for any region (all $p > 0.15$). For mPFC, transition frequency did not reach significance when fitted on RS trials alone ($p = 0.18$) and for amygd/hippo, transition frequency did not reach significance for CS trials alone ($p = 0.15$). However, given that we used a third of the data in each of these analyses compared to the original transition frequency analysis which was performed across all stimuli, we thought it was remarkable that in both mPFC and amygd/hippo three out of four effects reached significance (all $p < 0.05$), with the remaining effects spelled out above pointing in the same direction. Taken together, we believe this is strong evidence that effects were not driven by the RS trials only.

We have included these additional results in the revised manuscript as follows (page 11):

We confirmed in a control analysis that spatial distance and transition frequency responses were not driven by the subset of stimuli of the rewarded sequence. Splitting stimulus onsets into those of the rewarded and control sequences, each with their own spatial distance and transition frequency parametric regressor, showed that there was no difference between rewarded and control trials in either mPFC or amyg/hippo in terms of BOLD modulations by spatial distance and transition frequency (all $p > 0.15$; **Supplementary Results section Statistical learning of spatial distance and transition frequency is not driven by rewarded elements; Suppl Fig 4d**).

Page 4 Supplement:

Statistical learning of spatial distance and transition frequency is not driven by rewarded elements

To confirm that our measures of statistically acquired knowledge were indeed reflecting knowledge of relationships between *all* twelve stimuli, we repeated the analysis with a second GLM that split stimuli into those belonging to the rewarded and those belonging to the control sequence (RewSeq and ConSeq, respectively). The original GLM had one joint onset regressor for all twelve stimuli and parametric regressors across these twelve stimuli. In the control GLM, the onsets of rewarded and control stimuli were instead modelled separately and separately associated with parametric regressors for spatial distance and transition frequency. We note that this analysis would be expected to be slightly less powerful, as parametric regressors rely on 4 out of 12 stimuli and thus a third of the data in each case. Nevertheless, six out of eight effects reached significance (all $p < 0.05$; **Suppl Fig 4d**) and the remaining two pointed in the same direction as in the initial analysis: for mPFC, transition frequency did not reach significance when fitted on RewSeq trials alone ($p = 0.18$) and for amyg/hippo, transition frequency did not reach significance for ConSeq trials alone ($p = 0.15$). Taken together, this is strong evidence that our measures of statistical learning were not driven by rewarded stimuli alone.

(3) In the last part of our answer, we want to address any remaining concerns about the initial rr-cc contrast being driven by a subset of 'pairs'.

One of the reviewer's concerns was that in our ROI-defining contrast 'rr-cc' there could be repetition suppression only for the 'correctly ordered' transitions together with an overall effect of $rr > cc$. We think the simplest way to address this question is to split up the 'rr' pairs into forward pairs, i.e., correctly ordered pair-wise transitions ('ForwPair': AB, xBC, xxCD) and other within-sequence pairs that are *not*

forwards-directed in the correct order but still transitions within the same sequence ('OtherWithinPair': BA, CB, DC, AD, DA, AC, CA, BD, DB). We can then plot the effects separately for these two sub-groups of trials for the rr-cc contrast (which was originally made up of the sum of these two subgroups of trials). Note that for simplicity, forwards pairs exclude occurrences of ABC and ABCD, i.e., those where full third- or fourth-order sequence relationships were fulfilled. We talk more about those in response to a later comment of this reviewer.

The plots below are for all four ROIs from left to right, with blue bars showing the rewarded and grey bars showing the equivalent pairs for the control sequence. This shows suppression for rewarded over control pairs is present for both ForwPairs and OtherWithinPairs. If anything, suppression is stronger and not weaker for the OtherWithinPairs.

A direct test of the interaction (which is orthogonal to the ROI selection) shows no interaction in amyg/hippo and pOFC (both $p > 0.5$), a trend-wise interaction in temporal pole ($p = 0.085$) and a significant interaction in mPFC ($p = 0.047$). In both of these latter cases, the interaction or trend-wise interaction arises because there is less, rather than more, suppression in forward pairs. Thus, in all cases, the rr-cc effect is not driven by only the subset of correctly ordered sequence transitions/pairs. We have included this plot in the supplementary figures (**Suppl Fig 2e**).

We have now made this clear in the revised manuscript as follows (page 8):

Finally, we confirmed that effects of cross-stimulus suppression indeed broadly applied to transitions across the rewarded sequence and were not driven solely by a subset of sequence pairs. For example, one might imagine that the observed effects could have been driven only by pairs of stimuli that appeared in the correct order (forward-ordered pairs) as opposed to pairs of stimuli that appeared

in a different order to that in which they had occurred prior to reward during the pre-scan learning task. Therefore, we repeated the previous analysis but split 'rr' and 'cc' pairs into repetitions that were forwards-directed ('ForwPair': AB, xBC, xxCD) and other within-sequence pairs that were not forwards-directed in the correct order but which were still transitions within the same sequence ('OtherWithinPair': BA, CB, DC, AD, DA, AC, CA, BD, DB). As expected, repetition suppression effects were apparent in both analyses [**Supplementary Results, Sequence fusion effects are not driven by only the subset of correctly ordered ('forward') pairs; Suppl Fig 2e** shows the rr-cc contrast separately for these two sub-groups of trials].

Supplement Pages 3-4:

Sequence fusion effects are not driven by correctly ordered ('forward') pairs

Our ROI-defining sequence fusion contrast (**Figure 2**) examined repeated presentations of stimuli from the rewarded sequence (rr) with repeated presentations of any two control stimuli (cc). The rationale was that if stimuli from the rewarded sequence have more overlapping neural representations, this should lead to more cross-stimulus suppression, compared to control sequence stimulus repetitions (contrast: rr-cc). To ensure that the observed effects were not driven by only the correctly ordered forwards pairs, we repeated the analysis but split 'rr' and 'cc' pairs into repetitions that were forwards-directed ('ForwPair': AB, xBC, xxCD) and other within-sequence pairs that were not forwards-directed in the correct order but still transitions within the same sequence ('OtherWithinPair': BA, CB, DC, AD, DA, AC, CA, BD, DB). **Suppl Fig 2e** shows the effects separately for these two sub-groups of trials. Note that forwards pairs exclude occurrences of ABC and ABCD, i.e., those where full third- or fourth-order sequence relationships were fulfilled. Suppression for rewarded over control pairs was present for both ForwPairs and OtherWithinPairs. If anything, suppression was stronger for the OtherWithinPairs. A direct test of the interaction (which is orthogonal to ROI selection) showed no interaction in amyg/hippo and pOFC (both $p > 0.5$), a trend-wise interaction in temporal pole ($p = 0.085$) and a significant interaction in mPFC ($p = 0.047$). In both of these latter cases, the interaction or trend-wise interaction arose because there was less, rather than more, suppression for forward pairs. Thus, in all cases, the rr-cc effect was not driven by only the subset of correctly ordered sequence pairs.

We note that this also implied that sequence fusion effects were not relying on transitions truly experienced during the pre-scan learning task, when only ForwPairs but no OtherWithinPairs were experienced. This points towards a more abstract representation of which elements 'belong' or do not belong to the rewarded sequence.

In summary, we have tried to be a lot more explicit, in the manuscript, about the complicated relationships between different contrasts and added visualizations wherever possible for clarification.

Again, we are sorry that we didn't make it clear that not all effects are sub-effects of the initial ROI selection. The initial ROI selection broadly speaking asked 'which areas cared about our learning task'. Within these areas we then test different kinds of knowledge - some of it reward-related and some purely statistical. These are not break-downs of the original contrast. For instance, spatial distance and transition frequency describe relationships between *all* 12 stimuli while rr vs cc (Fig 2) and correct order (Fig 3) contrast particular transitions between the rewarded and control sequence.

We hope that our new illustrations of the relationships between the various contrasts and the additional analyses we performed have provided sufficient clarification.

3. The foregoing points out the importance of clearly distinguishing activity related directed vs undirected stimulus-stimulus relationships. Another issue which could be better reconciled is the relationship between spatial adjacency and transition sequences. As I understand it, during training there is an instructed stimulus-stimulus sequence that is spatially disjoint, but subjects actually experience these as mediated by spatial paths over a series of intermediate stimuli, which are not experimenter controlled and apparently never directly used in analysis. Then in the scanning phase, a jumping cursor now presents stimulus-stimulus sequences skipping the spatially intervening stimuli. Given the results in Figure 4d, it seems quite possible that subjects initially learn associations (eg conditional transition frequencies) not over the instructed stimulus pairs directly, but over the spatially adjacent transitions actually experienced in training. This raises a series of questions in that virtually any of the results in the paper (from rr vs cc on down) could at least be potentially better explained in these terms. For instance, does the rr vs cc effect also hold for stimuli which are not in the official sequences but were included in the paths? It also provides the opportunity for a unified, formal view of the result from Figure 4d, in which a single learning model trained on experience which is initially temporally smooth (the actual paths taken by the subjects) and later temporally jumpy (the cursor in-scanner) interpolates from one regressor to the other. Similarly does the neighborhood effect in Figure 5 actually just follow adjacent squares, or is it sensitive to the series of spatially adjacent transitions into or out of the reward square actually experienced in training? That is, the notion of "spatial spread" here is confounded by temporal spread, given the experienced path statistics.

Thank you – this is a really nice idea. We had thought about the trajectories/paths that participants transitioned through in the context of Figure 5, also mentioned by the reviewer. We were similarly wondering whether the signals we were describing were really coding for spatial spread rather than temporal spread or a mixture of the two. Because all the analyses we had previously conducted pointed towards a signature best explained by spatial spreading of reward, this is what we focused on in the manuscript (we explain this in more detail in Part 3 below). But we agree it is also interesting to consider the actual paths taken for some of the other contrasts.

We will structure the answer to this point according to the reviewer's sub-points:

- (1) We tested the **rr-cc effect taking into account the experienced paths**. This shows that the paths do not shape the rr-cc sequence representation, and thus we have no strong basis for a unified model that goes from ‘smooth’ to ‘jumpy’.
- (2) We present **additional analyses to test for the presence of temporal spread effects** and possible **relationships between spatial and temporal spread effects** (related to Figure 5).

Part (1)

The reviewer raises the question whether the rr-cc effect holds if we consider the empirical paths that participants take between sequence elements. We first lay out **summary statistics of the participants’ empirical paths**, then show that **adding in transitional elements to the contrast does not change our result** and finally demonstrate that the **rr-cc effect is driven by the within sequence transitions rather than those additional elements passed** during the empirical trajectories. This suggests that the difference in neural representation between reward and control sequence is indeed due to a differential merge of sequence elements.

As a side note, the result shown above regarding ForwPair and OtherWithinPairs (point 3a above) already hinted at a sequence representation that is less driven by actual experience (or memories of specific experienced transitions) but by an abstract sequence construct to which several stimuli belong. That’s because above, the suppression was present for both ForwPair and OtherWithinPairs. However, the ‘OtherWithinPairs’, such as going from C do A were not part of the pre-scan learning task and thus not experienced before entering the scanner. One could thus, by extrapolation, similarly expect that the actual paths experienced are unlikely to impact this abstract construct of the rewarded sequence.

To directly address this, we looked at individual participants’ trajectories/paths to get from A to D (and similarly A’ to D’ for the control sequence). Across the group, there was some variance with regards to the trajectories taken which can be appreciated in the group mean plotted below, e.g. when two paths between e.g. C and D had an equal number of steps. Here we are plotting the frequency of visiting a certain state and the relevant sequence is shown to the left of each row.

Individual participants, however, converged onto their own preferred route that they used consistently, shown below for one representative subject. All 12 blocks completed on day1 are shown on the left, sorted by sequence type in rows. The column 'MEAN' shows the average for each sequence type across blocks. In order to choose which stimuli were considered part of the path between rewarded sequence elements for further analysis, we chose an arbitrary cut-off of 0.5, thus considering all stimuli visited in at least half the trajectories from A to D. The rightmost column "MEAN>0.5" thus shows for this participant which stimuli we considered to be on the 'rewarded path' (R path).

We then repeated our rr-cc contrast but coded not just pairs within ABCD but pairs within the newly defined path. It is worth noting, that because paths could not be fully anticipated and because this was not part of our original research question, we did not optimize our stimulus set to include such pairs on the path. Nevertheless, while the original rr and cc columns contained ~95 trials per block, by including pairs from the path, we added another ~60 trials on average, resulting in a regressor with a total of ~155 trials. As expected, because the majority (~95) of trials were overlapping, the original rr and cc convolved regressors correlated quite highly with the newly generated rrPath and ccPath convolved regressors (all >0.7, mean $r=0.735$).

Thus, not surprisingly, the results from the original GLM for rr-cc (plotted in blue) and the results from the new GLM (in grey) which was identical except that rr and cc were replaced by rrPath and ccPath, were not significantly different from each other in any of our ROIs (all $p>0.8$; note that the ROI selection was performed based on the contrast reflected in the blue bars):

This suggests that this effect might be mostly driven by the pairs between ABCD that we included in the original rr-cc contrast, and that it was less influenced by the pairs involving stimuli on the path. To test this, we separated rrPath into two columns, modelling the original rr pairs in one, and the newly included rrPath pairs in a separate regressor (and the same for cc and ccPath).

The below plot shows, for all ROIs, the rr-cc contrast for the original pairs (between A,B,C,D) on the left and the rr-cc contrast for **only** the newly added pairs that lie on the *path* between A,B,C, and D on the right (this would include transitions from path elements to A,B,C, or D and vice versa, always modelling the second element). Thus, this GLM was identical to the original one with just two additional columns: pairs formed in combination with path elements for rr and pairs formed in combination with path elements for cc.

The rrPath-ccPath contrast (shown below on the right) was not significant in any of the ROIs (all $p > 0.15$) and thus, the encoding of the rewarded structure seemed to be specific to the highlighted elements part of A,B,C,D. This is consistent with the instructions we gave participants which told them to focus on the highlighted elements and also informed them that the trajectories they would take did not matter.

So while we like the idea of a unified model suggested by the reviewer, we think that there is no strong basis for assuming trajectory experiences had a big impact on shaping the representations of relevance here. Because the analysis did not reveal new insights into the brain regions that we are focusing on, we have not added these new analyses to the revised manuscript but would be happy to do so if the reviewer suggests that they should be added.

Part (2)

To test for the existence of temporal spread signals, we constructed a parametric regressor that weighted elements that participants had transitioned through on their trajectory towards reward (i.e. from D backwards) with decreasing importance. We constructed an equivalent control parametric regressor from D' backwards, i.e. the last element of the unrewarded control sequence. We used a parameter sigma ('recency' kernel width) to weight how far back in time (steps) the reward was allowed to 'spread' on the trajectory between A to D that each individual participant took. Using small sigmas, the temporal spread parametric regressor was nearly identical to the regressor modelling D. Using large sigmas, the reward spread beyond A (see an example participants and run below illustrating the elements on the 4x3 array that reward spread to using sigma=0.5 vs sigma=5).

We therefore ran a GLM at the whole brain using the reward and control temporal spread regressors with an intermediate sigma of 3 (below is an example for one block of each participant showing how far back on the taken trajectories the reward can spread from D using a sigma of 3 – note that half of the participants had a different sequence because rewarded and control sequence were counterbalanced).

We focused on the contrast between the parametric regressors for temporal spread for the rewarded and control sequences. We did not identify any voxels signaling temporal spread in this contrast. This was true using whole-brain cluster correction or using ROI-based SVC, and there were no results in the amygdala or its vicinity even at lower exploratory z-thresholds. We repeated this analysis with the temporal spread residuals after regressing out D, in case this was due to the high correlations between the two regressors, but again even at uncorrected $z > 2.3$ there were no voxels in the amygdala.

One reason for this could be that during scanning, participants no longer experienced trajectories and thus, any trajectory-based temporal or spatial spread signals might have been weak.

In the spatial spread analysis in the manuscript, we included any neighbors of D (and D') even if they were not on the trajectory between C and D. Thus, our original spatial spread and this new temporal spread contrast, which reflects the completed trajectories, are not correlated (mean $\rho = 0.03$, even with different sigmas, it is < 0.1 in all cases). Intuitively, this is because only one neighbor was usually part of the trajectory that led to D and would have been eligible for temporal spread. And vice versa, other elements that were not neighboring elements to D were included in the temporal spread but not spatial spread, including D itself.

Again, this analysis did not reveal new insights into the brain regions that we are focusing on, so we have not added these new analyses to the revised manuscript. We would be happy to do so if the reviewer suggests that they should be added.

Minor points

- How were subjects instructed? They were explicitly told they were trying to discover the rewarded sequence?

We instructed them to find out how they 'unlock the treasure' and while we did not explicitly use the word 'sequence' when it came to their learning experience, we gave some examples in the instructions that would have made them realise it is about more than one stimulus. We would be happy to publish the instructions alongside this manuscript but will just copy the relevant paragraphs here for the time being:

"In this study, we are interested in how humans learn that a certain sequence of actions leads to reward. For example you have to boil the kettle, get a teabag, pour the water etc in the right order to prepare a tea. In our task, you are going to play a little agent (purple dot) who is visiting different 'rooms' in a maze (stimuli on the screen; see below). There are a total of 12 rooms in three different environments.

In each environment, your aim is to learn how to unlock a hidden treasure. You will move yourself around using buttons on the keyboard, and you are going to visit the rooms the computer highlights for you (yellow border) one by one - you cannot choose yourself which rooms to visit. Occasionally, the pattern of rooms you have visited will make you unlock the treasure.

When you encounter the treasure, you need to press the space bar (left index finger) as fast as possible. Sometimes you will also encounter a bridge (shown on the right). It has no function and you can ignore it, but again you need to press the space bar as fast as possible to pass it. It might help you to imagine that each room visit can have an effect on the agent: for example, you might be collecting a key in one room, or in another one you might get the 'code word' etc. Each room could have a unique function but initially you do not know which rooms need to be visited for unlocking the treasure. This is what you will have to find out by careful observation of the rooms you are visiting and the times when the treasure is encountered. So paying very close attention is a critical part of doing well in this task. Rooms that you simply pass through between different 'visits', but that were not highlighted, do not play any role and can be ignored. Therefore it also doesn't matter which path you choose to go to the highlighted room.

The buttons you use to move around are 'Left Arrow' and 'Right Arrow' for 'left and 'right' and 'Up Arrow' and 'Down Arrow' for moving up and down (all using your right hand). We will briefly practice this. In general, you need to press and release a button before the computer registers it.

Note that the treasure is always unlocked in the one particular way specific to the environment, meaning you will have to learn a total of three patterns in today's session."

We have added clarification to the Methods as follows (p.38):

An array of 3x4 abstract stimuli were presented on the screen in a fixed configuration (**Fig 1a**, left; **Fig S1b,c**). Unbeknownst to the participant, four of these twelve stimuli made up a sequence (referred to as ABCD) which lead to reward (rewarded sequence **RewSeq**). The aim of the behavioral training was to learn 'what' caused the reward, i.e. to learn the association between the sequence of four stimuli and reward. **More precisely, participants were instructed to find out how to 'unlock the treasure' and while we did not explicitly use the word 'sequence' at any point, we gave some examples in the instructions that would likely have made them realize that learning was about more than one stimulus.**

- CS is unfortunate terminology for control sequence, given that it often means conditioned stimulus in this literature.

That's a very good point – thanks. We have changed it to control sequence/ConSeq and rewarded sequence/RewSeq throughout the manuscript.

- Figure 2c is missing axis labels

Thanks, these have been added now.

- I'm willing to concede to convention, authority, or the authors' poetic license here, but I'm personally wary about the verb "encoded" used extensively in this article to mean "is correlated with," as in "BOLD activity in area x is correlated with regressor y." I realize this is shorthand but to my ear this particular word smacks a bit of assigning a purpose to the modulation when of course, absent any causal manipulation (and given the possibility that significant correlations might be driven by other correlated regressors) there is no reason to assume the activity is actually doing anything or

“encoding” this particular signal. I find this most potentially confusing in the section on ventral striatum which concludes the area “plays no role” and has “no involvement” in this encoding, which combines this oddly active phrasing with null hypothesis affirmation.

Thanks, we have changed this verb throughout wherever possible.

- I don't see why expectation building up as the reward sequence progresses (figure 3c) has anything at all to do with the idea that the structure is learned by backpropagation (line 223). This will be true of any notion of temporally discounted reward expectancy or reward proximity, regardless how the associations underlying it were acquired or how it is computed.

Thanks, the reviewer is right that we cannot prove, based on this type of activity pattern, that the structure was learned by backpropagation. Nevertheless, it is hard to imagine how else it might have been learnt given the task and the behaviour observed in Figure 1d. This shows that earlier elements in the sequence were learned as part of the sequence later, or in other words, D was associated with reward first and A was associated with it last.

We have now added to the text that the BOLD signature we observe is also consistent with temporally discounted reward expectancy or reward proximity.

Page 9: Such a signal could reflect temporally discounted reward expectancy or reward proximity. However, since participants' behavior showed that the reinforced associative structure is formed backwards, another possibility is that this BOLD signature was formed via backpropagation of reinforcement (Figs 1d, Suppl Fig 1e). Note that because reward is predicted by multiple stimuli, unlike in experiments that only train first-order conditioned associations, here reinforcement learning leads to knowledge not only of stimulus-reward but also stimulus-stimulus relationships, just like in many animal studies that consider higher-order conditioning chains²¹.

Page 17 (discussion):

Our first key finding demonstrated BOLD representations of the reinforced sequence in its correct order in temporal pole and pOFC, with an increase in activity as participants progressed through the correctly ordered sequence associations (Fig 3 and Suppl Fig 3; Fig 7a). While such a BOLD increase would also be consistent with temporally discounted reward expectancy or reward proximity, participants' behavior clearly demonstrates that the reinforced sequence was first acquired for the elements closest to reward. Thus, while we did not look at BOLD changes during learning in this task, this suggests that the neural signals identified for the correct transitioning through the rewarded sequence might have formed in a manner consistent with backwards propagation of reward posited in RL models (Fig 1 and Suppl Fig 1).

- Is the A-B A-B-C A-B-C-D activity increase really dependent on the third and fourth order structure?

Would you see the same thing if you looked for X-B-C and X-X-C-D?

That's a very good question and again, we had checked this and apologize this didn't make it into the manuscript. Indeed, it does rely on the full (third or fourth) order structure.

For illustration, in the below time-course plots we show X-B-C and X-X-C-D as the reviewer is asking for but where X is *not* A or A-B, respectively. We separately show A-B-C and A-B-C-D for comparison, when the full third/fourth-order structure was present.

The number of trials that go into each of these bins was higher for A (~217) because A was not split by correct order but it was roughly comparable for B, C and D in the analysis looking at the correct pairwise order AB, xBC, xxCD (~83, ~47, ~73), but with diminishing trial numbers later in the sequence going from AB to ABC to ABCD in the analysis with correct third-/fourth-order contingencies (~83, ~48, ~37).

To better balance trial numbers, we merged across C and D which showed the same effect but slightly more convincingly.

The above time courses illustrate the effect. The critical time course is the green one showing neural responses to C/D if experienced correctly preceded by the entire sequence (1st and 3rd panel) and preceded only by one correct element (2nd and 4th panel). To run a statistical test probing for third/fourth-order dependencies, we extracted parameter estimates from a GLM estimated on convolved regressors and using the same basic structure as in GLM1 in the manuscript. Instead of ‘rr’ and ‘corrRSeqOrder’ we modelled one column with all occurrences of xBC and xxCD in, and one column with all occurrences of ABC/ABCD in (and all other rr pairs of no interest here in their own separate onset column). We ran a 2x2 repeated-measures ANOVA on the resulting parameter estimates with factors Area (pOFC/tempPole) x HigherOrder (fulfilled yes/no, i.e. ABC/ABCD vs xBC/xxCD) and found a significant effect of HigherOrder ($F(1,25)=5.712$; $p=0.025$) but no effects of Area or interaction between Area x HigherOrder. Consistently, the winning model in a Bayesian rs-ANOVA was a model with HigherOrder factor but no other factors ($BF_m=4.936$, $P(M|data) = 0.552$).

Repeated Measures ANOVA ▾

Within Subjects Effects

	Sum of Squares	df	Mean Square	F	p
Area	3.044e-4	1	3.044e-4	0.523	0.476
Residual	0.015	25	5.824e-4		
HigherOrder	0.003	1	0.003	5.712	0.025
Residual	0.013	25	5.368e-4		
Area * HigherOrder	7.675e-4	1	7.675e-4	2.451	0.130
Residual	0.008	25	3.131e-4		

Note. Type III Sum of Squares

Between Subjects Effects

	Sum of Squares	df	Mean Square	F	p
Residual	0.059	25	0.002		

Note. Type III Sum of Squares

Bayesian Repeated Measures ANOVA ▾

Model Comparison

Models	P(M)	P(M data)	BF _M	BF ₁₀	error %
HigherOrder	0.200	0.552	4.936	1.000	
Null model (incl. subject)	0.200	0.170	0.821	0.308	0.860
Area + HigherOrder	0.200	0.148	0.693	0.267	1.825
Area + HigherOrder + Area * HigherOrder	0.200	0.084	0.367	0.152	3.607
Area	0.200	0.046	0.191	0.083	1.538

Note. All models include subject.

Thus, in summary, pOFC and tempPole BOLD signatures of the correctly ordered sequence did indeed depend on the full third- or fourth-order structure. Thank you for raising this point which we believe has strengthened the manuscript.

We have included this as follows on page 9 of the main manuscript and in **Fig 3c**

Page 9: In these regions, the BOLD signal was stronger when a correct transition was experienced within the rewarded compared to the control sequence. Examination at the whole-brain revealed bilateral effects in both regions (**Suppl Fig 3**). Further inspection of the signal in pOFC and tempPole suggested a build-up of reward expectation as participants advanced through the rewarded sequence (GLM main effect of progress: $p=0.012$, $t(25)=2.70$, ROI and ROI x progress: $p>0.1$; illustrated in **Fig 3c**), and that the entire third- and fourth-order structure (e.g., ABC not just BC) had to be fulfilled for this build-up to be present (ANOVA ROI x HigherOrder, effect of HigherOrder $F(1,25)=5.712$; $p=0.025$; Supplementary Results and **Fig 3c**).

Page 4, Supplement:

Correct sequence order encoding depends on third- and fourth-order structure

We probed whether increases in BOLD activation in temporal pole and pOFC when transitioning through the correctly ordered rewarded sequence (i.e. A, AB, ABC, ABCD) were dependent on the third and fourth-order contingencies. Alternatively, this signal could be present even when only the pair structure (AB, BC, CD) is fulfilled. For this analysis, we focused on ABC versus xBC (a pair BC not preceded by A, bold indicates time-locking), and ABCD versus xxCD because A and AB by definition do not rely on higher-order chains. We extracted parameter estimates from a GLM that was almost identical to GLM1, except that instead of regressors (1) rr and (3) corrOrderRewSeq, we modelled one regressor with all occurrences of xBC and xxCD, and one regressor with all occurrences of ABC/ABCD, plus other rr pairs of no interest here in their own separate regressor. We ran a 2x2 repeated-measures ANOVA on the resulting parameter estimates with factors Area (pOFC/tempPole) x HigherOrder (fulfilled yes/no, i.e. ABC/ABCD vs xBC/xxCD) and found a significant effect of HigherOrder ($F(1,25)=5.712$; $p=0.025$) but no effects of Area or interaction between Area x HigherOrder. Consistently, the winning model in a

Bayesian rs-ANOVA was a model with HigherOrder factor but no other factors (BFm=4.936, P(M|data) = 0.552; Fig 3c).

Fig 3c:

Figure 3, Correct sequence order ('path to reward') in temporal pole and pOFC

a, The **encoding** of correctly ordered sequence associations were examined using occurrences of B after A, C after AB, and D after ABC, and by contrasting **RewSeq** and **ConSeq**. In other words, this contrast probed **RL-acquired directed higher-order stimulus-stimulus associations**. **b**, pOFC and tempPole, but not mPFC and amyg/hippo, showed stronger **BOLD responses** for correct sequence order for **RewSeq vs ConSeq** and thus tracked the path to reward learnt pre-scanning. **c**, Illustration of the effects: (Left) The average time-series to A, (A)B, (AB)C, and (ABC)D illustrate an increase in BOLD activation as participants progress through the correctly ordered associations. (Right) Higher-order associations were necessary for driving the BOLD signal: BOLD average for C and D elements when higher-order conditioned chains were respected (red; average of ABC or ABCD, bold highlights indicate time-locking) compared to when only the pair-structure was fulfilled (average of xBC, xxCD; grey). **d**, The **BOLD representation** of correct sequence order was robust and did not change over scan blocks despite being irrelevant for the task performed during scanning. Error bars denote S.E.M. See also Supplementary Figure 3.

- The transition learning model is confusing. What does it mean to demean it (by rows? columns??)? And given that it's expressed in transition counts, how strong is the prior? Is it initialized with elements between 0 and 1 according to the trained probabilities, but then counts up by adding 1? In this case the "prior" would wash out very quickly and play almost no role. Why not initialize with the number of transitions actually experienced during training?

This is another very good point, and we are sorry for not clarifying some of these points.

When it comes to the prior, it is indeed using the actual number of transitions experienced during training (for day1; for day2 it is still actual numbers of experienced transitions but taking into account

the first scan session in addition to the training). So it exactly as the reviewer thought, rather than numbers between 0 and 1. We have now clarified this in the text.

This also means that every experienced transition, whether from training or scanning, had equal weight and that transitions converged slowly, over the duration of scanning, from the training transitions towards the ones experienced during scanning.

For illustration, we plot one example scan block and show the model-updated transition matrix (1-12 are the stimuli and 13 the reward) after 20, 60, ... and up to 380 trials. The rewarded sequence ABCD was 1-5-3-10 in this context and the training and scan transitions are shown for comparison in the bottom row.

The reason for demeaning across the entire matrix (matrix = matrix - mean(matrix(:))) was so that the meaning of adding +1, i.e. the overall scale of numbers, remained unchanged, while at the same time avoiding a monotonically increasing parametric regressor. We did check carefully when writing up our manuscript, whether this made any difference and it does not. For example, alternatively, we could have just ignored the fact that values grow over time, or detrended the resulting parametric regressor. The non-demeaned and our original demeaned regressor are shown below in red and blue, respectively, and they correlate at $\rho=0.9936$. This means that the demeaning was not a critical part of our analysis and using a non-demeaned regressor or non-demeaned detrended regressor would not change our results. We now highlight this more clearly in the Methods section of our manuscript.

Pages 42-43:

Learning model of statistical relationships

We designed a simple model to capture the underlying statistical relationships present in the scan task. The aim was to obtain a trial-by-trial measure of the frequency (or probability) with which participants expected each of the possible transitions from one element to another within the task space. For the first scan session, we set the prior to the frequencies experienced during the pre-scan learning task. **Frequency here refers to the actual number of times a transitions was experienced during training, in integer numbers.** These were saved in a 13x13 asymmetrical matrix TF (12 stimuli + reward), storing, in each cell TF(i,j), the value of the frequency of going from stimulus i to stimulus j (as shown in **Suppl Fig 4d**). Note that the sum along the rows and columns indicated how often participants experienced stimulus i and j overall, respectively. During the pre-scan learning task, each stimulus was shown equally often, so normalizing TF by rows or columns, thus transforming frequencies into probabilities, did not have any effect apart from a scaling of the matrix as a whole. For each trial in the scan task (starting from trial 2), the prior was then updated for the experienced transition from i to j by adding 1, i.e.

$$TF(i,j) = TF(i,j)+1;$$

This meant that every experienced transition carried equal weight during updating. The matrix TF was then demeaned and used as a prior for the next trial. **We decided to demean to ensure that the overall scale of the numbers remained unchanged while at the same time avoiding a monotonically increasing parametric regressor. We checked carefully whether this decision mattered: for example, the correlation between the demeaned regressor and an alternative that did not include this demeaning step but detrended the resulting regressor at the end, was r=0.9936.**

- Why are the two types of test questions in supp figure 4 described as explicit vs implicit recall? I

think a memory person would describe this as a recognition (not a recall) test, and I don't see why endorsing lures should be implicit but correct recognition should be explicit?

We apologize for our lack of memory terminology. We have changed this to 'Recognition (correct stim)' and 'Error detection (wrong stim)'.

- The discussion of Figure 1e suggests the CS is learned in a forward manner, in contrast to RS. I don't see any evidence anywhere in the results that the CS is learned at all?

We have mentioned in a response to the reviewer's 1st point above that there is a main effect of speeding up for transitions later in the sequence (i.e., from AB to BC to CD) when just looking at the control sequence on its own (i.e. the grey bars in Fig 1c). That means that even in the control sequence, transitions are anticipated even if to a lesser extent than for the rewarded sequence. This has now been added to the manuscript.

Because we do not probe explicit knowledge of the control sequence, it is difficult to show the directionality of the learning for the control sequence. There is a hint that during training, when choosing between two stimuli that are both part of the control sequence (Figure S1e the reviewer is referring to), participants have a preference for earlier elements A and B over C and D. However, we agree this effect is small. The revised manuscript now summarizes the evidence that the control sequence is learned in the following way (pasted to one of the above points already, page 6):

Behavioral evidence for statistical learning

We next probed whether participants showed evidence for incidental learning which should be unrelated to reward and driven by repeated observation of statistical relationships, such as the transitions between control sequence stimuli. We returned to the RTs for the first movement that initiates transitioning between two stimuli of the control sequence during the pre-scan learning task (grey bars in Fig 1c, also part of the larger ANOVA above). We asked whether initiating the button press from A' to B', followed by B' to C' and C' to D' would show RT speeding because transitions, despite never being associated with reward, became increasingly predictable. Note that these transitions were experienced as frequently as those in the rewarded sequence. A 1x3 repeated-measures ANOVA with factor Transition (A'B'/B'C'/C'D') run only on the control sequence transitions showed a significant effect of transition ($F(2,50)=28.553, p<0.001$; Bayesian repeated-measures ANOVA shows evidence for a model with factor Transition: $BFm=1.522e+6, P(M|data)=1$). Thus RTs became faster as participants progressed through the control sequence, but as shown above, this speeding was less pronounced than in the rewarded sequence.

Reviewer #2 (Remarks to the Author):

The paper „Multiple associative structures created by reinforcement and incidental statistical learning mechanisms“ asks how the brain encodes the knowledge resulting from reinforcement learning. It contrasts associative structures that arose from reinforced stimulus sequences against structures that arose from non-reinforced stimulus sequences. The findings include reduced activity to reinforced sequence elements (compared to non-reinforced sequence elements) in medial prefrontal cortex, temporal pole, posterior orbitofrontal cortex (pOFC) and amygdala/hippocampus. Among these regions, correct reinforced sequences (compared to correct non-reinforced sequences) activated temporal pole and pOFC. Both spatially close and frequently experienced transitions were associated with medial prefrontal cortex and amygdala/hippocampus activation, regardless of sequence type. Spatial proximity to final elements from reinforced stimulus sequences activated amygdala/hippocampus more than final elements from non-reinforced stimulus sequences. Finally, the ventral striatum responded to the first element of the reinforced stimulus sequence over and above the first element of the non-reinforced sequence.

There are several aspects of the study to like, particularly the detailed analysis of how reinforcement affects the encoding of time and space within sequences. However, some aspects temper enthusiasm, particularly how the authors aim to claim novelty, the lack of explanation for the unexpected finding that typical reward regions are less active for reinforced than non-reinforced sequence elements and how the ventral striatum results are framed.

We would like to thank the reviewer for his/her very thoughtful and detailed points which we have addressed one by one below. We agree that there are some novel aspects to this study while other results have been shown before and we try to be clear about this in the new version of the manuscript.

On another note, however, we were concerned that the reviewer had been left with the impression that there is a main effect in activation whereby typical reward regions are ‘less active’ for the rewarded sequence. This is not the case as we will lay out in detail below. In brief we think that the confusion may have stemmed from the fact that some analyses are repetition suppression-style analyses (in such analyses, task event-related activity may *increase* each time the event occurs but the degree to which it does so is modulated by the degree of representational overlap, i.e. by how many of the same neurons were recruited for the previous and the current stimulus). We have revised the manuscript in light of the reviewer’s comments and hope that it will no longer be possible to misinterpret the results in this way.

Finally, we agree that in the original version of the manuscript, we did not emphasize enough that the role of the ventral striatum is actually entirely consistent with predictions from TD-learning. We have tried to change the relevant sections to appropriately reflect this. Thanks again for your helpful comments. We hope the below answers will sufficiently address the concerns raised.

I noticed the following issues that should be addressed:

1. The framing of the paper aims to set up a contrast between process (conditioning) and product (referred to as “learned knowledge”) of learning. If I understand correctly, learning about reinforced sequences essentially corresponds to a situation of higher-order conditioning. This kind of learning

has been investigated before (e.g., Seymour et al., 2004, Nature; Gilboa et al., 2014, Current Biology; Pauli et al., 2015, Journal of Neuroscience) which the authors may want to spell out. More importantly, neuroimaging took place in extinction, which should have led to continued learning (namely extinction learning), a point that the authors gloss over and that makes the presumed contrast between process and product much less clear-cut. By extension, the authors may have implicitly dealt with neural regions that show extinction learning at different time scales (for related work see e.g., Glascher & Buchel, 2005, Neuron). The fact that the identified regions show relations to reinforcement throughout the imaging phase could suggest that these regions learn slowly. This point seems to obliterate the present framing of the paper and in the minimum needs to be mentioned.

Thank you very much for this comment and for pointing us towards the literature on higher-order conditioning. We will try and structure our response as follows by discussing:

- (1) The framing of our work as **higher-order conditioning** in relationship to the various papers highlighted by the reviewer, and the consistent **role for VS in encoding TD-prediction errors**.
- (2) Whether it is indeed the case that we probed **learning in extinction**
- (3) The **speed of learning** we observe for RL-driven vs statistically learnt information

Part (1)

We agree that the sequence learning we describe can be likened to higher-order conditioning. We have now added this point to the manuscript. One key difference might be that learning in higher-order conditioning is usually performed in the presence of only the relevant cues (i.e. visual conditioned stimuli in Seymour et al and Pauli et al, or tones in Gilboa et al.). Thus, the cues are very salient and there is no need to learn which particular stimulus out of many potential stimuli are relevant, which was the case in our task.

Nevertheless, our findings in the striatum very much agree with Seymour et al. (Nature, 2004): ventral striatum, with the peak of activation in putamen in both our and their study, encodes prediction errors consistent with TD learning which would predict a positive encoding of A, the first predictive stimulus in the chain, as shown in the figure below pasted from Seymour et al.

Which matches exactly what we found (our original Figure 6d):

Although the section on the ventral striatum (VS) is not the focus of our study (rather our aim is to contrast the activity patterns there with those found elsewhere), we have now revised the section on VS to reflect the relationships between our findings in VS and those of Seymour et al., Pauli, et al., and Gilboa et al. and we have made sure that we have cited their work. The revised manuscript now includes the following sections:

Pages 2-3

Our aim in the current study was to take an alternative approach to investigate the neural mechanisms associated with model-free and model-based learning. Rather than focusing on the PEs that occur during learning, instead we focus here on the knowledge or ‘associative structures’ that are formed by, and subsequently guide, the learning process. **In order to do this we examined a situation in which participants did not just learn associations between a single predictive stimulus and subsequent reward but instead had the opportunity to learn both predictive links between chains of stimuli leading up to reward^{2,20,21} and statistical relationships between stimuli regardless of whether they were predictive of reward.** This allowed us to test whether the associative structures derived from different learning processes might prove more distinguishable than their PEs. **We therefore undertook two main series of analyses of behavior and of neural activity that contrasted the consequences of these two types of learning in our experiment.**

Pages 8-9

BOLD representations of correct sequence order: the RL-driven associative structure

Nevertheless, this type of response pattern – indicating a shared representation of the elements that belong to **the rewarded sequence** regardless of precise order – would not have been sufficient, in the pre-scan learning task, for anticipating each correct subsequent element. It would not have enabled a participant to predict reward because all four steps in the correct order were necessary before reward ensued and thus, it does not capture the full extent of RL knowledge acquired prior to scanning. We therefore reasoned that a BOLD signature of the correctly ordered **rewarded sequence** should also be present. A brain region in which reward is back-propagated to previous stimuli, for example via TD learning, should demonstrate a particular sensitivity for the correctly ordered stimulus sequence that has led up to reward in the past. Within the same GLM as used above (GLM1, **Suppl Fig 2d**), we tested whether any of the four regions reflected knowledge of the correct sequence order (**Fig 3a**). We compared activity between **RewSeq and ConSeq** when the correct stimulus transitions were experienced (i.e. B after A, C after AB, and D after ABC). Within the ROIs defined above, only two regions, the

temporal pole and adjacent pOFC, exhibited differences in BOLD for correct order transitions of **the rewarded** compared to **the control sequence** ($p(\text{pOFC})=0.0014$, $t(25)=3.59$; $p(\text{tempPole})=0.0051$, $t(25)=3.07$; **Fig 3b**). In these regions, the BOLD signal was stronger when a correct transition was experienced within **the rewarded** compared to **the control sequence**. Examination at the whole-brain revealed bilateral effects in both regions (**Suppl Fig 3**). Further inspection of the signal in pOFC and tempPole suggested a build-up of reward expectation as participants advanced through **the rewarded sequence** (GLM main effect of progress: $p=0.012$, $t(25)=2.70$, ROI and ROI x progress: $p>0.1$; illustrated in **Fig 3c**), and that the entire third- and fourth-order structure (e.g., ABC not just BC) had to be fulfilled for this build-up to be present (ANOVA ROI x HigherOrder, effect of HigherOrder $F(1,25)=5.712$; $p=0.025$; Supplementary Results and **Fig 3c**). Such a signal could reflect temporally discounted reward expectancy or reward proximity. However, since participants' behavior showed that the reinforced associative structure is formed backwards, another possibility is that this BOLD signature was formed via backpropagation of reinforcement (**Figs 1d, Suppl Fig 1e**). Note that because reward is predicted by multiple stimuli, unlike in experiments that only train first-order conditioned associations, here reinforcement learning leads to knowledge not only of stimulus-reward but also stimulus-stimulus relationships, just like in many animal studies that consider higher-order conditioning chains²¹.

Pages 15-16 (Results)

Prediction error-related activity in the ventral striatum reflects knowledge of the reinforced sequence

The ventral striatum plays an important role in learning from reinforcement. An extensive body of work has demonstrated detailed features of prediction error coding in ventral striatum at the time of outcome (see Introduction). We therefore tested, based on an *a priori* hypothesis, whether ventral striatum was also involved in **representing knowledge learnt** in our task **or if its activity was modulated by the prediction errors that would be experienced if such knowledge were represented elsewhere in the brain**. In other words, **first**, we asked whether **it exhibited activity reflecting** knowledge constructed via reinforcement (**RewSeq** versus **ConSeq**; correct sequence order), knowledge of the task state transition structure (spatial distance; transition frequency), or information about reward proximity (spatial spread), and thus any of the effects described in pOFC and temporal pole, mPFC and amygdala-hippocampus border, and (central) amygdala, respectively. In an anatomically defined ROI (see Methods), none of these effects were present (all $p>0.4$; **Fig 6a**), suggesting no **evidence for an** involvement of ventral striatum in representing RL-derived knowledge once it has been consolidated,

and no learning of general task structure. The absence of associative knowledge in ventral striatum cannot be explained by a generally lower signal in ventral striatum because the mean temporal signal-to-noise ratio (tSNR) in this ROI was better than in the amyg/hippo and pOFC ROIs, and no different from vmPFC or temporal pole ROIs, nor the mean tSNR measured across the brain.

However, even if it is the case that there is no evidence that ventral striatum itself possesses activity that reflects the presence of stored associative knowledge, it is possible that its activity reflects the prediction errors that might be experienced given that such knowledge is present elsewhere in the brain. Notably, RL theory would predict that the prediction error would move, over time as learning progresses, to the earliest stimulus that predicts reward. In our task the earliest predictive stimulus is the starting element A of the rewarded sequence. We therefore tested, whether ventral striatum would code A more strongly for the rewarded compared to the control sequence. Indeed, we found responses to A over and above A' in lateral ventral striatum (cluster-corrected $p < 0.05$ in the right hemisphere: peak -24,14,-8, $z = 3.30$) in a region often referred to as putamen⁴³. But even in an ROI based on this coordinate, none of the other tests described above were significant (all $p > 0.1$; **Figs 6b-d**), thus corroborating the idea that ventral striatum did not possess knowledge of the entire sequence or code flexible knowledge about the task structure but that it did reflect prediction errors that would be experienced given the presence of this knowledge elsewhere in the brain. Finally, consistent with RL predictions, in both anatomically and functionally defined ROIs, ventral striatum coded for the picture of the flower (the reward during the scan task $p(\text{anatVS}) = 4.47e-09$, $t(25) = 8.75$; $p(\text{funcVS}) = 1.32e-09$, $t(25) = 12.98$), and in the functionally defined ROI also the picture of the treasure (the reward during learning; $p(\text{funcVS}) = 0.009$, $t(25) = 2.82$). These appeared at somewhat (treasure) or entirely (flower) unpredictable times during scanning (**Figs 6a,b**).

Page 21 (Discussion)

Absence of complex RL or statistical knowledge but presence of prediction errors in ventral striatum

Our final striking observation was that the ventral striatum BOLD response did not distinguish between rewarded and control sequence elements, relate to correctly ordered reinforced knowledge, task statistics, or a spread-of-reward signal (**Fig 6 and Suppl Fig 6; Fig 7c**). Instead, ventral striatum showed prediction error-like responses at the occurrence of the first element of the reinforced sequence and thus precisely the signal that would be predicted by TD theory.

Part (2)

As the reviewer has correctly pointed out, we did not probe the *learning* of the rewarded sequence (which was done prior to scanning) but focused our design on the **‘product’ of RL learning** (i.e., the resulting knowledge). However, it is not so clear whether learning took place in **extinction** or not but there were certainly some clear changes between the learning and the scan task which it might be worth summarizing and highlighting more clearly:

First, the **‘treasure’ (reward) was no longer relevant** for behavior in the scanner. Instead of focusing on how to get to the treasure, participants were simply detecting flowers on the screen. Because the previously learnt representations became irrelevant we thought that the BOLD responses to the original rewarded sequence might fade or even vanish over time. This was what we aimed to test when we looked for effects related to the encoding of the correctly ordered sequence across blocks (the three blocks of scanning on day 1 followed by the three blocks of scanning on day 2). We note that each estimate in such an analysis is noisier as it relies on a sixth of the data but nevertheless, significant changes over time could be observed in other analyses (e.g. Fig 4d for spatial distance and transition frequency). However, no changes in responses to the correctly ordered sequence were observed in pOFC or temporal pole, highlighting the persistence of this signature of RL coding. This is shown in Fig 3d pasted here for reference:

Secondly, even if the treasure was no longer relevant, as mentioned in the Methods, during the scan task, the **treasure was still sometimes presented after A,B,C,D**. It was, however, also sometimes presented after only A,B or after x,x,C,D. We thought this might give us a handle on prediction errors but unfortunately any analyses that focused on AB-Reward or xxCD-Reward seemed too weak to show any conclusive results, maybe also because this was irrelevant to the participants' behavior at the time of scanning (in a different paper, we have previously shown that VS encodes prediction errors about whatever is relevant to learning, which doesn't have to be reward – Klein-Flugge et al., Neuron, 2011). So it seems fair to say that the scan task was not performed in complete extinction but in partial extinction. Reward only sometimes occurred after A,B,C and D, and sometimes it occurred in other places. Evidence of new knowledge of reward predictors (i.e. the new places where reward occurred) was not apparent perhaps because by this stage the reward had become irrelevant to the task at hand. This suggests that the representations of sequences that originally led to reward in the initial training

period were ‘frozen’ and unchanged because no new reward-learning interfered with the originally formed representations. We have revised the manuscript to make this clearer.

Pages 9-10:

Notably, unlike in the pre-scan learning task, knowledge of **rewarded sequence** was irrelevant for participants during the scan task. Performance in the scan task simply depended on responding to catch trials. We therefore tested whether the correct sequence order effects reported above decreased as the scanning session progressed. Contrast estimates for correct sequence order (**RewSeq vs ConSeq**) were extracted from pOFC and temporal pole, separately for each block. Because frequentist statistics cannot provide evidence in favor of the null hypothesis, we conducted a 2x6 Bayesian repeated-measures ANOVA with factors ROI (pOFC and tempPole) and block. **This JZS Bayes factor ANOVA^{33,34} with default prior scales revealed that the Null Model was the best model and almost six times more likely than any other model (BFm=5.81, P(M|data)=0.59), including a model with an effect of Block (BFm=1.43, P(M|data)=0.26).** Thus, representations of the correctly ordered reinforced sequence in temporal pole and pOFC were remarkably inflexible and long-lasting over the duration of our experiment despite changing task demands. **In summary, the representations of sequences that originally led to reward in the initial training period appear almost to have been ‘frozen’ in a stable state in pOFC and tempPole. It is possible that they are unchanged because no new reward-learning occurred during the scanning period that could interfere with the originally formed representations.**

Part (3)

We agree that our data might have identified **learning at different time scales** and indeed, the manuscript already makes some related points. However, we admit, the way we had thought about the speed of learning and how it relates to the different types of knowledge (RL versus statistical) was slightly different from the suggestions the reviewer is making.

Behaviorally, participants learned the sequences quite fast, in most cases within a few blocks of training (each block entailed six occurrences of the rewarded sequence), and thus after ~20 experiences of the rewarded sequence (e.g. Fig 1d). We believe it is possible that the encoding of the knowledge during learning happens at a fast time-scale while extinction/forgetting/unlearning is slow. This is how we had conceptualized the RL learning so far. In our minds, it was knowledge that was acquired quite fast, probably due to the boost in motivation provided by the reward, but despite scanning being performed in (partial) extinction and during an unrelated task, the signatures related to RL learning were persistent and did not significantly change over the six blocks of scanning. This was true for the sequence fusion

tests (rr-cc; mPFC: $p=0.56$; amyg/hippo: $p=0.38$; nor in tempPole: $p=0.6$ or pOFC: $p=0.41$) as well as the correct order representations (shown above in Part 1).

We think that this is maybe somewhat different from the situation in Glascher & Buchel's task where old learning had to be over-written by new learning continually because contingencies were constantly changing. The situation in the current study is also different from that used in some of our own previous work that also, like Glascher and Buchel, addressed the question of multiple learning rate-based representations (Wittmann et al., 2016; Meder et al., Nature Communications, 2017). It is possible that learning of a new sequence within the same context could have meant that representations in temporal pole and pOFC changed over the duration of scanning but unfortunately we cannot test this here because the ABCD-Reward associations were kept constant in each context.

By contrast, the **learning of statistical relationships** has in the literature in general been thought of as **slower** than reward-learning (for example in Tolman's maze, there was a group of rats that was rewarded when reaching the goal and another group that was not; the former group learnt faster but the latter group still learned, just more slowly). This is also true here: while there is some evidence for statistical learning in the pre-scan learning task, which we can probe by looking at the unrewarded control sequence, it is much less pronounced than the effects observed from reward-learning, which we probe by looking at the rewarded sequence. This is despite the fact that the control sequence was experienced just as frequently (six times per block) as the rewarded sequence.

Evidence for slow/weaker statistical learning can be appreciated in Fig 1c also part of the original manuscript and shown again on the left. This plots RTs for the initiation of a trajectory during learning. We originally only reported a main effect of 'transition' for both sequences together in addition to the interaction of sequence x transition. A result we did not report but which provides evidence for statistical learning comes from an ANOVA solely run on the control sequence transitions (i.e. the grey bars). This shows a significant effect of transition: $F(2,50)=28.553$, $p<0.001$; model with factor transition: $BF_m=1.522e+6$, $P(M|data)=1$, consistent with

anticipation of upcoming stimuli even in the unrewarded control sequence.

During scanning, participants continued to track and **update the experienced transitions**, which we modelled using trial-by-trial updates and probed using the resulting transition frequency regressor. We observed how a representation of the transition frequency code emerged over the six blocks of scanning (as shown in Figure 4D pasted below). These changes are slower than the initial acquisition of the rewarded sequence but faster than the 'extinction' of the rewarded sequence knowledge. Maybe statistically formed knowledge is constantly being updated on a **slow time scale**. This will have to be directly tested in future work but again, it is an important point we now discuss it in the manuscript. Thank you - we think that this comment has contributed important points to our revised manuscript.

In addition to the paragraph pasted above, the revised sections of the manuscript now reads as follows:

Page 6

Behavioral evidence for statistical learning

We next probed whether participants showed evidence for incidental learning which should be unrelated to reward and driven by repeated observation of statistical relationships, such as the transitions between control sequence stimuli. We returned to the RTs for the first movement that initiates transitioning between two stimuli of the control sequence during the pre-scan learning task (grey bars in **Fig 1c**, also part of the larger ANOVA above). We asked whether initiating the button press from A' to B', followed by B' to C' and C' to D' would show RT speeding because transitions, despite never being associated with reward, became increasingly predictable. Note that these transitions were experienced as frequently as those in the rewarded sequence. A 1x3 repeated-measures ANOVA with factor Transition (A'B'/B'C'/C'D') run only on the control sequence transitions showed a significant effect of transition ($F(2,50)=28.553, p<0.001$; Bayesian repeated-measures ANOVA shows evidence for a model with factor Transition: $\text{BF}_m=1.522\text{e}+6, P(M|\text{data})=1$). Thus RTs became faster as participants progressed through the control sequence, but as shown above, this speeding was less pronounced than in the rewarded sequence.

Page 12

To complete the picture, we tested whether statistically learnt knowledge in mPFC and amyg/hippo was, over blocks of scanning, replacing the undirected knowledge of the rewarded sequence, as probed by our initial ROI-defining contrast (sequence fusion: 'rr-cc'). If so, we would expect a decline in the sequence fusion contrast over blocks of scanning. We did not observe such a change in either region (mPFC: $p=0.56$; amyg/hippo: $p=0.38$; nor in tempPole: $p=0.6$ or pOFC: $p=0.41$), suggesting that even though mPFC and amyg/hippo hold flexible statistical knowledge, the reward-driven

associations they also hold were persistent and inflexible. We note that in our hands, BOLD modulations related to rewarded knowledge (rr-cc) were consistent with cross-stimulus suppression effects and were thus smaller for stronger associations. By contrast, BOLD modulations related to representation of statistical knowledge were stronger for more expected transitions. This suggests that these relationships are not stored via shared neural representations because this would have led to a suppressed BOLD signal to the second element.

Pages 19-20

Intriguingly, activity in mPFC and amyg/hippo reflected several statistically dissociable aspects of the task at the same time. On the one hand its activity reflected statistical learning – knowledge of transition frequencies between task states regardless of reinforcement. On the other hand, its activity reflected associations between rewarded sequence elements, regardless of whether elements occurred in the correct or incorrect order, more than it reflected associations between control sequence elements; in contrast to the first type of representation this second type does depend on reinforcement. While the knowledge acquired from statistical learning was flexibly updated and constantly changing in mPFC, as in other areas, reinforcement-based task representations in both amyg/hippo and mPFC were robust and inflexible and did not change over time, even though this knowledge was irrelevant to the task performed during scanning. In summary, RL and statistical learning mechanisms converge in mPFC but have different temporal properties. It has previously been pointed out that learning related representations acquired over different time scales exist simultaneously in different brain regions⁶⁵ and even simultaneously within the same brain region^{66–68}. In the current study, however, the representations are not simply derived from the same learning process operating at different timescales but as a result of different learning processes operating at different timescales.

2. It seems somewhat counterintuitive that a direct comparison of elements from the reinforced sequences should activate typical reinforcement regions such as OFC and medial prefrontal cortex *less* than elements from the control sequences. This result is counterintuitive and critical because the regions for subsequent ROI analysis are identified by this analysis.

We apologize for not explaining this initial contrast more clearly. Perhaps the key issue to clarify first is that the result is essentially a type of repetition suppression result; lower activity in response to the second event suggests that the brain area is processing some aspect of the event that is shared with the first presentation. More precisely, the result makes sense because it is probing BOLD activation to the *repetition* of two rewarded sequence elements, one after the other = rr (compared to the repetition of two control sequence elements = cc). It therefore probes the overlap in the representations of the

different stimuli that are part of the rewarded sequence. If, as the reviewer states, OFC and medial frontal cortex are reward areas, and if they are concerned with representing the contingencies that predict reward in the current experiment then we should expect lower responses to repeated reward elements than to repeated control events.

If overlapping ensembles of neurons encode A, B, C, and D, then – because repeated activation of the same neurons causes suppression in their activation – these neurons will fire less when one of the stimuli A, B, C, or D is presented after having shown one of A, B, C or D just before. Using BOLD, we should see less activity at the time of the second stimulus.

This logic is explained really nicely in this illustration by Barron et al. (Phil Tarns R Soc, 2016) where Z and X share some of the neurons ('representational overlap') which would correspond to e.g. A and B of the rewarded sequence here.

Schematic illustration of the principle underlying fMRI adaptation. (a) The raw BOLD signal measured in conventional fMRI paradigms provides only a measure of the mean activity of the population of neurons within a given voxel. In this example, the raw BOLD signal in response to stimuli X, Y and Z is the same, because the average neural activity within the voxel is comparable for all three stimuli. The raw BOLD signal alone is therefore invariant to the relationship between representations of stimuli X, Y and Z. (b) In fMRI adaptation paradigms, the relationship between different stimulus representations X, Y and Z may be indirectly measured. If stimulus X is preceded by stimulus X (X!X), then the fMRI signal in areas encoding features particular to stimulus X is suppressed. If stimulus X is preceded by stimulus Y (Y!X), the response to stimulus X should not show any suppression, as the representations for X and Y are not overlapping. If stimulus X is preceded by stimulus Z (Z!X), the response in areas encoding the features that are shared between X and Z should show suppression which scales with the amount of overlap between representations of X and Z.

To confirm there was no main effect, we tested for a difference between xR (a RewSeq stimulus not preceded by another RewSeq stimulus but by one of the other eight stimuli) and xC (a ConSeq stimulus not preceded by another ConSeq stimulus). Using our original ROIs, this showed no difference between xR and xC in any of the four ROIs (all $p > 0.1$; shown on the left below), and even at low exploratory thresholds, there were no activations in the vicinity of our ROIs. In summary, this demonstrates that there is no significant deactivation in any of the brain regions that is associated with the rewarded sequence.

Thus, we think that our initial ROI-defining contrast (rr-cc) correctly probes knowledge of the existence of undirected relationships between rewarded sequence elements compared to control sequence elements. We hope that the above explanations have sufficiently clarified this. We have made this point clear in the revised manuscript as follows:

Pages 6-7

To test for differences in BOLD activity between rewarded and control sequence elements, we contrasted **reward-sequence** elements that were preceded by another **reward-sequence** element (i.e., 'rr' pairs) with **control-sequence** elements that were preceded by another **control-sequence** element (i.e., 'cc' pairs). This contrast captures effects of cross-stimulus adaptation to elements from the same sequence **and is intentionally broad and non-directional in the sense that it merely probes knowledge of the existence of undirected relationships between rewarded sequence elements compared to control sequence elements. In other words,** sequence elements included in this contrast were not always shown in the correct order but sometimes backwards or with jumps in-between sequence elements (e.g. CB or BD, see Methods). The identified brain regions therefore encode associations between **rewarded sequence** elements such that elements **part of the rewarded sequence** have become more similar in their neural representation than the corresponding elements of **the control sequence** ('sequence fusion'; **Fig 2a**).

Page 8

In a second test, we confirmed that none of our ROIs were simply showing a main effect difference between rewarded and control elements. We examined rewarded and control elements when they were preceded by an element not part of their own sequence (xr or xc). None of our ROIs showed a significant difference for the contrast xr-xc (all $p > 0.1$; **Suppl Fig 2b**). Thus, our ROI-defining contrast captured relationships between pairs of stimuli as probed by cross-stimulus suppression, rather than differences in the overall BOLD main activation to rewarded and control stimuli.

Fig 6-7, Supplement

Supplementary Figure 2, related to Figure 2: temporal modulation of effects, ROI definition and GLM orthogonality

a, The modulation of BOLD repetition suppression as a function of the temporal gap between two successive RewSeq or ConSeq elements was only present in temporal pole and amyg/hippo, and only for the second of two RewSeq elements (rr) but not the second of two ConSeq elements (cc); left is as shown in **Fig 2**, right is added for completeness. **b**, **The main difference in BOLD response to rewarded and control elements was not significant in any of our ROIs (xr-xc), suggesting that the ROI-defining contrast (rr-cc) indeed captured cross-stimulus repetition suppression effects, and thus probed relationships within the rewarded and control sequences, rather than main activation differences. Main activation differences are tested here based on trials where a rewarded stimulus was preceded by any non-rewarded stimulus (xr; and the same for the control sequence: xc). This test was independent of the ROI-defining contrast.**

3. If I understand correctly, ventral striatum activity follows exactly what one would expect from a region that implements temporal difference (TD) learning (perhaps with the exception that extinction appears to take relatively long. However this may partly be explained by the long acquisition period and by specifics of the instructions for the scanner session, as also suggested by the other findings of persistent reinforcement-related activity). In particular, according to the TD model the earliest element (A) of the reinforced sequence should be encoded (uniquely in case of perfect learning) as predictive of reinforcement and that is exactly what the data show. However, the authors do not seem to interpret (and value) this finding as indicative of learned knowledge. Instead, they focus on negative findings in the ventral striatum (even though negative findings are hard to interpret and non-ventral striatal regions have been involved in both model-based and model-free learning previously (see e.g. the summary figure of the comment by Chen et al., 2015, Journal of Neurophysiology)). Moreover, it is conceivable that spatial proximity to initial elements A is coded more strongly in the striatum than to final elements. In any case, a more balanced interpretation of the ventral striatum findings seems warranted.

We apologize and did not intend to give a misleading presentation of the results in the ventral striatum. Our interpretation of the VS results is, in fact, as far as we can tell, the same as the reviewer's. We agree that the results are entirely consistent with what might be expected from a TD learning account. We were, however, attempting to contrast the activity pattern in VS with that seen in the other brain areas. We have now rephrased that section to reflect the reviewer's comment. We now emphasize that an encoding of A, the first element of the sequence, is also indicative of learnt knowledge, specifically of the first stimulus that is predictive of an upcoming reward.

We did look at the encoding of xA (rewarded – control) not only in VS but at the whole-brain as well, which is reported in Figure 6b and pasted below for convenience (note though that this was an analysis with exploratory thresholds). The strongest bilateral activation was indeed in the putamen (still quite ventral but more lateral than the nucleus accumbens/VS core area). We now comment on that slightly further in light of the diverging views highlighted in the review by Chen et al.

Mask from A (RS-CS)

The revised manuscript section on the ventral striatum was pasted above already (Pages 15-16 in the manuscript, Results section).

4. The authors make a clear-cut distinction between time and space. However, in their task, the two appear to be correlated to some degree by design (it takes longer to move to spatially more distant elements). If the authors cannot exclude this issue they should discuss it.

We are slightly unsure which ‘time and space’ distinction the reviewer is referring to here but we think he/she is referring to Figure 4b where we examine BOLD modulations by spatial distance between two stimuli. This spatial distance regressor is orthogonal to the time interval between two stimuli during scanning (highlights simply jump from one stimulus to the next, and the time of presentation of each stimulus is taken from a truncated Poisson-distribution with mean 2.8s, range 1.5-10s). But if we are not mistaken, the reviewer’s point is that this spatial distance code would likely have emerged during the learning phase when distances and time to travel the distance would have been correlated. We entirely agree with this statement and have added a comment to reflect this important point.

We examined the correlation between the time taken to complete a trajectory and the length of a trajectory (distance in steps). On day 1 of training, correlation coefficients varied from 0.06 to 0.76 (0.12 to 0.84 after rejection of outlier trajectories) across participants, with an average of $\rho=0.50$ ($\rho=0.61$ after outlier rejection). Correlations between trajectory time and length got higher once trajectories became stereotypical on day 2 of training: min 0.33, max 0.79 (min=0.6, max=0.86 with outlier rejection); mean $\rho=0.66$ ($\rho=0.78$ after outlier rejection) across subjects. Thus, indeed, the reviewer is absolutely right in saying that it is difficult to disentangle their contribution which we now clearly state in the manuscript as follows:

Pages 10-11, Results:

We first asked whether information about spatial distances between elements (regardless of whether they comprised **rewarded or control sequence elements**), measured in terms of steps that make up a path on the grid, **was present** in our previously defined ROIs (GLM2, **Suppl Fig 2d; Fig 4a**). **This ‘spatial distance’ contrast was not correlated with the original ROI-defining contrast rr-cc (Pearson’s $r=-0.02$; see also **Suppl Fig 2f**).** BOLD signals in both mPFC and amyg/hippo but not temporal pole or pOFC scaled with distance, with stronger responses for stimuli that were closer in space to the previous stimulus ($p(\text{amyg/hippo})=7.91e-6$, $t(25)=-5.6$; $p(\text{mPFC})=7.54e-6$, $t(25)=-5.62$; $p(\text{pOFC})$ and $p(\text{tempPole})>0.1$; mPFC global peak: 0,50,-14, $z=5.0$; peak within mPFC spherical ROI: 0,36,-6, $z=4.01$; amyg/hippo global peak lies within ROI sphere: 30,-4,-24, $z=5.26$; **Fig 4b-c**). At the whole brain, spatial relationships were reflected in the activity of an extended network of brain regions which included mPFC and the amygdala-hippocampus border (all cluster-corrected; Supplementary Results and **Suppl Fig 4a**). **It is worth noting that during the pre-scan learning task, larger spatial distances were associated with longer travel times (correlation between trajectory time and spatial distance Pearson’s $r=0.5$ and $r=0.66$ on days 1 and 2 respectively of the pre-scanning learning task), and thus the experience of both time and distance could have contributed to the formation of this neural representation.**

5. The summary appears to lack a concluding interpretative sentence

Apologies, we have added such a concluding a sentence. The summary now reads as follows:

Learning the structure of the world can be driven by reinforcement but also occurs incidentally through experience. Reinforcement learning theory has provided insight into how prediction errors drive updates in beliefs but less attention has been paid to how the knowledge resulting from such learning is encoded. Here we contrast associative structures formed through reinforcement and experience of task statistics. BOLD neuroimaging in human volunteers demonstrated temporal pole and posterior orbito-frontal cortex **represent** sequences of **events leading to reward, which were constructed backwards from reward, in a robust and inflexible manner**. By contrast, medial prefrontal cortex and a hippocampal-amygdala border region not only carry **reward-related knowledge** but also flexible statistical knowledge of the currently relevant task model. Intriguingly, ventral striatum **exhibits prediction error responses reflecting knowledge contained in such representations but it does not itself contain the full RL-derived or statistically derived knowledge**. **In summary, representations of task knowledge are derived via multiple learning processes operating at different time scales that are associated with partially overlapping and partially specialized anatomical regions.**

6. Did the authors also test participants' ability to produce the correct control sequence after each pre-scan block?

No we didn't. In fact, participants were not informed about the presence of the control sequence and there was no signaling of when it started and ended so the only way to figure it out was through repeated observation of consistencies in the transitions. From debriefing participants at the end of the experiment, anecdotally, they had no awareness of the control sequence but we have no data to back this up with.

Therefore, all measures proving **knowledge of the control sequence** are implicit. As mentioned above in the response to point 1 (Part 3), the initiation of trajectories became faster as participants moved from A'B' to B'C' and C'D' showing that there was some anticipation of upcoming transitions (ANOVA performed only on the 'grey' control bars in Figure 1c; significant effect of transition: $F(2,50)=28.553$, $p<0.001$; model with factor transition: $BF_m=1.522e+6$, $P(M|data)=1$). As mentioned and pasted above this has now been included in the manuscript. It strengthens the point that some learning occurred even in the control sequence and therefore in the absence of reward and via observed statistical relationships.

In addition to the above new paragraph on page 6, the corresponding figure legend (Fig 1, p. 34) reads:

c, Participants RTs when initiating the first movement towards the next highlighted element were faster as they progressed through **the rewarded sequence** (i.e., from A to B, B to C and C to D), and speeding was more pronounced for **the rewarded** compared to **the control sequence**. Thus, participants anticipated the correct sequence successor elements. **Looking at ConSeq transitions on their own (grey bars), there was a significant effect of transition, suggesting the presence of statistical learning in the absence of reward.**

7. Conceptually, it does not seem to make much sense to combine cluster-level correction with small volume correction/ROI analyses. Voxel-level correction should be used instead

Sorry about the confusion. The reviewer is referring to two sentences relating to results in the amygdala which specified that for the initial ROI-defining contrast (rr-cc) and the spatial spread (neighbours of D) contrast, amygdala is only significant using small-volume correction.

We stated “cluster-level $p < 0.05$ using small-volume correction ... ” in brackets for both of those results which it can be argued does not make sense.

To address this concern, we now report two additional pieces of information.

First, for the initial ROI-defining contrast (rr-cc), we performed nonparametric threshold-free permutation tests using TFCE (threshold-free cluster enhancement method; Smith & Nichols, 2009). Eight voxels within the amygdala survived $p < 0.05$.

Second, we defined an ROI (3mm sphere) around the average peak coordinate reported in an fMRI study showing temporal spread of reward in the human amygdala (Jocham et al.). We hypothesized that a spatial spread response might occur in a similar location within the amygdala. Indeed, in this sphere, the spatial spread contrast was significant ($t(25) = 2.30$, $p = 0.03$).

Second, to be fully transparent, we now also report the peak z-values and cluster extent at uncorrected levels. For the initial contrast (rr-cc) the peak in the amygdala is $z = 2.8$ within the amygdala anatomical mask and $z = 2.97$ in the same cluster but just adjacent reaching into the hippocampus. The cluster extent at $p < 0.01$, $z > 2.3$ is 27 voxels within the amygdala mask in the right hemisphere, and 46 in the right and 11 in the left hemisphere when considering voxels of the same cluster within either amygdala or adjacent hippocampus, i.e. including the border.

For the spatial spread contrast, the strongest activation lies within the anatomical amygdala mask and is $z = 3.0$ (precisely, $z = 2.9976$). The cluster extent at $p < 0.01$, $z > 2.3$ is 19 voxels. This is clarified in the revised manuscript as follows:

Page 7:

Several regions showed differences in the neural representation of **rewarded** compared to **control sequence** elements (rr-cc): (a) medial prefrontal cortex (mPFC), spanning parts of areas 32 and 14m, (b) temporal pole (tempPole), and within the same extended cluster (c) posterior orbitofrontal cortex (pOFC), comprising posterior area 47/12 and aspects of anterior insula (all cluster-level corrected; peak MNI coordinates mPFC: 0,40,-6, z=4.0; tempPole: 48,10,-28, z=4.19; pOFC: 30,18,-22, z=3.73; **Figs 2a,b**; for a complete table of activations see Table S1). Because of *a priori* hypotheses about the role of the amygdala (see Introduction) and because the size of this region means that it is more difficult for it to survive whole-brain cluster-correction, we also examined the same contrast within the amygdala and found a significant activation in right amygdala (peak coordinate: 28,-6,-22; **z=2.97; p<0.05 using threshold-free cluster enhancement (TFCE³² in a small volume containing** left and right amygdala; **Fig 2a**).

Pages 14-15:

Establishing eligibility through spread of reward in space

Tracking state transitions as well as representing learned sequence-reward associations is essential for forming an appropriate representation of the external world. However, sometimes rewards can influence the response to stimuli that were not actually predictive of the reward but which simply occurred close in time to the reward^{28,41}. In some cases reward changes the response not just to the stimuli that preceded reward but even to the stimuli that followed close in time after the reward. Activity in the amygdala has been related to the spreading of reward to proximal elements in time in bandit-type tasks in which rewards on subsequent trials are logically independent from those on the current trial. However, in our experiment, states are not only experienced sequentially, but they have a spatial relationship on the grid shown to participants. This enabled us to test whether we could see neural signatures of a spatial spread of reward. In other words, we asked whether the amygdala showed specific changes in activity to stimuli that were simply close in space to the location where reward had occurred during the pre-scan learning task. We defined any element that could be reached by one step (either horizontally or vertically) as ‘neighbors’ and compared activity for neighboring elements of D with the neighboring elements of D’ (the final elements of **the rewarded and control sequences; Figs 5a,b**). **This contrast was examined in an amygdala ROI previously associated with temporal spreading of reward (see Methods,⁴²).** Consistent with our hypothesis, we found **evidence for** a stronger response to neighbors of D compared to neighbors of D’ in the amygdala (**p=0.03, t(25)=2.30; peak in left amygdala: -18,-8,-16, z=3.0; ; note that the amygdala was the strongest peak at the whole-brain**). **It is important to note that the number of neighbors of the final element was exactly matched, partly by counterbalancing across participants, for the reinforced and non-reinforced sequences. This ensured that the difference**

between rewarded and non-rewarded sequences was not confounded with any difference in spatial uncertainty.

Page 46-47

Because of *a priori* hypotheses, significance in the amygdala was established using a mask. For the initial ROI-defining contrast rr-cc we had no strong hypotheses about the location within the amygdala, so we applied non-parametric threshold-free cluster enhancement (TFCE;³²) to establish significance at $p < 0.05$ within an anatomical mask of bilateral amygdala (Harvard Subcortical Atlas). For establishing effects of spatial spread, we used the average amygdala coordinate from⁴² ($x = -21, y = -6, z = -19$) and tested for significance in a 3mm sphere around this coordinate. For full transparency, we also report the cluster extent at uncorrected levels: for the initial contrast (rr-cc) the peak in the amygdala was $z = 2.8$ within the amygdala anatomical mask and $z = 2.97$ in the same cluster but just adjacent reaching into the hippocampus. The cluster extent at $p < 0.01, z > 2.3$ was 27 voxels within the amygdala mask in the right hemisphere, and 46 in the right and 11 in the left hemisphere when considering voxels of the same cluster within either amygdala or adjacent hippocampus, i.e. including the border. For the spatial spread contrast, the strongest activation was located within the anatomical amygdala mask at $z = 3.0$. The cluster extent at $p < 0.01, z > 2.3$ is 19 voxels.

8. Was the number of neighbors of the final element exactly matched for reinforced and non-reinforced sequences? At least the example in figure 1 suggests otherwise. A potential confound with an incomplete match would be a difference in (spatially defined) uncertainty

That's a very good point. We had three configurations ('contexts') of rewarded and control sequence. The reviewer is correct that in an individual configuration, like the one in Figure 1, D of the rewarded sequence had four neighbours because it was located in the centre of the 3x4 layout and D' of the control sequence only had three neighbours because it was on an edge.

In the three configurations, these were the numbers of neighbours (see also Fig S1b):

Context 1	D (middle): 4	D' (edge): 3
Context 2	D (edge): 3	D' (edge): 3
Context 3	D (corner): 2	D' (corner): 2

So there was only one mismatch which thus affected a third of the data. Importantly, though, we swapped the rewarded and control sequences in half the participants ('group 1' and 'group 2'), so we

don't think this can explain any of our results. In the revised manuscript we have clarified this in the Methods as shown in the paragraph pasted above (pages 14-15 of the manuscript).

9. In the last results paragraph, the sentence “The encoding of the first predictive element A was not specific to ventral striatum” seems to be a bit lost.

We have removed this sentence and rewritten several parts of this paragraph based on some of the previous comments by this reviewer. The updated ventral striatum section was pasted above and can be found on pages 15-16 of the manuscript.

We would like to thank this reviewer again for the many helpful suggestions and we hope that the reviewer will find that the changes we have made have improved the manuscript and led to a more balanced and fair presentation of our results.

Reviewer #3 (Remarks to the Author):

In this manuscript by Miriam Klein-Flugge et al., the authors explore how cognitive maps are formed through reinforcement and updated through experience in humans. Specifically, the authors state three goals: to explore 1) the neural signature of knowledge learned through reinforcement, 2) the neural mechanisms that track statistical relationships over time, and 3) whether the eligibility of reward might be assigned to stimuli in close spatial proximity to reward (a similar process has been shown for stimuli temporally close to reward).

Through a carefully designed sequence learning task where the start of the sequence was not signaled to the participant, the authors confirmed a central prediction of a popular computational model of reinforcement learning named TD RL: sequence knowledge is constructed backwards from reward. Using an impressively thought-out and carefully designed task during MRI scanning, the authors were able to test the neural signature of sequence knowledge learned during the previous task independent of task demands. The authors identified four brain regions that coded for reinforced sequence knowledge: medial prefrontal cortex (mPFC), posterior orbitofrontal cortex (pOFC), temporal pole, and a region at the border of amygdala and hippocampus. More detailed and careful analyses revealed that the path to reward (elements in the correct sequence) was coded by temporal pole and pOFC as an inflexible map, whereas knowledge about the statistics of transitions between elements of the task space (i.e., a more flexible map of task space) were coded by mPFC and amygdala/hippocampus. A more central amygdala region responded to elements that were spatially adjacent to the element previously associated with reward during training, complementing previous research that uncovered a role for the amygdala in spreading of reward to elements that were spatially proximal to reward. None of these regions coded reward information directly, suggesting that the ventral striatum may be involved in forming the knowledge about rewarded sequences, but did not carry a map of that knowledge in the absence of reward.

I find the paper timely, important, and of interest to a wide readership. There has been a lot of interest in better understanding how cognitive maps are formed during learning and used during decision making. This study is meticulous, the authors do a good job at presenting complex analyses clearly, and the results confirm some clear predictions and present some new intriguing effects (e.g., involvement of temporal pole).

I recommend publication. I have only minor points to bring up:

1) I was surprised that the authors stated that they had no a priori reason to look at the hippocampus. The authors mention cognitive maps as described by Tolman. A rich literature on cognitive maps implicates the hippocampus, hence my surprise. If the authors would elaborate on why they had no a priori reason to look at the hippocampus, that would be useful to readers like me.

Thank you very much for this comment. In hindsight, we agree, that perhaps we should have expected effects to arise in the hippocampus. When we originally designed the experiment, one of our key interests was in the formation and encoding of rewarded sequence knowledge. Because of several previous papers on 'spread of reward' in time (e.g. Walton et al., Chau et al.; Jocham et al.), we thought the amygdala might be important in the acquisition of this type of knowledge by spreading reward in time or space.

By contrast, with respect to the question of ‘map-like’ knowledge we were mostly focused on the medial PFC (this hypothesis reflects our laboratory’s long-term interest in the medial PFC but it is also consistent with recent reports from Constantinescu et al. Science; Schuck et al., Neuron.) Nevertheless, we agree that the reviewer is completely right that we should have added the hippocampus to our set of *a priori* ROIs. We now find ourselves in a somewhat unusual position in that while a good argument can be made as to why this might have been an *a priori* ROI, in actual fact it was not. In order to address this point we now present a summary of the results of analyses similar to those used elsewhere in the manuscript but now applied to additional hippocampal ROI based on the work of Garvert and colleagues (eLife; 2017). This material is noted in the supplementary information for **Suppl Fig 6** and in addition it is referred to in the Discussion as follows:

Page 18:

Our second main finding related to BOLD signatures of statistical knowledge in mPFC and an area at the interface of amygdala and hippocampus (**Fig 4 and Suppl Fig 4; Fig 7b; activity patterns from an adjacent hippocampal region studied by⁶² are reported in Suppl Fig 6 and show that BOLD activity in hippocampus reflected statistical knowledge but did not distinguish between rewarded and control sequences**).

Supplement, page 8:

Supplementary Figure 6, related to Figure 3, 4 and 6: BOLD signatures of statistical learning in the hippocampus

The hippocampus showed effects of transition frequency (**a**) and spatial distance (**b**); contrasts are the same as in **Suppl Fig 4a** but with views showing the hippocampus (both $z > 3.1/p < 0.001$ cluster-corrected). **c**, All effects in the hippocampus are illustrated for a spherical ROI centered on a coordinate taken from Garvert et al., eLife, 2017. This shows BOLD signatures related to statistical learning but no knowledge of the rewarded over and above the control sequence.

2) The authors make a strong argument for the inflexibility of correct sequence mapping in pOFC and temporal pole. The parameter estimates plotted in Figure 3d seem variable across blocks, with perhaps an effect of day (block 1-3 vs block 4-6). I appreciate the authors' attempts to support the null using Bayes factors, but the methods include few details of what priors were used in this test. Inclusion of these details is appreciated.

Yes, that's a very good point. We agree that the analyses that split the data across blocks are inherently a bit noisier and more variable, and particularly the one the reviewer refers to, Figure 3d.

We apologize for not specifying the priors. We used JASP's default prior scales (Bayarri et al 2012; Love et al., 2015; Morey & Rouder, 2015; Rouder et al. 2012).

Prior		Samples	
r scale fixed effects	0.5	<input checked="" type="radio"/> Auto	No. samples 10000
r scale random effects	1	<input type="radio"/> Manual	
r scale covariates	0.354		

We also checked whether there was an effect of day in a 2 x 3 Anova with factors Day and Block, run separately on the data from pOFC and temporal pole for the correct order contrast shown in Figure 3d that the reviewer is referring to above (pasted below for reference). As shown below, the null model was favoured in both regions, but there was a trend for an effect of Day in pOFC ($F(1,25)=3.7, p=0.065$). We append the results below for the reviewer's information but think they are maybe not conclusive enough to warrant inclusion in the manuscript.

pOFC

Within Subjects Effects

	Sum of Squares	df	Mean Square	F	p
Day	1.509e +6	1	1.509e +6	3.714	0.065
Residual	1.016e +7	25	406271.040		
Block	1.773e +6	2	886707.392	1.254	0.294
Residual	3.535e +7	50	706971.488		
Day * Block	262931.714	2	131465.857	0.105	0.900
Residual	6.257e +7	50	1.251e +6		

Note. Type III Sum of Squares

Between Subjects Effects

	Sum of Squares	df	Mean Square	F	p
Residual	2.135e +7	25	853923.121		

Note. Type III Sum of Squares

Bayesian Anova

Model Comparison

Models	P(M)	P(M data)	BF _M	BF ₁₀	error %
Null model (incl. subject)	0.200	0.602	6.061	1.000	
Day	0.200	0.246	1.307	0.409	0.799
Block	0.200	0.103	0.462	0.172	2.575
Day + Block	0.200	0.042	0.177	0.071	2.749
Day + Block + Day * Block	0.200	0.005	0.021	0.009	2.352

Note. All models include subject.

Temporal pole

Within Subjects Effects ▼

	Sum of Squares	df	Mean Square	F	p
Day	1.239e +6	1	1.239e +6	1.290	0.267
Residual	2.402e +7	25	960666.783		
Block	3.892e +6	2	1.946e +6	2.313	0.109
Residual	4.208e +7	50	841514.375		
Day * Block	2.615e +6	2	1.308e +6	1.202	0.309
Residual	5.438e +7	50	1.088e +6		

Note. Type III Sum of Squares

Between Subjects Effects

	Sum of Squares	df	Mean Square	F	p
Residual	1.405e +7	25	561829.919		

Note. Type III Sum of Squares

Bayesian Anova

Model Comparison

Models	P(M)	P(M data)	BF _M	BF ₁₀	error %
Null model (incl. subject)	0.200	0.507	4.107	1.000	
Day	0.200	0.169	0.814	0.334	1.843
Block	0.200	0.223	1.145	0.439	0.997
Day + Block	0.200	0.075	0.323	0.147	5.227
Day + Block + Day * Block	0.200	0.027	0.112	0.054	4.796

Note. All models include subject.

We have therefore just revised the manuscript so that the Bayes factor analysis that was already employed is now explained more clearly (including description of priors) as shown below (pages 9-10):

Notably, unlike in the pre-scan learning task, knowledge of **rewarded sequence** was irrelevant for participants during the scan task. Performance in the scan task simply depended on responding to catch trials. We therefore tested whether the correct sequence order effects reported above decreased as the scanning session progressed. Contrast estimates for correct sequence order (**RewSeq vs ConSeq**) were extracted from pOFC and temporal pole, separately for each block. Because frequentist statistics cannot provide evidence in favor of the null hypothesis, we conducted a 2x6 Bayesian repeated-measures ANOVA with factors ROI (pOFC and tempPole) and block. **This JZS Bayes factor ANOVA^{33,34} with default prior scales revealed that the Null Model was** the best model and almost six times more likely than any other model (BF_M=5.81, P(M|data)=0.59), including a model with an effect of Block (BF_M=1.43, P(M|data)=0.26). Thus, representations of the correctly ordered reinforced sequence in temporal pole and pOFC were remarkably inflexible and long-lasting over the duration of our experiment despite changing task demands. **In summary, the representations of sequences that originally led to reward in the initial training period appear almost to have been ‘frozen’ in a stable state in pOFC and tempPole. It**

is possible that they are unchanged because no new reward-learning occurred during the scanning period that could interfere with the originally formed representations.

3) The authors mention a second control sequence in the methods, but include no analyses of this CS2. What was the rationale behind this second control sequence and can the authors offer justification for not including any analyses of this CS2, even in the supplement.

Yes, that's a very good point. CS2 was unrewarded, like CS, but less frequent than CS. When designing the task, we thought of CS2 as a 'frequency control' for CS in which we would expect less pronounced statistical learning because CS2 transitions were only experienced half as many times as CS transitions. While this initially seemed a sensible approach, in retrospect, of course, the low frequency of the CS2 means that it is difficult to obtain an accurate estimate of the neural activity with which it is associated. A direct comparison between CS1 and CS2, therefore, is considerably lacking in power. Instead the model-based transition analysis that included all stimuli (Fig 4) is unaffected by the lack of power for any analysis employing CS2-related data and therefore remains a powerful analysis. We have included a sentence explaining this in the Methods (page 39).

Originally, we had planned to compare ConSeq2 to ConSeq to probe statistical learning, or in other words ConSeq2 served as a frequency control for ConSeq. Because of the rare occurrences of ConSeq2, however, this analysis was lacking in power and model-based analyses that included all stimuli seemed more appropriate to investigate statistical learning (see below).

4) Do the authors have any behavioral data to test the spatial spreading of reward? For example, did the participants choose between a D-adjacent element and a non-rewarded element during the choice task?

Again, we agree this is a very interesting question. In our experience, for behavior to show 'spreading of reward' effects in time, the task needs to be made very hard (e.g. see Jocham et al., Neuron, 2016 where the standard trial structure of most behavioural tasks was abandoned to sufficiently confuse participants in order to make such effects appear in their behaviour). That's probably because enough 'contingent' mechanisms are in place in a healthy brain that make sure spread of reward signals do not (wrongly) influence behavior in a contingent task – but spread of reward does appear in behavior in animals with lesions to OFC (Walton et al., Neuron, 2010). We would expect the same to be true for spatial spreading of reward, but there is currently no data available on this as far as we are aware.

As suggested, we went back to the choice data collected after each block of the pre-scan learning task and checked if we had trials where participants made choices between a neighbor of D and a non-RS element. We had an average of eleven such trials per choice block (in a total 12 blocks per day) but choices did not differ significantly from chance ($p > 0.4$ on both Day1 and Day2 of training), not even when looking at only the very first block of training. Similarly, there was no RT speeding on neighbours of D compared to neighbours of D' during the flower task performed in the scanner ($p = 0.14$). Overall,

this is consistent with our intuition that contingent learning dominates behavior. We have noted this in the revised manuscript as follows (page 15):

We next examined whether the spread of reward effect observed in the amygdala might be reflected in the behaviour of the participants. To do this, first, we examined the choice data collected after each block of the pre-scan learning task and checked if we had trials where participants made choices between a neighbour of D and a non-RS element. We had an average of eleven such trials per choice block (in a total of 12 blocks per day) but choices did not differ significantly from chance ($p > 0.4$ on both Day1 and Day2 of training), not even when looking at only the very first block of training. Similarly, there was no RT speeding of responses to neighbours of D compared to neighbours of D' during the task performed in the scanner ($p = 0.14$). We therefore next examined whether the spread of reward effect in the amygdala might be related to the strength of BOLD representation of the rewarded sequence. One hypothesis is that sequence knowledge is encoded using a distinct mechanism, which would imply people who strongly respond to relationships between elements of the rewarded sequence (more suppression for rr compared to cc) might have a weaker code for spatial spread (positive correlation because of sign of repetition suppression). Alternatively, if these mechanisms work hand in hand, then the two signatures might correlate negatively. Within the amygdala region defined above, we found a negative correlation between the sequence fusion (rr-cc) and spatial spread of reward, implying more suppression for rewarded sequence elements in people with a more pronounced spatial spread signal. This effect was only significant in the first block of scanning, i.e. when learning effects were the most pronounced (p values corrected for multiple comparisons: $p = 0.0036$ for block 1, all other blocks: $p > 0.15$; **Fig 5c**). In summary, a spread of reward effect is apparent in amygdala and it is related to the representation of RL knowledge in the same structure. Therefore, the spread of reward signal may exist to guide the construction of a more refined representation of sequence associations by motivating behavior in the vicinity of a potential reward. The precise order representations of a detailed route to reward in temporal pole, and pOFC and representations of overall task space in mPFC would be needed to guide an agent from faraway points in the environment or state space.

5) Finally, I would recommend that the authors make their statistical fMRI maps publicly available, for example through neurovault.org. This will allow the reader to have a full picture of the exploratory analyses and can facilitate future meta-analyses.

Yes – thank you, we fully agree. It was indeed what we had planned to do upon publication of this manuscript. We have clarified that this is the case by adding the following sentence to the Data availability statement (page 49):

Data availability

The data underlying all figures are provided as Source Data files (‘.mat’). The source data folder includes a spreadsheet detailing the names of the source data files and scripts relevant for producing each figure (‘make_figX.m’). Unthresholded fMRI maps of all contrasts will be available on Neurovault upon publication.

Reviewers' Comments:

Reviewer #1:

Remarks to the Author:

The authors have completed a very thorough revision indeed and grappled thoughtfully with the various reviewer considerations, such that the interesting results in the paper shine through more clearly. I have remaining only one somewhat extended presentational/interpretational suggestion, and two relatively minor and localized methodological quibbles.

1. I think I have come to understand the authors' argument about the independence of the different analyses and effects, albeit with some effort which I hope I can help to minimize for future readers. The authors view the initial rr-vs-cc effect as a distinct phenomenon from the other accompanying effects. However, my own inclination as a reader (at least as a starting point) was to try to deduce a single description of the overall activity pattern, net of all these effects. After all, they all come down to responses to different stimuli preceded by other stimuli, compared various ways – so they could all be thought of as different levels of cross stimulus suppression, modulated by the various factors. The authors may be correct that the best conception of the overall pattern is instead as the superimposition of two separate patterns (e.g. one undirected and suppressive, vs another directed and excitatory). And indeed, various lines of evidence ultimately emerge that this is the right way to view things (mainly p. 12 line 386 and p. 21 line 632). But they emerged slowly enough that I got myself tangled up trying to figure things out in the meantime. So I wonder if more signposting, prefiguring, and handholding might be helpful for future readers.

The kind of reasoning I found myself going through is: the response to C following B is reduced compared to the response to C' following B' (this is the RR vs CC effect), but in temporal pole/pOFC this suppression effect is *attenuated* (ie the response to C is larger / less suppressed) when C is preceded by AB (Figure 3C – parenthetically, I don't get how this ultimately is consistent with the story from Supp Fig 2E since these ABC cases are among the forward pairs and should drive the CSS effect down relative to the "other" pairs). Analogously, in the other areas we could say CSS is attenuated for short spatial distances and frequent transitions. I think there is a good argument (p. 12) why this is the wrong way to think about things (because you would expect suppression to be *enhanced* in those situations) but it arrived late enough that I had already gotten frustrated trying to keep track of positives and negatives. So maybe this point could be frontloaded and emphasized more.

2. Many results in the paper concern followup analyses on ROIs defined by extremizing the initial RR-vs-CC contrast, which always raises concern about bias due to double dipping. For most of these it is argued that the new tests are orthogonal and thus presumably unbiased. But I'm worried that at least a couple of these tests are biased. In particular I think the results in Supp Fig 2e and also the test on page 9-10 excluding extinction are likely biased by the ROI-defining contrast (in each case, in favor of the null, which is what is affirmed). Suppose, as a thought experiment, that "real" RR vs. CC activity was driven by only a subset of transitions (e.g. forward ones, for Figure 2E or early ones for the extinction test), which is correlated with the testing contrast and could drive the initial result. Then searching to maximize the testing contrast would tend to identify voxels where the noise tended to match the remaining parts of the contrast regressor (RR vs CC effects for backward transitions or late transitions), which would then erroneously favor the null in the two tests. This is important, in part, because the temporal (non) effect is the other empirical evidence the authors point to for concluding the suppression effects are distinct from the effects in Figure 4d.

Finally, I don't get the logic of the new test under "Behavioral evidence for statistical learning," which seems to argue subjects learn the control sequence because their reaction times differ from B' to C' to

D'. I don't see why it is the case that RT should progressively speed up along this sequence. Aren't they all equally expected once A' is observed?

Reviewer #2:

Remarks to the Author:

The authors have adequately addressed my points. Thank you for a responsive revision.

Reviewer #3:

Remarks to the Author:

I commend the authors on their thorough efforts to address the reviewers' comments. I find the study interesting and compelling. The revised manuscript contains much rich information that I think will be of interest to the readership. I recommend publication.

Reviewer #1

The authors have completed a very thorough revision indeed and grappled thoughtfully with the various reviewer considerations, such that the interesting results in the paper shine through more clearly. I have remaining only one somewhat extended presentational/interpretational suggestion, and two relatively minor and localized methodological quibbles.

Thanks very much for your positive assessment of our first revision and for again bringing up some helpful comments which we will address below.

1. I think I have come to understand the authors' argument about the independence of the different analyses and effects, albeit with some effort which I hope I can help to minimize for future readers. The authors view the initial rr-vs-cc effect as a distinct phenomenon from the other accompanying effects. However, my own inclination as a reader (at least as a starting point) was to try to deduce a single description of the overall activity pattern, net of all these effects. After all, they all come down to responses to different stimuli preceded by other stimuli, compared various ways – so they could all be thought of as different levels of cross stimulus suppression, modulated by the various factors. The authors may be correct that the best conception of the overall pattern is instead as the superimposition of two separate patterns (e.g. one undirected and suppressive, vs another directed and excitatory). And indeed, various lines of evidence ultimately emerge that this is the right way to view things (mainly p. 12 line 386 and p. 21 line 632). But they emerged slowly enough that I got myself tangled up trying to figure things out in the meantime. So I wonder if more signposting, prefiguring, and handholding might be helpful for future readers.

The kind of reasoning I found myself going through is: the response to C following B is reduced compared to the response to C' following B' (this is the RR vs CC effect), but in temporal pole/pOFC this suppression effect is *attenuated* (ie the response to C is larger / less suppressed) when C is preceded by AB (Figure 3C – parenthetically, I don't get how this ultimately is consistent with the story from Supp Fig 2E since these ABC cases are among the forward pairs and should drive the CSS effect down relative to the "other" pairs). Analogously, in the other areas we could say CSS is attenuated for short spatial distances and frequent transitions. I think there is a good argument (p. 12) why this is the wrong way to think about things (because you would expect suppression to be *enhanced* in those situations) but it arrived late enough that I had already gotten frustrated trying to keep track of positives and negatives. So maybe this point could be frontloaded and emphasized more.

Thanks very much for this point. This perspective is quite helpful to work out where we can still be clearer to communicate the different signs of our effects and make it easier for the reader to follow.

Firstly, to summarize again in our own words all the analyses relating to successive stimuli:

(1) AB, BC, CD, AC, AD, BD, BA, CB, DC, CA, DB, DA are all RR pairs for which overall, there is *less activity* and thus suppression for the second element (a repetition within the rewarded sequence), compared to the equivalent pairs of the control sequence (this is what you refer to above as 'undirected and suppressive') → this effect is present in pOFC, temporal pole, mPFC and amyg/hipp; **Figure 2/S2**

(2) ABC and ABCD but not xBC and xxCD show an excitatory 'build up' effect not present in the control sequence. During training, transitioning through ABCD coincides with a build-up of reward-expectation (and thus this effect you refer to correctly as 'directed and excitatory'). This excitatory effect seems to be stronger than the suppression effect, otherwise we would not be able to detect it → this effect is present in pOFC and temporal pole; **Figure 3/S3**

(3) For pairs with short spatial distance or high frequency of transition, activity to the second element of the pair is enhanced/stronger (compared to larger spatial distance or less likely transition) → this effect is present in mPFC and amyg/hippo; this is the only effect that goes in the opposite direction to what we expected, which we comment on; **Figure 4/S4**

To briefly comment on the point raised in parentheses, we appreciate this is a long manuscript but please note that we tried to explain in the supplement that the bar shown for forward pairs in Figure 2E exclude occurrences of ABC and ABCD to keep things simple. Because ABC/ABCD are a small subset of all possible forwards pairs, the effect in Figure 2E would not entirely disappear if ABC/ABCD were included but as rightly noted by the reviewer, it would be weakened by the positive effect in ABC/ABCD which is why we thought it is easier to keep things simple and leave those two conditions out of this analysis. See paragraph in Supplement on page 3 (unchanged):

“To ensure that the observed effects were not driven by only the correctly ordered forwards pairs, we repeated the analysis but split ‘rr’ and ‘cc’ pairs into repetitions that were forwards-directed (‘ForwPair’: AB, xBC, xxCD) and other within-sequence pairs that were not forwards-directed in the correct order but still transitions within the same sequence (‘OtherWithinPair’: BA, CB, DC, AD, DA, AC, CA, BD, DB). **Suppl Fig 2e** shows the effects separately for these two sub-groups of trials. Note that forwards pairs exclude occurrences of ABC and ABCD, i.e., those where full third- or fourth-order sequence relationships were fulfilled.”

We have now **added a supplementary table (Table S2)** that explains the three effects summarized above including their signs and the pairs that went into each analysis and we refer to it early on, to aid the reader.

Pairs of stimuli included in analysis	Contrast	Sign of effect	Regions in which this effect is observed	Figures
AB, BC, CD, BA, CB, DC, AC, AD, BD, CA, DA, CA (and equivalent ones for the ConSeq)	RR-CC	Smaller BOLD for repetitions within the RewSeq compared to the ConSeq (consistent with greater repetition suppression within RewSeq)	tempPole pOFC mPFC amyg/hippo	Figure 2/S2
AB, ABC and ABCD vs A'B', A'B'C' and A'B'C'D'	Correct Order RewSeq-ConSeq	Stronger BOLD for RewSeq compared to ConSeq consistent with build-up of reward expectation/reward proximity	tempPole pOFC	Figure 3/S3
Pairs with close vs far spatial proximity Pairs with frequent vs infrequent transition probability	spatDist (parametric) transFreq (parametric)	Stronger BOLD for spatially close or likely transitions (not consistent with repetition suppression)	mPFC amyg/hippo	Figure 4/S4

We have also added some clarifications in various places in the text:

Page 6: “To test for differences in BOLD activity between rewarded and control sequence elements, we contrasted reward-sequence elements that were preceded by another reward-sequence element (i.e., ‘rr’ pairs) with control-sequence elements that were preceded by another control-sequence element (i.e., ‘cc’ pairs). **In this contrast, we expected suppression, and thus a smaller BOLD signal, for repeated occurrence of rewarded sequence elements (rr), compared to control sequence elements (cc) [...]** In other words, sequence elements included in this contrast were not always shown in the correct order but sometimes backwards or with jumps in-between sequence elements (e.g. CB or BD, see Methods, **see Supplementary Table 2 for all pairs of stimuli included in this analysis**).”

Page 8/9: “We therefore reasoned that a BOLD signature of the correctly ordered rewarded sequence should also be present. A brain region in which reward is back-propagated to previous stimuli, for example via TD learning, should demonstrate a particular sensitivity for the correctly ordered stimulus sequence that has led up to reward in the past. **Importantly, we expected such a signature to get stronger as reward is approached, thus going in the opposite direction to the suppression effects reported for undirected pairs.**”

Page 10: “**We expected suppression of BOLD, and thus weaker signals for nearby or likely transitions. However, note that the effects described below went in the opposite direction (see also Supplementary Table 2).**”

Page 12: “By contrast, BOLD modulations related to representation of statistical knowledge were stronger for more expected transitions (**Supplementary Table 2**). **Thus, the sign of the effect was opposite to our prediction.**”

2. Many results in the paper concern followup analyses on ROIs defined by extremizing the initial RR-vs-CC contrast, which always raises concern about bias due to double dipping. For most of these it is argued that the new tests are orthogonal and thus presumably unbiased. But I’m worried that at least a couple of these tests are biased. In particular I think the results in Supp Fig 2e and also the test on page 9-10 excluding extinction are likely biased by the ROI-defining contrast (in each case, in favor of the null, which is what is affirmed). Suppose, as a thought experiment, that “real” RR vs. CC activity was driven by only a subset of transitions (e.g. forward ones, for Figure 2E or early ones for the extinction test), which is correlated with the testing contrast and could drive the initial result. Then searching to maximize the testing contrast would tend to identify voxels where the noise tended to match the remaining parts of the contrast regressor (RR vs CC effects for backward transitions or late transitions), which would then erroneously favor the null in the two tests. This is important, in part, because the temporal (non) effect is the other empirical evidence the authors point to for concluding the suppression effects are distinct from the effects in Figure 4d.

Thanks for this point. Yes, when a test is orthogonal to the one used for defining the ROI, it cannot bias the subsequent test. This is why we tried to emphasize, in the previous round of revision, that the tests that follow the initial ROI-defining contrast are orthogonal to it which is clearly very important.

We can see that this reviewer is not convinced this is true for all tests, so we will try and clarify our argument further:

(1) **Supplementary Figure 2e** was created in order to probe this reviewer's concern about the RR-CC suppression being caused by a subset of pairs, and indeed because we extracted the ROIs from the RR-CC contrast and are now splitting that same contrast into two of the subsets of pairs that contribute to it, the reviewer is right in noticing that we are biased (equally for all subsets) towards again seeing the RR-CC difference, i.e. the difference between blue and grey bars. However, critically, the difference between the two subsets ForwPairs and OtherPairs is **not** biased. In somewhat more formulaic terms, that is because the original contrast was

$(\text{ForwPairsRwSeq} + \text{OtherPairsRwSeq}) - (\text{ForwPairsConSeq} + \text{OtherPairsConSeq})$
[contrast: 1 1 -1 -1]

So both ForwPairsRwSeq -ForwPairsConSeq and OtherPairsRwSeq – OtherPairsConSeq are biased – but crucially, they are not individually assessed.

What we assess was the difference between the two conditions to establish whether one of them is driving the RR-CC contrast more than the other, which is

$(\text{ForwPairs RwSeq} - \text{OtherPairs RwSeq}) - (\text{ForwPairs ConSeq} - \text{OtherPairs ConSeq})$
[contrast: 1 -1 -1 1]

Importantly, this test of a difference between differences is orthogonal. This can be shown by considering the dot product of the two contrasts:
 $[1\ 1\ -1\ -1] * [1\ -1\ -1\ 1] = 1\ -1\ +1\ -1 = 0$

This is why we included the test of the interaction in the Supplementary Materials in the previous round of revisions (Page 3 of the supplement)

“A direct test of the **interaction (which is orthogonal to ROI selection)** showed no interaction in amyg/hippo and pOFC (both $p > 0.5$), a trend-wise interaction in temporal pole ($p = 0.085$) and a significant interaction in mPFC ($p = 0.047$). In both of these latter cases, the interaction or trend-wise interaction arose because there was less, rather than more, suppression for forward pairs. Thus, in all cases, the rr-cc effect was not driven by only the subset of correctly ordered sequence pairs.”

This is a rationale that is used quite often, the summed (or merged) contrast across, in the simplest scenario, two conditions, is not correlated with the difference between the two conditions, and thus one can be used to test for the other in an unbiased fashion. We have added the following clarification to the paragraph above in the updated Supplement (page 3):

“While testing each effect individually (rr-cc for just ForwPairs or rr-cc for just OtherWithinPairs) is not orthogonal to our ROI selection, the interaction between the effect of rr-cc in ForwPairs compared to OtherWithinPairs is orthogonal to ROI selection. This is true in general because the mean of two effects is not correlated with, and thus independent of the difference between the same two effects. The direct test of the interaction...”

Nevertheless, we can also in principle follow the reasoning of the reviewer's argument above and we were intrigued whether in some cases, this bias between sum and difference of two effects can exist, as s/he suggests. To test this, we simulated some data. We simulated two conditions – one which contains a true signal (μ_1) and one which contains no signal (μ_2), see first row of subplots below. μ_1 is modelled as a two-dimensional gaussian with the peak reflecting a hypothetical BOLD peak and μ_2 is just a constant null effect. We examined in this simulation if selecting the peak of a main effect (mean of μ_1 and μ_2) biases the difference we observe between μ_1 and μ_2 at the location of this main effect.

In the first instance, we assumed that there would be noise in both measurements/conditions, i.e. μ_1 and μ_2 . In each of 10,000 iterations, we picked the peak of the mean effect across μ_1 and μ_2 and probed whether the difference at that peak over- or under-estimated the true difference in effects (because it is simulated data, we know the ground truth of the difference before adding noise). What is plotted in row 2 on the left hand-side of the below plot is the true difference vs the estimated difference at this peak for 10,000 simulations. The bottom row shows the measurement error for the difference effect at the peak of the mean effect. This shows that there is no bias to find a larger or smaller difference between conditions when selecting based on the peak of the mean effect across conditions.

In a second instantiation of this simulation, instead we only added noise to μ_2 , i.e. the condition with no effect. The same plots just described are shown for this simulation on the right in rows 2 and 3. Indeed, when only μ_2 has noise but the 'signal' (μ_1) does not contain any noise, the reviewer is right that picking the peak of the mean/sum of two effects does bias *against* finding a difference between them.

However, in standard fMRI analyses we assume that noise is unrelated to signal and thus we believe that the right assumption for BOLD data is to assume that there is noise in each measurement. The plots on the left convincingly show that the mean does not bias for or against finding a difference between the two components that contribute to the mean (here μ_1 and μ_2).

(2) Extinction test on p 9-10

The same logic then holds for testing the change across time, because it is just an extension of the scenario with two conditions. If we now imagine we had six conditions (in our case blocks), and we show that some effect is true across all six conditions (here across the six time points, we have an effect of sequence order). This test is orthogonal to the test of 'differences' between the conditions, which is what our 'extinction' test is asking (is there a change across time). You could have both or either or none of the two effects.

We hope this clarifies the reviewer's concern. Thanks again for your careful reading of our manuscript.

3. Finally, I don't get the logic of the new test under "Behavioral evidence for statistical learning," which seems to argue subjects learn the control sequence because their reaction times differ from B' to C' to D'. I don't see why it is the case that RT should progressively speed up along this sequence. Aren't they all equally expected once A' is observed?

Apologies for not explaining the rationale of this analysis. Progressive speeding when transitioning through an expected sequence has been observed before in the statistical learning literature, which is the main reason for using the same test in our manuscript. For example, Turk-Browne, Junge & Scholl showed the same effect here, a decrease in RT for later positions in learned triplets (https://ntblab.yale.edu/wp-content/uploads/2015/01/Turk-Browne_JEPG_2005.pdf ; see their Fig 5A pasted below).

Very similar plots are shown in Kim, ... Shams, Neurosci Letters, 2009, Fig 2A and 3A (https://faculty.ucr.edu/~aseitz/pubs/Kim_Seitz_Feenstra_Shams09.pdf), in Batterink & Paller, Cortex, 2017 (Fig 3B for syllables) (<http://faculty.wcas.northwestern.edu/~paller/Cortex2017.pdf>) and in Siegelman ... Frost, Cogn Sci, 2018 (<https://www.ncbi.nlm.nih.gov/pubmed/28986971>).

The reviewer is correct that each transition A'B', B'C', C'D' is experienced equally frequently. Nevertheless, it seems to be true in these and other data, and consistent with our intuition, that there is a RT speeding which may result from a reduction in uncertainty as participants transition further through the sequence.

We have added the above references to the paragraph on statistical learning as follows:

“We asked whether initiating the button press from A' to B', followed by B' to C' and C' to D' would show RT speeding because transitions, despite never being associated with reward, became increasingly predictable. Note that these transitions were experienced as frequently as those in the rewarded sequence. **Progressive RT speeding, in the absence of reward during pure statistical learning, has previously been reported. For example when learning triplets XYZ, the RT to Y after X is slightly faster and the RT to Z after XY is fastest³²⁻³⁵. Consistently, a 1x3 repeated-measures ANOVA with factor Transition (A'B'/B'C'/C'D') run only on the control sequence transitions showed a significant effect of transition ($F(2,50)=28.553$, $p<0.001$; Bayesian repeated-measures ANOVA shows evidence for a model with factor Transition: $BFm=1.522e+6$, $P(M|data)=1$).**”

Reviewer #2

The authors have adequately addressed my points. Thank you for a responsive revision.

Thank you very much. We have appreciated your time and thoughtful comments and we believe your suggestions have greatly improved our manuscript.

Reviewer #3

I commend the authors on their thorough efforts to address the reviewers' comments. I find the study interesting and compelling. The revised manuscript contains much rich information that I think will be of interest to the readership. I recommend publication.

Thank you very much for your thoughtful suggestions. We think they have helped to improve the manuscript in multiple ways, and we have greatly appreciated your time.

Reviewers' Comments:

Reviewer #1:

Remarks to the Author:

Thanks for the fulsome and thoughtful response to my final suggestions and concerns. I think the final article is very interesting indeed.